

# Development of a protocol for the auto-generation of explicit aqueous-phase oxidation schemes of organic compounds

Peter Bräuer[1, a], Camille Mouchel-Vallon[2, b], Andreas Tilgner[1], Anke Mutzel[1], Olaf Böge[1], Maria Rodigast[1, c], Laurent Poulain[1], Dominik van Pinxteren[1], Ralf Wolke[3], Bernard Aumont[2], and Hartmut Herrmann[1]

[1]Atmospheric Chemistry Department (ACD), Leibniz Institute for Tropospheric Research (TROPOS), Permoserstr. 15, 04318 Leipzig, Germany
[2]Laboratoire Interuniversitaire des Systèmes Atmosphériques, UMR CNRS/INSU 7583, Université Paris Est Créteil et Université Paris Diderot, Institut Pierre Simon Laplace, 94010, Créteil, France
[3]Modelling of Atmospheric Processes Department (MAPD), Leibniz Institute for Tropospheric Research (TROPOS), Permoserstr. 15, 04318 Leipzig, Germany
[a]Now at Wolfson Atmospheric Chemistry Laboratories and National Centre for Atmospheric Science, University of York, York, YO10 5DD, UK
[b]Now at National Center for Atmospheric Research, Boulder, Colorado, USA
[c]Now at Indulor Chemie GmbH & Co. KG Produktionsgesellschaft Bitterfeld, 06749 Bitterfeld-Wolfen, Germany

*Correspondence to*: Hartmut Herrmann (herrmann@tropos.de)

**Abstract.**

This paper presents a new CAPRAM/GECKO-A protocol for mechanism auto-generation of aqueous-phase organic mechanisms. For the development, kinetic data in the literature was reviewed and a database with 464 aqueous-phase reactions of the hydroxyl radical with organic compounds and 130 nitrate radical reactions with organic compounds has been compiled and evaluated. Five different methods to predict aqueous-phase rate constants have been evaluated with the help of the kinetics database: gas-aqueous correlations, homologous series of various compound classes, radical reactivity comparisons, Evans-Polanyi-type correlations, and structure-activity relationships (SARs). The quality of these prediction methods was tested as well as their suitability for automated mechanism construction. Based on this evaluation, SARs form the basis of the new CAPRAM/GECKO-A protocol. Evans-Polanyi-type correlations have been advanced to consider all available H-atoms in a molecule besides the H-atoms with only the weakest bond dissociation enthalpy (BDE). The improved Evans-Polanyi-type correlations are used to predict rate constants for aqueous-phase $NO_3$ + organic compounds reactions.

Extensive tests have been performed on essential parameters and highly uncertain parameters with limited experimental data. These sensitivity studies led to further improvements in the new CAPRAM/GECKO-A protocol, but also showed current limitations. Biggest uncertainties were observed in uptake processes and the estimation of Henry's Law coefficients as well as radical chemistry, in particular the degradation of alkoxy radicals. Previous estimation methods showed several deficits, which impacted particle growth.

For further evaluation, a mesitylene oxidation experiment has been performed at the aerosol chamber LEAK at high relative humidity conditions and compared to a multiphase mechanism using the MCMv3.2 in the gas phase and a methylglyoxal oxidation scheme of about 600 reactions generated with the new CAPRAM/GECKO-A protocol in the aqueous phase. While it was difficult to evaluate single particle constituents due to concentrations close to the detection limits of the instruments applied, the model studies showed the importance of aqueous-phase chemistry in respect to SOA formation and particle growth. The new protocol forms the basis for further CAPRAM mechanism development towards a new version 4.0. Moreover, it can be used as supplementary tool for aerosol chambers to design and analyse experiments of chemical complexity and help understanding them on a molecular level.





## 1 Introduction

The ubiquitous abundance of organic compounds in natural and anthropogenically influenced ecosystems impacts climate, air quality, human health, the oxidation capacity of the troposphere, crop yields, particle growth and composition. Their detrimental cause enormous economic effects cost (Boucher et al., 2013; Brasseur et al., 2003; Dunmore et al., 2015; Hallquist
et al., 2009; Pereira et al., 2018). With large annual emissions of about 1300 Tg C/a (Goldstein and Galbally, 2007), non-methane volatile organic compounds have been a research interest for many decades. Despite intense research efforts, the system of organic compounds is still insufficiently understood because of its complexity. The large emissions do not only lead to a large mass of organic compounds in the atmosphere, but also to a large variety of organic compounds. Currently, $10^4$ to $10^5$ different organic compounds have been identified, but are assumed to be only a small fraction of the actual number
(Goldstein and Galbally, 2007).

The multiphase nature of the oxidation of organic compounds further contributes to the complexity. Organic compounds are ubiquitously found in aerosol particles around the globe with contributions between 20% and 90% of the total aerosol mass (Jimenez et al., 2009). While primary organic aerosol (POA) is an additional source of organic compounds in the atmosphere to direct gas-phase emissions, gas-to-particle conversion, multiphase and heterogeneous processes form secondary organic
aerosol (gasSOA) and influence the composition in either phase (Ervens et al., 2011; Hallquist et al., 2009; Jimenez et al., 2009). With aerosol water being in excess of dry aerosol mass by a factor of 2 to 3 (Ervens et al., 2011), aqueous-phase chemistry plays an important role in the degradation of organic compounds. Organic aerosol oxidation state, size and product distribution as well as relative humidity dependence can only be explained by aqueous-phase chemistry forming secondary organic aerosol (aqSOA; Ervens et al., 2011).

Numerical modelling provides a useful tool for the analysis of such comprehensive and complex processes. Comprehensive benchmark mechanisms exist for either the gas phase (e.g., the Master Chemical Mechanisms, MCM, http://mcm.leeds.ac.uk/; Jenkin et al., 2003; Saunders et al., 2003) or the aqueous phase (e.g., the Chemical Aqueous Phase RAdical Mechanism, CAPRAM, http://projects.tropos.de/capram/ Herrmann et al., 2005, with latest updates by Bräuer et al., 2013). With a growing mechanism size in either phase of currently about 17,000 gas phase (MCM v3.3.1) and 777 aqueous-phase reactions
(CAPRAM 3.0), automated mechanism construction becomes increasingly important (see also Vereecken et al., 2018). This method has several advantages over manual mechanism construction. Among its key strong points are a reduced proneness to errors as any errors embedded in the generation routines produce systematic errors in the output chemical mechanisms, which are easier to detect than random errors in manual mechanisms. Most importantly, with suitable estimation methods for the prediction of the necessary kinetic and mechanistic data, it is possible to overcome the inevitable lack of experimental data for
the large number of organic compounds measured in field and laboratory experiments or produced by mechanism auto-generation. The detailed explicit mechanism produced with an auto-constructing method can help to interpret experimental data (e.g., La et al., 2016; Lee-Taylor et al., 2015; Mcvay et al., 2016). Modelling can suggest isomer information along with detailed information about the production and degradation processes.

Currently, atmospheric organic chemistry is more widely studied and understood in the gas phase than in the aqueous phase,
whether through experimental or modelling studies (Ervens, 2015). The current study aims to reduce this gap by providing a protocol for the auto-generation of comprehensive aqueous-phase mechanisms. While with the Generator for Explicit Chemistry and Kinetics of Organics in the Atmosphere (GECKO-A; Aumont et al., 2005), tools for mechanism auto-construction are already available for the gas phase, only first attempts have been presented for the aqueous phase for such an approach by Li and Crittenden (2009). Mouchel-Vallon et al. (2013) investigated a protocol for phase transfer processes
between the gas and the aqueous phase and effects on the composition in either phase. The current study takes the study by Mouchel-Vallon et al. (2013) a step further with a new protocol for mechanism auto-construction of chemical processes in the aqueous phase. A new protocol has been developed and incorporated in the expert system GECKO-A. Test mechanisms were generated and validated against experiments performed at the aerosol chamber LEAK (Leipziger Aerosol Kammer; Mutzel et



al., 2016). Beforehand, an extensive literature review of kinetic data and suitable estimation methods for the prediction of kinetic and mechanistic data for aqueous-phase reactions of organic compounds have been performed to ensure the generation of comprehensive, state-of-the-art aqueous-phase mechanisms.

## 2 Evaluation of kinetic data and prediction methods

For the construction of a protocol for mechanism auto-generation, a large database with kinetic data is a prerequisite. Data are used directly to assign rate constants to reactions created by the mechanism generator and indirectly to evaluate or advance current prediction methods for missing rate constants based on the known experimental values. In the present study, the aim of the kinetics database is to deliver a comprehensive set of reviewed rate constants in the aqueous phase for reactions of hydroxyl and nitrate radicals with a wide range of organic compounds. The recommendations of experimental data include

aromatic compounds, however, at the current state this compound class is excluded to evaluate prediction methods for mechanism auto-generation.

An extensive review and evaluation of available kinetic data from literature has been undertaken. Overall, a database with 464 aqueous-phase reactions of the hydroxyl radical with organic compounds and 130 reactions of the nitrate radical reactions with organic compounds has been created and is introduced in section 2.1. Tables with lists of the recommended values can be

found in the electronic supplementary material (ESM).

Secondly, the literature review was focused on already available prediction methods for radical reactions with organic compounds. Prediction methods investigated include gas–aqueous phase correlations, homologous series of various compound classes, reactivity comparisons between radicals, Evans-Polanyi-type correlations (including the BDE prediction method after Benson, 1976), and structure-activity relationships (SARs).

In the following sections, these prediction methods are discussed for the OH and the $NO_3$ radical with a focus on the applicability in mechanism auto-generation.

### 2.1 Evaluation of experimental kinetic data

Several comprehensive compilations and recommendations exist for aqueous-phase kinetic data (e.g., Buxton et al., 1988; Ross et al., 1998; Herrmann, 2003; Herrmann et al., 2010). Experimental data relevant for tropospheric chemistry of organic

compounds were collected from these compilations and assembled in a comprehensive kinetics database. Moreover, kinetic data from further laboratory experiments were collected from literature and, in exceptional cases where no other data existed, from unpublished TROPOS measurements. Rate constants were carefully reviewed. In general, already recommended values from compilations were taken over from literature. Further experimental data (including more recent data, where no recommendations existed) were selected by actuality, but also quality of the data. If no clear preference was obtained, averaging

was used for the recommendation of the kinetic data. As a result, a database was created with a total of 464 reactions of OH radicals with organic compounds and 130 reactions of $NO_3$ radicals with organics. Recommended kinetic data can be found in the ESM in Table S1 and S2, respectively. Organic compounds include alkanes, alcohols (including di- and polyols), carbonyl compounds, carboxylic acids and diacids, esters, ethers, unsaturated, cyclic, aromatic compounds, and polyfunctional compounds composed of these functionalities. So far, only the reactions of aliphatic compounds (including 282 OH and 76

$NO_3$ reactions) have been used to design a protocol for automated mechanism construction. Besides the use of the database to assign kinetic data to the constructed mechanisms, it is also used for testing and advancing the prediction methods. Those prediction methods are introduced in the next sections.



### 2.2 Simple correlations

A number of prediction methods exist, which include straight-forward correlations that are easy to implement in automated tools for mechanism generation. These correlations include gas-aqueous phase correlations, extrapolations of homologous series of compound classes, and reactivity comparisons between different radical oxidants.

These methods were therefore evaluated for their use in mechanism generators. Correlations were derived for the various compound classes and absolute errors are analysed with box plots. Results from the analysis are shown in Figure 1, while a detailed analysis is given in section S2 of the ESM. For such simplistic approaches, good results are obtained from these methods and one can expect to at least predict the correct order of magnitude for a given rate constant (see Figure 1). In general, slightly negative absolute errors are observed in Figure 1, thus, the predictions tend to overpredict the rate constants somewhat.

The largest deviations of the absolute errors of the logarithmised rate constants range from -2.1 to 0.9. Simple structures like pure hydrocarbons are predicted far more accurately than more complex structures. Largest deviations are observed for carbonyls and carboxylic acids.

While the easy implementation of the above correlations in computer-assisted tools make their usage desirable, they disqualify for several reasons. Overall, their accuracy is satisfactory, but can be improved. More importantly, all of the above correlations

have a very restricted application. To predict aqueous-phase rate constants by correlating gas- and aqueous-phase kinetic data, gas-phase kinetic data is needed. Gas-phase kinetic measurements are not sufficient to support this method in complex mechanism generators such as CAPRAM/GECKO-A. Gas-phase kinetic data would have to be estimated in order to apply this method for mechanism auto-generation, which would increase the errors significantly. Extrapolations of homologous series of compound classes is limited to very defined structures, such as terminal mono-alcohols or diols. The restrictions are best

demonstrated by the simplest compound class of alkanes. While extrapolations are very accurate, one has to carefully revise the existing data to derive accurate predictions. Different regression lines have to be used for linear and branched alkanes. Furthermore, often the smallest members of a compound class have to be excluded to derive best results of the correlation lines as in the case of carboxylic acids. For alkanes, methane is excluded in the evaluation of homologous series in Figure 1 as its absolute log error of 16.8 does not fit in the scale of the plot. Dicarboxylic acids follow a sigmoidal rather than a linear

regression as explained in section S2.2 of the ESM. The regression yields negative rate constant values for the smallest members oxalate and malonate. Therefore, these species have been excluded in the evaluation of homologous series in Figure 1. Combined with the fact that only a small dataset of aqueous rate constants exists, overfitting is likely and the predicted correlations should be handled with care.

In reactivity comparisons, where the rate constants of different radical oxidants in the aqueous phase are compared and

correlated, a large kinetic data set with a sufficient intersect of organic reactants for the radical oxidants is needed to derive reliable correlations. While this goal is already only partially achieved for a limited set of structures, the method completely disqualifies for mechanism auto-generation as kinetic measurements are of one radical oxidant are needed to predict rate constants of another. Kinetic datasets are too small to ensure a functional predication in automated mechanism generators. An analysis with box plots was not performed due to the limited data points. Due to their limited applicability, the methods are

briefly described in the ESM, correlations are provided, but are not considered for use in CAPRAM/GECKO-A.

### 2.3 Evans-Polanyi-type correlations

More advanced correlations for the prediction of rate constants of organic compounds are Evans-Polanyi correlations. Originally developed for the gas phase by Evans (1938), they have been successfully applied to the aqueous phase (see, e.g., Herrmann and Zellner, 1998; Hoffmann et al., 2009). In an Evans-Polanyi-type correlation, a linear relationship between the

activation energy and the bond dissociation enthalpy (BDE) of a molecule is derived by:

$$E_A = a' + b' \cdot BDE(C-H) \tag{1}$$





Using the Arrhenius expression ln $k = A \cdot \exp(-E_A/RT)$ and assuming only the weakest H-atoms in molecule will be abstracted during a reaction leads to:

$$\log(k_H) = \log\left(\frac{k_{2nd}}{n_H}\right) = a - b \cdot BDE(C-H) \qquad (2)$$

where $a = \log(A/n_H) - a'/(RT\ln 10)$ and $b = b'/(RT\ln 10)$, $n_H$ is the number of H-atoms with the lowest BDE in a

molecule, $R$ is the universal gas constant, and $T$ is the temperature. $A$ is the pre-exponential factor of the Arrhenius equation, which has to be similar for all reactants correlated to fulfil equation 2. $k_H$ is the second order rate constant of the reaction scaled by the number of weakest C–H bonds. The method relies on BDEs as input for the correlation, which can be estimated reliably with a precision of ±8 kJ mol⁻¹ (Benson, 1976). The method by Benson (1976) estimates the strength of a bond as a function of the adjacent atoms/molecular groups.

**2.3.1 OH rate constant prediction**

The kinetics database was used to derive Evans-Polanyi-type correlations, which are plotted in Figure S5 in the ESM. Parameters for the regression equations and further statistical data are given in Table S8 in the ESM. Even for a large database as used in this assessment, a reliable correlation is hardly achieved (see Figure S5 in the ESM). Several outliers can be found in the dataset. A critical evaluation of the data is necessary and depending on this evaluation, several possible correlations can

be found. Furthermore, there are several cases where data points have exactly the same BDE, but vary in their reaction rate constants by more than one order of magnitude. The reason for this behaviour is the correlation of the rate constants to only the weakest BDE, which corresponds to the main radical attack site in a molecule or the main reaction channel. When correlating a homologous series of compounds to the bond dissociation enthalpy, the lowest BDE in each molecule varies only slightly. On the other hand, rate constants increase significantly with the carbon number due to side attacks of the radical

oxidants at the remaining carbon skeleton. With increasing chain length, side attacks become more likely even if the BDE is higher compared to the weakest bond. With the correlation of $\log(k_H)$ against only the smallest BDE of a molecule, this fact cannot be considered in the correlation.

Accordingly, if the Evans-Polanyi correlations are used this way, large uncertainties of the prediction method are resulting. The evaluation of the absolute errors of the predicted versus experimental data with box plots confirms the discussion above.

Relatively large errors are seen for carbonyl compounds and carboxylic acids while errors are somewhat smaller for alcohol compounds and alkanes (see Figure 2). The performance is not significantly better than the performance of the much simpler correlations described in the preceding section. However, a major advantage of Evans-Polanyi-type correlations is the much wider applicability.

For further details, the prediction method has been re-analysed with the help of scatter plots, where the predicted data derived

from the correlations is plotted over the experimental data. In a perfect correlation, all data should line up on the 1:1 line with a correlation coefficient $R^2$ of 1 in such a plot. Any deviations of the y-intercept of the regression line from 0 or the slope of 1 indicate a general bias of the prediction method. Reductions of the correlation coefficients are due to deviations of the single predictions.

Figure 3 shows that, except for the pure hydrocarbons, rate constants of fast reacting compounds are underpredicted while rate

constants of smaller compounds are overpredicted. This fact demonstrates the limitations of the current Evans-Polanyi-type correlations to accurately predict rate constants of larger compounds as it was already seen during the derivation of the correlation. Faster rate constants in the dataset belong to larger compounds. Hence, the increase of their observed rate constants is larger than the increase expected from their lowest BDE due to increased side attacks of the hydroxyl radical at the remaining carbon skeleton. As these side attacks are not considered in the correlation, where only the weakest bound H-atoms of the

major reaction pathway are correlated against $\log(k_H)$, an underprediction is inevitable. As a consequence, the regression line of the correlation is tilted resulting in an overprediction of smaller or slow reacting compounds.





### 2.3.2 NO₃ rate constant prediction

The much smaller dataset of available NO$_3$ reactions of aliphatic compounds further reduces the accuracy of Evans-Polanyi-type correlations for reactions of organic compounds with this radical oxidant. It is considerably harder to derive reliable correlations, which can be achieved only for alcohol compounds and carboxylic acid with appreciable correlation coefficients

$R^2$ between 0.6 and 0.7. Correlations are shown in Figure 4a with the respective statistical data given in Table S9 in the ESM. Predictions of NO$_3$ reactions with organic compounds inherit large errors. For a large fraction, the right order of magnitude is not even predicted. Again, an underprediction of fast reactions (or large) molecules is observed, while the rate constants of smaller or slow reacting compounds are overpredicted (see Figure 4a) as discussed for OH in the previous subsection.

### 2.3.3 Advanced Evans-Polanyi-type correlations

The evaluation of current Evans-Polanyi-type correlations has shown strong limitations for the use of automated prediction methods, which aims at large molecules, where the formation of many intermediate compounds makes a manual mechanism construction infeasible. As the accuracy of the predictions of rate constants for these compounds is significantly reduced, a revision and improvement of the current correlations are needed and are described in this subsection.

**Development of improved Evans-Polanyi-type correlations**

The need to improve Evans-Polanyi-type correlations for larger compounds, where minor reaction channels become increasingly important as outlined in the preceding subsections, led to a revised correlation method, where the sum of all BDEs of bonds including hydrogen atoms ($\sum$BDE) was correlated against the overall second order rate constant:

$$\log(k_{exp}) = a - b \cdot \sum BDE \tag{3}$$

The consideration of minor pathways is achieved by including all BDEs rather than only the weakest ones and the overall second order rate constant instead of the second order rate constants scaled to the number of weakest H-atoms. A certain disadvantage of this method is that information about branching ratios and reaction products is lost.

*OH radical rate constant predictions*

Compared to previous Evans-Polanyi-type correlations, significant changes are observed as can be seen from Figure 5. The range of $\sum$BDE is much broader ranging from about 500 to 8000 kJ mol$^{-1}$ compared to a BDE range of about 360 – 440 kJ mol$^{-1}$. More importantly, in the new correlation there is a positive correlation between $\sum$BDE and $k_{exp}$. The positive correlation

derives from the fact that $\sum$BDE is correlated to all abstractable H-atoms and that $k_{exp}$ increases with larger molecules, i.e. an increasing number of H-atoms or $\sum$BDE. In the original Evans-Polanyi-type correlations, a negative correlation is observed due to higher reactivities at molecular sites with low BDEs. The most striking difference are quadratic correlations in the advanced Evans-Polanyi-type correlations compared to linear correlations previously. In the new correlations, rate constants plotted over $\sum$BDE follow downward-opened parabolas. All data fits with in an upper and lower parabola, which converge

towards $\log(k_{exp} / M^{-1} s^{-1}) = 10$ and $\sum$BDE = 8000 kJ mol$^{-1}$ in the $\log(k_{exp})$ over $\sum$BDE diagram.

A likely reason for the different behaviour of the old and the new regressions is the diffusion limit of reactions. The diffusion limit does not come into effect in the old correlation, where the partial rate constant ($k_H$) refers to only the weakest bound H-atoms. When the overall rate constant is considered, the diffusion limit is reached for high values of $\sum$BDE. Thus, despite an increasing $\sum$BDE due to an increasing number of abstractable H-atoms, the rate constants might not increase accordingly as

they are levelling off and undergo a transition to control by diffusion rather than by chemical control. This view is supported by the upper limit of all data around $\log(k_{exp} / M^{-1} s^{-1}) = 10$, the approximate diffusion limit of rate constants of OH with organic compounds (see, e.g., Haag and Yao, 1992; Schöne et al., 2014).





The correlations show significant improvements with correlation coefficients $R^2$ up to 0.99 (see Figure S6 and Table S10 in the ESM). A weak correlation is still observed for ketones ($R^2 = 0.24$) and alcohol compounds except linear terminal alcohols ($R^2 = 0.35$). Monocarboxylic acids show only a moderate correlation ($R^2 = 0.55$). The better performance can also be evaluated from the absolute errors analysed as box plots in Figure 6. Almost all data are predicted with the correct order of magnitude.

The weak correlations of the above-mentioned compound classes are reflected by larger errors. More importantly, the objective to reduce biases of larger molecules was achieved as can be observed from the scatter plot of the calculated over experimental data in Figure S7 in the ESM. Except for ketones, regression lines now have a slope close to 1 (between 0.76 and 1.16). Omitting the outlier acetylacetone in the analysis of ketones gives a slope of the regression line of 0.54, which is still a big improvement compared to the original correlation. Although an omission of data points for the derivation of a correlation is

not ideal, a large modified Z score after Iglewicz and Hoaglin (1993) (see equation 4) of 4.21 justifies the removal of the data point from the dataset. Acetylacetone is the only compound, which has a modified Z score well above the suggested threshold of 3.5 (Iglewicz and Hoaglin, 1993).

$$Z_i = \frac{0.6745(x_i - \tilde{x})}{MAD} \quad (4)$$

In equation 4, $x_l$ is the individual measurement, $\tilde{x}$ the median, and MAD the median absolute deviation.

*NO₃ radical rate constant predictions*

When deriving the advanced Evans-Polanyi-type correlations for nitrate radical reactions with organic compounds, a major
difference to hydroxyl radical reactions is observed. $NO_3$ reactions follow a linear trend rather than a quadratic correlation. The additional quadratic polynomial does not reduce errors significantly. An explanation for the different behaviour can be found again in the diffusion limit of both reactions. The diffusion limit of nitrate radical reactions is only insignificantly smaller compared to hydroxyl radical reactions ($\sim 9.5 \cdot 10^9 \, M^{-1} \, s^{-1}$ vs. $\sim 10^{10} \, M^{-1} \, s^{-1}$). However, rate constants for nitrate radical reactions are in the order $10^4 - 10^8 \, M^{-1} \, s^{-1}$. Most reactions are below 1% of their diffusion-controlled limit. Therefore, the diffusion limit
does not affect the correlation. Even though a quadratic correlation could be applied, values are so far away from the maximum of the parabola, that a linear equation is an equally valid approximation.

The correlations are shown in Figure 7 with the raw data in subfigure (a) and the final correlations in subfigure (b). The respective statistical data is provided in Table S11 and S12 in the ESM. Both linear (solid lines) and quadratic regressions (dashed lines) are shown in Figure 7b. Another advantage of the advanced Evans-Polanyi-type correlations is that certain
compound classes can be grouped together without a loss of accuracy. This increases the number of data points used for each regression and, therefore, its reliability. For the derivation of the final correlations, data from tert-butanol and acetone have not been considered due to their high modified Z scores of 5.55 and 4.57, respectively, well above the suggest threshold of 3.5 (Iglewicz and Hoaglin, 1993).

As can be seen from Figure 7b and the statistical data in Table S11 and S12 in the ESM, only a weak correlation with large
associated errors is observed for carboxylic acids. Moreover, for this compound class, a negative correlation is observed for which the reasons remain unclear. More data are needed to constrain a more reliable correlation for these types of molecules and their respective $NO_3$ reactions.

Overall, the advanced Evans-Polanyi-type correlations perform well with significantly reduced errors as can be seen from the box plot analysis in Figure S8 in the ESM. The correct order of magnitude is met for almost all compounds and usually errors
are much smaller. Only one carboxylic and two dicarboxylic acids are not predicted within the correct order of magnitude. Biases of the slopes of the regression lines in the scatter plot of the calculated over experimental data are significantly reduced except for the very uncertain correlation of carboxylic acids (see Figure S7 in the ESM).





### 2.4 Structure-activity relationships

A more sophisticated prediction of rate constants of the hydroxyl radical with organic compounds can be achieved by means of structure-activity relationships (SARs). In an SAR, the assumption is made that a molecule can be split into increments and the overall reaction rate is the sum of the individual reaction rates of each increment. Rate constants are assigned to the individual increments, which are modulated by the effects of the neighbouring groups in α- and, in the case of the SAR by Doussin and Monod (2013), in β-position as well as ring effects of the molecule. Two structure-activity relationships from literature have been compared – the SAR by Minakata et al. (2009) and the one by Monod and Doussin (2008) with updates by Doussin and Monod (2013).

### 2.4.1 SAR by Minakata et al.

A big advantage of the SAR of Minakata et al. (2009) is the large dataset of 434 aqueous-phase reaction rate constants covering all important compound classes for tropospheric aqueous-phase chemistry including alkanes, alcohols, carbonyl compounds, carboxylic acids, ethers, esters, sulphur-, nitrogen- and phosphorous containing species, polyfunctionals, unsaturated compounds, and aromatics. Their dataset was split in a training set of 310 reactions to derive the SAR and a test set of 124 reactions for validation. The large dataset enables a robust prediction of aqueous hydroxyl radical reaction rates with all organic compounds of atmospheric relevance. The authors were able to achieve a high accuracy and predict rate constants of 83% of the compounds in their training set and 62% of the compounds in their test set within a factor of 2.

### 2.4.2 SAR by Monod and Doussin

The structure-activity relationship by Monod and Doussin (2008) covers the prediction of OH rate constants for alkanes, alcohols, carboxylic acids and bases as well as polyfunctional compounds derived from these functions. 72 compounds were used to derive the SAR. A major improvement of the SAR by Monod and Doussin (2008) is the introduction of a second correction factor for the incremental rate constants taking the effects of the β-neighbours into account. However, this improvement is also one of its biggest weaknesses as over-fitting becomes likely. With the dataset of 72 compounds, 22 descriptors have been defined. Moreover, a wide range of atmospherically relevant compounds is still missing and currently only H-abstraction reactions are possible to predict with this SAR. Yet, high accuracy is achieved and according to their own validation, Monod and Doussin (2008) predicted 60% of their compounds tested within the range of 80% of the experimental value.

The original SAR by Monod and Doussin (2008) was extended to include carbonyl compounds by Doussin and Monod (2013). The original parameters were kept constant and new parameters have been introduce to describe the partial rate constant at aldehyde groups and α- and β-effects of carbonyl (keto and aldehyde) groups. Moreover, a modulating factor for gem-diol groups in hydrated carbonyls was introduced.

### 2.4.3 Comparison of both SARs

In general, both SARs by Minakata et al. (2009) and Monod and Doussin (2008) with updates by Doussin and Monod (2013) are suitable tools for automated rate constant prediction of aqueous-phase hydroxyl radical reactions with organic compounds with the above-mentioned strengths and restrictions. The accuracy is high and implementation in computer-assisted tools is easy. All parameters are delivered by the structure-activity relationships and the only input variable is the chemical structure of the organic compound. Both SARs were tested using the kinetics database described in section 2.1. They show a very good agreement with the experimental values, especially for simple molecules (see scatter plots in Figure 8). Best results are achieved for pure hydrocarbons and errors increase when introducing substituents. Largest errors occur for polyfunctional compounds. The box plots of the absolute errors of both SARs compared to the experimental values demonstrate these facts



(see Figure 9). More detailed information on the statistical data of the evaluation for every compound class can be found in the review by Herrmann et al. (2010).

The box plots in Figure 9 show that both SARs have difficulties predicting the hydroxyl radical rate constants for dicarbonyl compounds (compound class 5). There is a large variation of about 3 orders of magnitude in the absolute errors between the minimum and the maximum value. Both SARs underpredict the rate constants. The difficulties may be caused by the small dataset of only 5 dicarbonyl compounds.

In general, errors are smaller in the SAR by Doussin and Monod (2013) due to the second descriptor for β-neighbours, however, it is noted again that over-fitting might be a problem in the SAR of these authors. Errors in both SARs are very small and besides a few exceptions well within one order of magnitude.

## 3 Sensitivity runs of crucial parameters

Based on the evaluation of the kinetic data and prediction methods for aqueous-phase rate constants a preliminary protocol has been designed for mechanism auto-generation. The protocol uses structure-activity relationships for hydroxyl radical reactions with organic compounds and the linear correlations of the refined Evans-Polanyi-type correlations for nitrate radical reactions with organic compounds. For uptake, the structure-activity relationship GROMHE (GROup contribution Method for Henry's law Estimate, Raventos-Duran et al., 2010) was used and dissociations of carboxylic acids have been estimated with the method by Perrin et al. (1981). Fixed rate constants and branching ratios have been used for radical species. A more detailed description of all processes implemented in the protocol in its final state can be found in Table 3 in section 4.

With the preliminary protocol, test mechanisms have been designed and evaluated focusing on the influence of critical parameters or the quality of estimates of parameters, where the scarcity of data allowed only rough estimates. All sensitivity studies have been performed with the parcel model SPACCIM (SPectral Aerosol Cloud Chemistry Interaction Model; Wolke et al., 2005) in a general, non-specific urban environment. Model runs are for 4.5 days at 45ºN in mid-June. The trajectory of the parcel model involves 8 cloud passages for about 2 hours at noon and midnight of each day (marked by a blue shaded area in the results plots given in the following). This scenario allows a detailed investigation of either particle or cloud chemistry and interactions between both (see Tilgner et al., 2013 for further information on the model scenario).

### 3.1 Degradation of polycarbonyl compounds

In a first sensitivity test, the formation of polycarbonyl compounds mostly in their hydrated form was observed in high concentrations. Diol functions were found at every site of the molecule, which is an unstable structure and should decompose thermally and, therefore not exist in high concentrations. Hence, a decay of polycarbonyl compounds by C–C bond breaking has been introduced to the protocol when at least three carbonyl groups (hydrated or unhydrated) are found within four adjacent carbon atoms. The rule is based on considerations in the gas-phase mechanism MCMv3.3 (Master Chemical Mechanism, http://mcm.leeds.ac.uk/MCMv3.2/), where in the n-alkane oxidation series no more than 3 carbonyl groups are found within four adjacent carbon atoms independently of the chain length of the parent molecule.

For the estimation of the decay rate, sensitivity studies have been performed, where the first order rate constant was varied between 0.01 and 1 s$^{-1}$. Moreover, the importance of OH radical induced oxidation of polycarbonyl compounds and the monomolecular decay has been investigated by allowing only molecular decay in the protocol or by treating both processes in parallel. The set of sensitivity studies was compared to the base case, which treats only the OH attack of polycarbonyl compounds. A short description of all sensitivity studies and the nomenclature used in the following is given in Table 1.

In the sensitivity runs, GECKO-A with a preliminary aqueous-phase protocol was used to extend CAPRAM 3.0n. The generator was used to revise the chemistry of all stable organic compounds in CAPRAM 3.0n and introduce branching ratios for their degradation reactions. The preliminary protocol was applied to all newly evolving intermediates, but was stopped at



radical species, which were already previously defined in CAPRAM, to reduce the size of the mechanism. Besides a more detailed chemical scheme due to the treatment of minor reaction pathways in contrast to most reactions in the previous CAPRAM version, the hydration of carbonyl compounds is treated with greater complexity allowing the hydration of keto functionalities according to the structure-activity relationship GROMHE.

Figure 10 shows the concentrations of 2-oxo-3-hydroxy-succinaldehyde, which illustrates the effects of the new protocol as it was already part of CAPRAM 3.0n. In this Figure and all other Figures, a '∑' before a species names or chemical formula in the unhydrated and protonated form means that the sum of the concentrations of all hydration and dissociation states of a given species are shown. Hydration is only considered in the sensitivity runs with the new protocol rules. Figure 10 shows that negligible concentrations are reached for the highly oxidised polycarbonyl compound for rate constants of at least $0.1 \text{ s}^{-1}$.

Additional oxidation by hydroxyl radicals seems negligible as concentrations of the sensitivity runs with and without oxidation by OH radicals are indistinguishable from each other. However, product distributions are affected by the additional oxidation of OH radicals as can be seen from Figure 11. The two acids shown in Figure 11a/b are the main degradation product of the OH initiated oxidation. While bond breaking reduces their concentrations compared to the base run with no monomolecular decay of polycarbonyls (woBB), this effect is counteracted in the runs with additional oxidation treated in the protocol

especially for the run with the slow decay rate of $10^{-2} \text{ s}^{-1}$ (BBe-2+OH). Figure 11c/d show the concentration-time profiles of oxo-lactic acid and tartronic acid, the major first and second-generation products of the monomolecular decay of 2-oxo-3-hydroxy succinaldehyde. It can be seen that the order of magnitude of the reaction rate constant has little effect on the concentrations of these products. Moreover, additional oxidation by hydroxyl radicals does not affect the formation of oxo-lactic acid and tartronic acid.

Finally, the implemented additional monomolecular decay of polycarbonyl compounds does not significantly increase the oxidation capacity of the aqueous phase. Macroscopic particle properties such as pH or organic particle mass are not affected by the additional decay channel. However, the high concentrations of highly oxidised and thermally unstable polycarbonyl compounds can be effectively suppressed with this new reaction type in the protocol. Therefore, it has been added to the final protocol. A decay rate of $0.1 \text{ s}^{-1}$ was chosen as an ideal compromise between the suppression of high concentrations of

polycarbonyls and an increased oxidation capacity of the aqueous phase. For complete description in the final aqueous-phase protocol of GECKO-A, additional oxidation by hydroxyl radicals is considered as well to determine the product distribution with the best current knowledge.

### 3.2 Influence of the chosen SAR

SARs form the basis of the protocol for automated mechanism generation. Therefore, their influence on the oxidation

mechanism and modelled concentrations of important constituents of the tropospheric aqueous phase has been investigated thoroughly. Figure S9 in the ESM shows the concentrations of selected organic compounds in the aqueous phase. Concentrations of compounds whose major source is the uptake from the gas phase are in excellent agreement for the two investigated SARs by Minakata et al. (2009) and Doussin and Monod (2013). However, significant differences in concentration-time profiles can be observed for second or older generation products mainly produced by aqueous-phase

processes (see Figure S9).

This behaviour is explained by two facts. First, errors in the SARs increase with increasing complexity of the molecule and, hence, with more oxidised compounds in the aqueous phase. More importantly, this behaviour is a result of the excellent prediction of the overall rate constants by SARs in general, but significant uncertainties in the predicted products of aqueous-phase SARs due to missing experimental determinations of products and their branching ratios. Therefore, significant

differences can be observed in the branching ratios while the rate constants of the overall decay compare well. More experimental data is needed to derive more reliable product distributions in aqueous-phase SARs. The macroscopic properties



of particles such as pH and organic particle mass are not affected by these uncertainties (see Figure S9 in the ESM), but significant differences in the prediction of the concentrations of single species might be observed between the two SARs.

### 3.3 Processing of the organic mass fraction

As the focus of this study is the aqueous-phase processing of organic particulate matter, a more detailed investigation has been performed for an urban environment with the improved mechanism from section 3.1. From Figure 12, two deficits of the current protocol can be seen. For such a polluted environment the growth of the organic mass fraction is too small. A significant mass increase of 166 $\mu g\ m^{-3}$ is observed in the base model run as seen in Figure S10 of the ESM, while the organic mass increases by only 1.7 $\mu g\ m^{-3}$ contributing less than 5% to the total mass at the end of the model run. This is significantly less than observed all around the world (Jimenez et al., 2009). The organic mass increase is significantly less than observed previously (Brégonzio-Rozier et al., 2016; Ervens and Kreidenweis, 2007). Secondly, unrealistically high concentrations of organic nitrates are observed during non-cloud periods. During non-cloud periods, the majority (>80%) of the particle mass of the explicit scheme (excluding the generic model species HULIS, WSOC, and WISOC) consists of organic nitrates. In general, organic nitrate fraction of up to 20% are observed (Day et al., 2010) and up to 45% in (submircron aerosol and PM1; Bean et al., 2016; Kiendler-Scharr et al., 2016). To address these issues, further investigations were performed on the processing of the organic particulate matter with a focus on organic nitrates. A series of test studies were performed, which led to a significant improvement of the description of the processing and the composition of organic particulate matter. These studies are described in the following. The best estimates of each test scenario are taken over and included in the following model runs unless stated otherwise.

### 3.3.1 Proxy reactions for OH and NO₃ radical reactions with residual OM

Despite its great detail, the mechanism generated by the preliminary protocol for these sensitivity studies is still limited as it only treats the chemistry of compounds with up to four carbon atoms explicitly. Therefore, radical concentrations might be overestimated due to missing scavenging by larger organic compounds, which further affects important particle properties and particle growth. Therefore, proxy reactions for the processing of the generic water-soluble organic compounds (WSOC) and humic-like substances (HULIS/HULIS⁻) have been introduced to CAPRAM 3.0n to account for the interactions of hydroxyl and nitrate radicals with longer-chained organic compounds. The generic model species WSOC, HULIS/HULIS⁻ as well as water-insoluble organic matter (WISOC) are referred to as residual organic matter (residual OM) in the following. All proxy reactions are given in Table 2. Besides the oxidation by OH and NO₃, photolysis of humic-like substances is included to account for an accurate description of the iron(II) to iron(III) ratio.

The new proxy reactions decrease radical concentrations as expected (see Figure S11 in the ESM for OH and NO₃ concentrations in the run WSOC). OH radical daytime concentrations are decreased by more than one order of magnitude and nitrate radical concentrations by almost an order of magnitude by the end of the simulation after 4.5 simulated days. Consequently, particle growth increases as can be seen from Figure 13. In this plot, the total organic particle mass is given as well as the mass of all organic nitrates and the residual OM for the runs with the initial mechanism from section 3.1 and the sensitivity run (WSOC). As the sensitivity runs focus on non-cloud periods, high concentrations during cloud periods are not shown in Figure 13 and all other Figures of this subsection for a better resolution of the differences during non-cloud periods. Figure 13 shows that previously particle growth was observed only during cloud periods, while the organic mass during non-cloud periods decreased. With decreased OH and NO₃ radical concentrations from the WSOC and HULIS proxy reactions, less oxidation is observed. The reason for this behaviour is that the current protocol consists mainly of fragmentation reactions, which produce mainly high volatility products. Thus, the decreased oxidation in the WSOC runs suppresses the production of volatile organic compounds and reduces their release to the gas phase. As a consequence, the observed decrease in organic particle mass during non-cloud periods is less with the proxy reactions for WSOC and HULIS species.



These sensitivity runs also demonstrate a weakness of the current protocol, which only includes oxidative processes. Non-oxidative processes, such as accretion reactions are currently not considered. This is partly owed to very limited experimental data and partly due to the fact that non-oxidative processes will rapidly lead to very large products, which require large explicit oxidation schemes. With the currently very comprehensive protocol, the mechanism size would increase beyond the capabilities of current numeric models. Furthermore, fragmentation seems overestimated in the current protocol, a fact that is addressed in another sensitivity study introduced in section 3.3.3. While the new proxy reactions help to increase organic particle growth rates to more realistic values, they do not affect particle composition. Therefore, the overestimation of organic nitrate concentrations remains. Therefore, further tests are introduced in the next subsection to investigate the formation and degradation of organic nitrates and identify missing processes in the current protocol.

### 3.3.2 Detailed studies of organic nitrate sinks and sources

Further tests have been performed on the sinks and sources of organic nitrates to investigate the cause of their high concentrations in the model simulations. In the current protocol, photolysis is excluded due to the very limited data, which makes it hard to determine rules for automated prediction of photolysis rates. To evaluate effects on the processing of organic nitrates in the aqueous phase, photolysis has been included for this compound class leading to alkoxy radicals and $NO_2$:

$$RONO2 \rightarrow RO + NO_2 \tag{5}$$

For estimating a maximum photolysis efficiency, the photolysis rate was estimated twice as high as gas-phase photolysis of typical organic nitrates. This estimate recognises the lens effects deriving from the spherical shape of cloud droplets and particles (e.g., Mayer and Madronich, 2004), but fully neglects any solvent cage effects to have an upper limit of the sink effects of photolysis on organic nitrate concentrations in the aqueous phase. For neutral photofragments, effective quantum yields similar to their gas-phase counterparts have been discussed before (Herrmann, 2007).

The scenario 'Phot' in Figure 14 indicates that effects of photolysis in a sensitivity run with these additional photolysis reactions are minor even with the upper limit estimates described here. Only little reduction in organic nitrate concentrations is resulting and these compounds remain the dominant constituents of the particle phase. Therefore, photolysis for organic nitrates has been neglected in the final protocol (and the following sensitivity studies) to be more consistent with the other compound classes and until the development of a more vigorously tested photolysis protocol for all chromophores.

Further investigation focused on possible overestimations of the sources of organic nitrates. The only aqueous-phase source of organic nitrates besides possible uptake from the gas phase is the addition of nitrate radicals to double bonds of unsaturated organic compounds. Therefore, a sensitivity study has been performed with reduced rate constants of the addition reaction. In the original protocol, different estimates for different compound classes were made based on experimental determinations by Alfassi et al. (1993) as given in Table S2 in the ESM. Their measurements indicate high reaction rate constants in the order of $10^9$ M$^{-1}$ s$^{-1}$ for unsaturated compounds and rate constants in the order of $10^7$ M$^{-1}$ s$^{-1}$ for carboxylic acids. However, recent measurements by Schöne et al. (2014) indicate reduced rate constants for unsaturated compounds. Therefore, rate constants of unsaturated compounds with nitrate radicals were restricted to $10^7$ M$^{-1}$ s$^{-1}$ in another sensitivity study labelled with 'NO3add' in Figure 14.

The reduction of the rate constants of $NO_3$ addition reactions significantly reduces the mass fraction of organic nitrates, but consequently also suppresses particle growth (see scenario 'NO3add' in Figure 14). The described attempts have been unable to resolve the opposing effects of overestimated organic nitrate concentrations and underestimated particle growth in an accurate description of the processing of the organic particle mass. Therefore, further investigations were performed aiming at the phase transfer in the aqueous-phase protocol of GECKO-A.





### 3.3.3 Phase transfer of oxygenated organic compounds

GECKO-A uses the structure-activity relationship GROMHE (Raventos-Duran et al., 2010) to estimate Henry's Law coefficients (HLCs) needed to describe the uptake process. However, recent research indicates that due to the data used to derive the structure-activity relationship, HLCs might be underestimated in GROMHE (Compernolle and Müller, 2014b,

2014a). To address a possible overestimated release of organic particulate constituents to the gas phase, a sensitivity study ('O:C=1') has been performed, where the HLC of all compounds with an O:C ratio greater or equal to one has been set to a fixed value of $1 \cdot 10^9$ M atm$^{-1}$.

Results are shown in Figure 15. The new estimates increase the organic mass concentration during non-cloud periods to values between 9 and 13 µg m$^{-3}$. A noticeable difference is the faster particle growth during non-cloud periods in the model run with

the revised uptake, reaching the (non-cloud) peak value of 13 µg m$^{-3}$ directly after the daytime cloud on the second model day. Thereafter, peak values after cloud periods are stagnant or slightly decreasing. This is in contrast to the model runs that use only the GROMHE estimates, where a smaller particle growth is observed, however, with constantly increasing peak values after cloud periods. The new estimates lead to an increase in organic particulate mass of 2 µg m$^{-3}$ at the end of the model run. Other studies have shown previously that different estimation methods of HLCs lead to important differences of several orders

of magnitude (Wang et al., 2017). The current deficits can only be overcome with further development of the estimation methods for HLCs using an updated database of experimentally determined Henry's Law coefficients. For the current study, HLCs of species with an O:C ratio larger or equal to 1 are set to $1 \cdot 10^9$ M atm$^{-1}$. However, the description of uptake processes still inherits one of the largest uncertainties in the protocol.

### 3.3.4 Decay of alkoxy radicals

Further investigation focused at the chemistry of alkoxy radicals in the database. Previous CAPRAM estimates derived from the MOST Project (George et al., 2005) considered two degradation pathways with the following rate constants:

$$R_1R_2O^{\cdot} \longrightarrow R_1^{\cdot} + R_2{=}O \quad (5 \cdot 10^5 \ s^{-1}) \tag{6}$$

$$RO^{\cdot} + O_2 \longrightarrow R{=}O + HO_2^{\cdot} \quad (5 \cdot 10^6 \ M^{-1} \ s^{-1}) \tag{7}$$

With typical modelled aqueous-phase oxygen concentrations of $3.5 - 4.0 \cdot 10^{-4}$ M, this means that almost all alkoxy radicals

decay by C–C bond breaking as the pseudo first order reaction rate constant of the oxygen channel is about a factor of 250 smaller. Therefore, this estimate favours fragmentation over the oxidation to more polar products with an increased O:C ratio whose further oxidation can potentially lead to highly soluble multifunctional compounds. Hence, the degradation reaction has been re-investigated by varying the decay rate of the fragmentation channel over 5 orders of magnitude from $5 \cdot 10^0$ to $5 \cdot 10^5$ s$^{-1}$. Results are shown in Figure 16. As expected, a big increase in particulate matter is observed in the sensitivity run with a

decay rate of $5 \cdot 10^2$ s$^{-1}$, where the ratio of the rate constant of the fragmentation channel to the pseudo first order rate constant of the oxygen channel decreases to 0.25. With the competitiveness of the oxygen channel, particle growth increases by about 3 µg m$^{-3}$ in 4.5 days (see Figure 16). Further reductions of the rate constant of the monomolecular decay channel affect particle growth less leading to the production of one additional µg m$^{-3}$ at the end of the model run. Therefore, $5 \cdot 10^2$ s$^{-1}$ has been used as best estimate for the degradation channel in the final protocol. However, measurements of the decay of alkoxy radicals are

urgently needed for a wider range of compounds and, in particular, for the ratio of the different product channels. The current mechanism is based on just two measurements of aliphatic alkoxy radicals – the methoxy radical (Schuchmann and von Sonntag, 1984) and the ethoxy radical (Bonifačić et al., 2003).

### 3.4 Influence of nitrate radical chemistry

The explicit description of nitrate radical chemistry can lead to an increase in the size of the generated mechanisms, in

particular, when unsaturated organic compounds are involved. Unsaturated compounds preferably react by addition reactions





with the attacking radicals. Therefore, organic nitrates are formed in reactions of $NO_3$ radicals with unsaturated compounds, which were not treated in previous CAPRAM mechanisms. For each new organic nitrate, a complete oxidation scheme has to be generated up to $CO_2$. Only one reaction type has been implemented to remove the nitrate group from the carbon skeleton: the hydrolysis of carbonyl nitrates, which was estimated in accord with peroxyacyl nitrate (PAN) hydrolysis:

$$RC(=O)ONO_2 \xrightarrow{H_2O} RC(=O)OH + NO_3^- + H^+ \quad\quad\quad (8)$$

However, with estimated rate constants of $k_{2nd} = 7.6 \cdot 10^{-6}$ M$^{-1}$ s$^{-1}$ and $E_A/R = 6600$ K (Kames and Schurath, 1995) from PAN hydrolysis, this process is very slow and only important for highly oxidised species, which possess no carbon bound H-atoms and can therefore not be oxidised by radical attack. Therefore, the nitrate function is likely to remain in the molecular structure throughout the whole oxidation process to small molecules. Hence, a large additional reaction scheme is needed. The sensitivity

runs in subsection 3.3 have demonstrated the need to include these reactions in polluted environments. However, in clean environments or for chamber modelling under low-NOx conditions, such detailed chemistry might not be needed. Therefore, GECKO-A has been equipped with 3 options to treat nitrate radical chemistry. For every chemical mechanism, 3 levels of reduction are generated:

- *α version*: Treatment of complete nitrate radical chemistry. Full mechanism generated.
- *β version*: Reactions with unsaturated compounds prohibited to suppress organic nitrate formation. H-atom abstraction reactions are allowed with saturated compounds, but not with unsaturated compounds to prevent a shift in the reaction products.
- *γ version*: No nitrate radical chemistry allowed at all except for the chemistry already treated in previous CAPRAM versions.

Sensitivity studies have been performed under urban and remote conditions, which confirm the importance of nitrate radical chemistry in highly polluted regimes and the necessity for a complete treatment of these compounds as stressed in subsection 3.3. Under remote conditions, organic nitrate formation is negligible with a total organic nitrate mass of less than $5 \cdot 10^{-12}$ g m$^{-3}$. Therefore, it is safe to use the β version of the generated mechanism in these environments. The significance of nitrate radical chemistry is low in general in these environments according to the model simulations. Differences in the

concentration-time profiles between the β and the γ version are only observed for highly oxidised dicarboxylic acids, where electron transfer dominates over H-abstraction reactions. Results for these remote conditions are presented for selected organic compounds in Figure S12 in the ESM. Therefore, for pristine environments or very clean chamber conditions, the γ version of the mechanism might be suitable as well, however, errors increase especially for highly oxidised organic compounds. Moreover, the potential for mechanism reduction is much smaller for the omission of H-abstraction reactions as in general the

same products are produced in these reactions as for the hydroxyl radical attack. Therefore, the generator does not create new reaction schemes for new intermediate compounds and the mechanisms are only reduced for the nitrate radical reactions themselves.

### 3.5 Investigation of further parameters in the protocol

Further sensitivity studies have been employed to investigate the sensitivity of the generated mechanisms towards the mass accommodation coefficient α in the uptake process and the choice of the threshold percentage to omit minor reactions pathways. These results are outlined here only in brief.

*Uptake parameters*

Only very little experimental determinations of mass accommodation coefficients exist in literature (see, e.g., the most recent
IUPAC recommendation by Ammann et al., 2013; or the compilations by Davidovits et al., 2006, 2011). Therefore, it is impossible to derive advanced estimation methods for this parameter. Previous CAPRAM and GECKO-A (Mouchel-Vallon





et al., 2013) estimates used a fixed estimate of 0.1. To evaluate the sensitivity of this parameter, different model runs with varying mass accommodation coefficient estimates from the lowest value in the CAPRAM database of 0.0067 for methyl hydroperoxide to the highest value of 0.5 for HONO.

No effects on any concentration-time profile could be detected. Therefore, this parameter seems insensitive in the typical range found for organic compounds relevant for tropospheric multiphase chemistry and the previous CAPRAM estimate of 0.1 has been taken over in the GECKO-A protocol.

*Cut-off thresholds*

Furthermore, thresholds for the cut-off of minor reaction pathways have been investigated as a means of mechanism reduction. In the aqueous-phase protocol of GECKO-A, a choice of different cut-off parameters is possible. The generator also ensures

that the omission of minor reaction pathways does not result in the loss of too many reaction pathways as described in detail in the next section. As shown in Figure S13 in the ESM, the number of reduced reactions increases linearly with the percentage of the cut-off parameter. Significant differences occur mainly for very crude thresholds of 10% or higher (see Figure S14 in the ESM). In the current protocol a 3% threshold was chosen as standard to compromise between a most detailed description for the use as benchmark mechanism and the reduction of the mechanism size.

**4 The new multiphase mechanism generator GECKO-A**

Based on the evaluation of kinetic data and prediction methods in section 2 and the sensitivity studies in section 3, a protocol for automated aqueous-phase mechanism generation has been defined. The final protocol is described in section 4.1. Its implementation in GECKO-A and the workflow of the new multiphase mechanism generator GECKO-A are explained in section 4.2.

**4.1 The new aqueous-phase protocol for GECKO-A**

This section details the rules defined to predict mechanistic, kinetic, and thermodynamic data for mechanism auto-generation in GECKO-A. An overview of all processes in the new aqueous-phase protocol and the estimates used to derive rate constants and further kinetic and thermodynamic parameters is given in Table 3.

Structure-activity relationships form the basis of the protocol due to the highest accuracy and the ease to implement them in

automated computer tools. Furthermore, a major advantage over all other estimation methods is the ability to predict reaction products. In the current protocol, the SAR by Doussin and Monod (2013) is preferred over the SAR by Minakata et al. (2009). Inspite a similarly successful generation of overall rate constants, attempts have been made to acknowledge branching ratios from measurements by Asmus et al. (1973) in the SAR by Monod and Doussin (2008) (pers. comm. with A. Monod). Therefore, the SAR was chosen for a best possible description of the product distribution. In the protocol, the SAR is

supplemented with parameters from the SAR by Minakata et al. (2009) for unsaturated compounds or H-atom abstraction reactions on carboxyl groups, which is neglected in the SAR by Monod and Doussin (2008) due to the very low partial rate constants. Hence, it is possible to have a mixture of different structure-activity relationships used to predict the rate constant for one compound. In the generator, molecules are treated group-wise for every carbon atom bearing group. Where possible, the SAR by Doussin and Monod (2013) is used and only those groups are supplemented with the SAR by Minakata et al.

(2009) were the SAR by Doussin and Monod (2013) fails (see also Table 3).

Branching ratios for the various product channels are determined by scaling the individual rate constants, which have been modulated by the effects of the neighbouring groups, to the overall rate constant. Immediate oxygen addition to alkyl radicals resulting from H-atom abstraction is assumed in GECKO-A directly producing peroxy radicals. A threshold to omit minor reaction pathways and reduce mechanism size is implemented in GECKO-A where the current protocol uses 3% as standard.

When minor branches are omitted in the generator, the overall rate constant is reduced. No attempts are made to rescale the



remaining reaction pathways to the overall rate constant, hence, reducing the rate constant when minor reaction pathways are omitted. This could result in a sufficient loss of reactivity, if too many minor reaction pathways are omitted, which accumulate to a significant portion of the overall reaction. Therefore, a second parameter is introduced, which ensures that the overall reactivity stays above a defined percentage (80% as standard). In GECKO-A, 10 levels of accuracy exist for the thresholds of

minor reaction pathways ranging from 0.1 to 25%. If the overall reactivity is reduced below the threshold of 80% (or any other user input), then the generator automatically switches to the next more accurate threshold level for minor branches until the reactivity stays above the define threshold or the highest accuracy of 0.1% cut-off for minor branches is reached. This is a major improvement to previous CAPRAM versions, where only a few branching ratios with experimental evidence had been implemented in the mechanism. The approach is similar to that by Mouchel-Vallon et al. (2017) in the CLEPS 1.0 model,

where branching ratios are determined by an SAR and a reduction was applied, where the contribution of each reaction pathway was determined to maintain at least 75% of the total reactivity. After the reduction, branching ratios were recalculated to maintain the global rate constant (Mouchel-Vallon et al., 2017).

Nitrate radical reactions are estimated with the new advanced Evans-Polanyi-type correlations introduced above. Evans-Polanyi correlations are designed for gas-phase H-atom abstraction reactions only, but have proven to work equally well in the

aqueous phase (e.g., Hoffmann et al., 2009). However, this results in limitations of the applicability of Evans-Polanyi-type correlations as they are unsuitable for unsaturated compounds and ions. Therefore, estimates with fixed rate constants of $1 \cdot 10^7$ $M^{-1}$ $s^{-1}$ had to be used for these compounds based on the sensitivity studies described in section 3. Moreover, no branching ratios can be derived from these estimates. Therefore, branching ratios from hydroxyl radical reactions have been taken over except for reactions with carboxylates, where electron transfer is assumed to be the only process. For unsymmetrical

dicarboxylates, equal branching is assumed.

Uptake is described with the structure-activity relationship GROMHE. Due to the large uncertainties, which derive from the experimental data used to obtain the relationship, the SAR is only applied to compounds with an O:C ratio below one. Higher oxidised compounds are assumed to remain in the aqueous phase until a refined structure-activity relationship becomes available based on recent findings. Furthermore, uptake is restricted to species with a Henry's Law constants between $10^2$ and

$10^{12}$ M atm$^{-1}$. This measure has been taken to reduce the size of the generated mechanism as species below the lower threshold will predominantly exist in the gas phase and above the range, species will almost exclusively remain in the aqueous phase (see also discussions by Mouchel-Vallon et al., 2013). Therefore, reaction schemes for these compounds can be omitted in the corresponding phases with only small errors.

Further parameters needed to describe the phase transfer process are the gas-phase diffusion coefficient $D_g$ and the mass

accommodation coefficient α. $D_g$ can be calculated from the molecular diffusion volumes with the FSG method as described by Fuller (1986). For α, a fixed value of 0.1 is applied in the new aqueous-phase protocol based on previous CAPRAM estimates and the sensitivity studies presented in section 3.

For the estimation of hydration equilibrium constants of carbonyl compounds, GROMHE can be used as well. Hydrations are important to determine the effective Henry's Law constants. Therefore, their prediction is already incorporated in GROMHE

and can directly be used for the new GECKO-A protocol. To reduce the size of the generated mechanisms, hydration channels are omitted for equilibria, where the hydration equilibrium constant for a channel is below 5% of the overall equilibrium constant $k^*_{hyd}$ taking all possible hydration forms into account.

Dissociation of carboxylic acids are estimated using the estimation method by Perrin et al. (1981). No threshold for minor dissociation states is currently used in the standard mode of GECKO-A as the threshold is highly dependent on particle and

cloud droplet pH. With varying pH in the model scenarios with non-permanent clouds as applied in this study, this can lead to significant inaccuracies. Furthermore, different dissociation states possess different reactivities. Dissociated carboxylic acids are likely to react by electron transfer reaction (ETR), which is generally faster than H-abstraction reactions (see also Table S1 and S2 in the ESM). Therefore, with increased reactivities of the products of minor branches, significant turnovers can still





be achieved through these reaction pathways. These considerations apply also to hydration processes, but are less critical for this equilibrium type with less variation of the reaction rate constants between the different hydration forms. Future versions of the generator should use more advanced determinations of the threshold for minor reaction pathways, which include the consideration of the reactivities of the products in each channel. However, these estimates would depend on typical radical

oxidant concentrations and pH and, hence, would be environment specific.

For highly oxidised polycarbonyl compounds, a monomolecular decay has been implemented in addition to the radical attack, if three carbonyl groups are found within four adjacent carbon atoms. Bond breaking occurs always between two carbonyl groups. If three carbonyl groups are directly adjacent to each other, equal branching is assumed leading to a maximum of four fragments. The decay rate is estimated with 0.1 s$^{-1}$ based on the sensitivity studies in section 3.

For organic nitrates with an α-carbonyl group, hydrolysis is assumed in addition to radical attack. The second order rate constant is assumed equal to PAN hydrolysis with an estimated second order rate constant of $k_{2nd} = 7.6 \cdot 10^{-6}$ M$^{-1}$ s$^{-1}$ and $E_A/R =$ 6600 K (Kames and Schurath, 1995). Products formed are a carboxylic acid and dissociated nitric acid.

Due to the experimental difficulties to determine the very fast reaction rate constants of organic radical compounds, measurements are scarce. Therefore, no sophisticated estimation method could be derived for these compounds and estimates

use fixed rate constants and branching ratios based on previous CAPRAM estimates. Organic peroxy radical reactions are based on the mechanism proposed by von Sonntag (1987) and von Sonntag and Schuchmann (1991) with updated mechanistic and kinetic data by Schaefer et al. (2012) and include the reactions and estimated rate constants as given in Table 3. Peroxy radical reactions are included as pseudo first order reactions based on the methodology of the gas-phase mechanism MCM (Jenkin et al., 1997; Saunders et al., 2003), where a given peroxy radical reacts with the sum of concentrations of all peroxy

radicals to account for recombinations and cross-reactions. Several exceptional reaction pathways of certain peroxy radicals have been included in the protocol. Peroxy radicals bearing a hydroxyl group in α-position solely decay by HO$_2$ elimination to form a carbonyl compound and a hydroperoxyl radical in a unimolecular decay. Decay rates are estimated with 200 s$^{-1}$ for one hydroxyl group in α-position and with 1000 s$^{-1}$ for a gem-dial function in α-position based on previous CAPRAM estimates (Tilgner and Herrmann, 2010). Recombination and cross-reactions are neglected for this type of peroxy radicals due to the

considerations given in section S3.5 of the ESM. Peroxy radicals with an adjacent carboxyl or carboxylate group decay by CO$_2$ elimination in addition to the recombination- and cross-reactions (see Table 3 for more details on mechanistic and kinetic data). Decay rates are estimated based on the measurements of the acetate peroxy radical (CH$_2$(OO·)C(=O)O$^-$) by Schuchmann et al. (Schuchmann et al., 1985).

Alkoxy radical decay by reaction with O$_2$ or by monomolecular decay have been implemented in the new protocol with the

rate constants determined in subsection 3.3. Immediate oxygen addition is assumed for any carbon centred radical fragments, which will directly lead to the formation of peroxy radicals. For acyloxy radicals, a monomolecular decay by C–C bond breaking of the acyloxy group is assumed leading to dissolved carbon dioxide and a peroxy radical as immediate oxygen addition is assumed for the resulting alkyl radical. In contrast to previous CAPRAM estimates, a reduced decay rate of $5 \cdot 10^2$ s$^{-1}$ is assumed in the current protocol in accord with the reduced bond braking rates of alkoxy radicals.

For every auto-generated CAPRAM version, three subversions are created. The standard α version describes the full nitrate radical chemistry. The β mechanism includes the full inorganic and organic nitrate radical chemistry of the CAPRAM core mechanism 3.0n, but omits any reactions of nitrate radicals with unsaturated organic compounds. This measure significantly reduces the size of the mechanism as organic nitrate formation is suppressed resulting in a significantly reduced number of intermediate species in the oxidation chain of organic compounds. In the most reduced γ mechanism, only nitrate radical

chemistry of the core mechanism CAPRAM 3.0n is allowed, while nitrate radical chemistry in the auto-generated mechanism part is suppressed. Reduction potential is far less as for the first reduction step as generally the same products are produced in reactions of nitrate radicals with saturated organic compounds as in the corresponding hydroxyl radical reactions. Thus, the mechanism is only reduced by the number of nitrate radical reactions, but no reduction in the number of species is achieved.



The latter two subversions of the mechanism are meant for modeling in clean environments or for low-NOx chamber simulations.

Two modes of operation exist in GECKO-A. The generator can either be used to produce "stand-alone mechanisms", where the generator creates complete oxidation mechanisms up to the final product $CO/CO_2$ solely based on the rules of the protocol

described above. As the other option, the generator may be used to produce supplementary modules to CAPRAM 3.0n or higher versions. This mode uses the CAPRAM mechanism as core and new oxidation schemes are only created by GECKO-A for those species not yet described in CAPRAM. Thus, the generator will stop for any species already present in CAPRAM and acknowledge the chemistry of CAPRAM rather than producing own oxidation schemes.

### 4.2 Implementation of the protocol in GECKO-A

The above protocol rules have been implemented as new module to the existing gas-phase mechanism generator GECKO-A for mechanism auto-generation. The gas-phase mechanism generator is described in detail by Aumont et al. (2005). This section details the incorporation of the new aqueous-phase protocol. The workflow of the aqueous-phase module of GECKO-A is schematically shown in Figure 17. GECKO-A will always be initialised in the gas phase as most organic compounds are emitted as trace gases. However, it is possible to extract just the aqueous-phase mechanism from GECKO-A and couple it to

further mechanisms as it was done in this study.

To generate oxidation schemes with GECKO-A, any number of 'primary gas-phase species', for which an oxidation mechanism is desired, can be added to the 'primary stack'. The mechanism generator will start with the first species in this stack and generate all possible gas-phase reactions as described by Aumont et al. (2005). In a second step, Henry's Law constants (HLCs) will be generated by the GROMHE subroutine and phase transfer will be described in the generated

mechanisms if the HLC is above a given threshold. Otherwise, the assumption is made that the compound is insoluble and exclusively available in the gas phase. Therefore, uptake is suppressed to reduce the aqueous-phase mechanism size.

In a next step, the protocol rules described in subsection 4.1 will be applied (see also Figure 17). This includes checks for radical functions and any other functional groups in the molecule and the application of the appropriate rules from the protocol as schematically depicted in Figure 17. The generator determines any mechanistic, kinetic, and thermodynamic data for the

given compound. It furthermore checks whether reaction products have previously been treated. New intermediate species will be added to an 'aqueous stack' for later treatment. Before the data is written into a mechanism, the solubility of the compound is checked again. If the HLC is below a pre-defined threshold, release to the gas phase is treated. The compound will be checked for its treatment in the gas phase. Untreated species will be added to a *gaseous stack* for later treatment in the gas-phase mechanism. If a compound exceeds the HLC limit, exclusive existence in the aqueous phase is assumed to reduce the

size of the gas-phase mechanism.

In a last check, GECKO-A consults an experimental database. Any available experimental kinetic or thermodynamic data is preferred over theoretically predicted data. All reactions for the given compound are written into a mechanism as determined by GECKO-A. The generator then returns to the 'aqueous stack' for untreated species and generates reactions for these compounds as well. Thus, the *aqueous stack* will be filled by new intermediate species from the generated reactions. After the

treatment of a species it will be marked to avoid double treatment of species generated in several reactions. GECKO-A proceeds with all unmarked species in the aqueous stack until every species in the stack is marked and the products of a current reaction are only species already treated previously or the end product $CO/CO_2$. In this case, the generator returns to the gas-phase stack and continues to generate gas-phase reactions for any unmarked species in this stack. The gas and aqueous-phase mechanism will be generated alternately in this way until a complete multiphase mechanism is generated for the species of the

'primary stack' and all intermediate products.

After the generation of a complete multiphase oxidation scheme for a species of the *primary stack*, the generator returns to this stack to treat the next species in it. The reaction mechanism for this compound will complement the mechanism of the previous




species of this stack. Therefore, species of the *gaseous* and *aqueous stack* will not be emptied until the complete treatment of the 'primary stack'. Instead, they will be used as markers to indicate previous treatment in the generator and avoid double reactions. GECKO-A finishes, when all species of the 'primary', 'gas', and 'aqueous stack' are marked as treated and a complete multiphase reaction scheme has been generated for these species.

## 5   5 Validation of the CAPRAM/GECKO-A protocol against aerosol chamber experiments

### 5.1 Chamber experiments

The oxidation of 1,3,5-trimethyl benzene (TMB, mesitylene) was conducted in the aerosol chamber LEAK (Leipziger Aerosolkammer). A detailed description of LEAK can be found elsewhere (Mutzel et al., 2016 and references therein). The experiments were conducted in the presence of sodium sulfate seed particles mixed with 30% sodium peroxide. The photolysis

of ozone with UV-C light ($\lambda = 254$ nm) served as gas-phase OH radical source. Ozone was generated by UV irradiation of $O_2$ using a flow rate of 3 L minute$^{-1}$. Furthermore, sodium peroxide was used as OH radical in-situ source in the particle phase via the reaction sequence:

$$Na_2O_2 + 2H_2O \longrightarrow 2\,NaOH + H_2O_2 \tag{9}$$

$$H_2O_2 + h\nu \longrightarrow 2\;^{\cdot}OH \tag{10}$$

1,3,5-TMB ($\approx 85.5$ ppb) was injected into LEAK with a microliter-syringe. The oxidation of 1,3,5-TMB was conducted at a relative humidity of 75% and room temperature. The consumption of the precursor compound ($\Delta$HC) was monitored with a proton-transfer-reaction mass spectrometer (PTR-MS) over a reaction time of 210 minutes. The volume size distribution of the seed particles was measured with a scanning mobility particle sizer (SMPS). An average density of 1 g cm$^{-3}$ was used to calculate the produced organic particle mass ($\Delta$M). The particle phase was sampled after the experiments. 1.8 m³ and 0.6 m³

of the chamber volume was collected on a PTFE filter (borosilicate glass fiber filter coated with fluorocarbon, 47 mm in diameter, PALLFLEX T60A20, PALL, NY, USA), which was connected to a denuder (URG-2000-30B5, URG Corporation, Chapel Hill, NC, USA; Kahnt et al., 2011) to avoid gas-phase artefacts. The particle-phase products were also collected with a condensation growth and impaction system (C-GIS, Sierau et al., 2003) with a time resolution of 15 min. Particle mass was monitored by an aerosol mass spectrometer (High-Resolution Time-of-Flight Aerosol Mass Spectrometer; HR-ToF-AMS).

### 5.2 Model setup

For the gas-phase description of the chemistry, the MCMv3.2 was used (http://mcm.leeds.ac.uk/MCMv3.2/; Jenkin et al., 2003; Saunders et al., 2003). GECKO-A was initialised only with methylglyoxal as both, the chamber and model results (see results in section 5.3), did not show significant concentrations of any other intermediate compound in the oxidation chain between 1,3,5-TMB and methylglyoxal. The final version of GECKO-A as described above was used to generate the aqueous-

phase oxidation scheme including phase transfers (with Henry's Law coefficients set to $1 \cdot 10^9$ M$^{-1}$ s$^{-1}$ for species with an O:C greater or equal to 1). The generated scheme includes important aqueous-phase compounds such as oxalate, pyruvate, acetate, formate, formaldehyde or several di- and polycarbonyls. The total aqueous-phase scheme comprises 264 species, 596 reactions and 25 phase transfers. The mechanism is supplied as SPACCIM input file in the ESM.

Despite very low gas-phase concentrations, chemical turnovers can be substantial for some of the larger TMB oxidation

products. Furthermore, some of the larger intermediates are already very oxidised with high O:C ratios and the aqueous-phase might be a potent sink for them. Therefore, several sensitivity studies have been conducted, where uptake into the aqueous phase was considered for 20 additional intermediates. As some of the intermediates were still very large with carbon numbers up to 9, an explicit description model with the currently very detailed protocol is beyond the capabilities of the current box with its detailed microphysical scheme. Therefore, the generic model species WSOC was used as reaction product for those

intermediates in the sensitivity run UPT.



As the majority of those intermediates bear double bonds, carbonyl functions, and/or carboxyl functions, reactive uptake is possible due to the fast kinetics (addition reactions, hydrolysis, and dissociations) of those compound classes. Therefore, another sensitivity run ("RXN") was performed, where the initial aqueous-phase reactions were described explicitly following the generator protocol rules. Reaction products were described only, if they were part of the explicit methylglyoxal oxidation

scheme, otherwise WSOC was used as reaction product. In a last sensitivity run, WSOC yields in the additional reaction scheme of the "RXN" run were reduced to 0.4 to match the model predictions of the particle mass with the AMS and SMPS measurements. The additional species and reactions of the sensitivity runs UPT, RXN, and RXN$_{0.4}$ alongside with the kinetic and thermodynamic data used can be found in Table S15 in the ESM.

Model runs were performed with the box model SPACCIM (Wolke et al., 2005) with a detailed description of microphysical

and chemical processes. The model was initialised with $2.14 \cdot 10^{12}$ molecules cm$^{-3}$ of 1,3,5-TMB, $3.11 \cdot 10^{10}$ molecules cm$^{-3}$ of Ozone and traces of NO and NO$_2$ to account for any background chemistry (1.5 and $2.0 \cdot 10^{10}$ molecules cm$^{-3}$, respectively). The aqueous-phase was initialised with sodium sulfate particles and 30% sodium peroxide. OH concentrations were held constant in both phases at a gas-phase concentration level of $5 \cdot 10^6$ molecules cm$^{-3}$ and an aqueous concentration of $10^{-11}$M. This was necessary as the current model version is designed specifically for atmospheric applications with a solar radiation

spectrum under tropospheric conditions. Thus, it was not possible to simulate a constant light source from non-solar irradiation. The same microphysical conditions as in the chamber experiments were used with constant relative humidity of 75% and a constant temperature of 298K.

### 5.3 Comparison of chamber and model results

Figure 18 shows the gas-phase concentration time profiles of mesitylene and its major oxidation products methylglyoxal and

acetic acid as monitored by PTR-MS (squares) or simulated in the model run RXN$_{0.4}$. It should be noted that concentrations for methylglyoxal and acetic acid had to be scaled by a factor of 1.35 and 0.14, respectively, which can at least in part be explained by different protonation efficiencies for different molecules in the PTR-MS. This leads to inaccuracies in the PTR-MS measurements, which justify a re-scaling. Another, yet more unlikely, source of error would be another overlying compound with the same mass-to-charge ratio, which modifies the PTR-MS measurements for the respective *m/z*.

With the rescaled concentrations, however, the MCMv3.2 is able to explain the measured concentrations well. The general degradation of mesitylene and the subsequent formation of methylglyoxal and acetic acid are well captured by the model. The concentration-time profile of mesitylene shows a slightly steeper curve in the model results with slightly stronger degradation rates in the first 1.5 hours and slower degradation rates thereafter. Up to 15 ppb less mesitylene are observed in the model compared to the experiment in the first hour. Consequently, methylglyoxal production is slightly over-predicted in the first 1.5

hours leading to ~2 – 3 ppb more methylglyoxal in the model compared to the measurements and the magnitude of the decay of methylglyoxal towards the end of the experiment is smaller in the model than in the measured concentration-time profile. At the end of the experiment, the model overpredicts the experimental results by about 3 ppb. Agreement of acetic acid concentrations between model and experiment is very good with slightly lower modelled concentrations towards the end of the experiment (2 – 3 ppb).

Possible explanations for the discrepancies between measured and modelled concentrations in Figure 18 are uncertainties in the mechanistic and kinetic data either directly in the gas phase or indirectly in the aqueous phase influencing gas-phase concentrations through reactive uptake. Furthermore, uptake processes of intermediates in the aqueous-phase oxidation chain are a likely explanation for the discrepancies as the sensitivity studies have shown large uncertainties for these processes (see section 3).

In the aqueous phase, methylglyoxal chemistry alone fails to predict an increase in particulate matter (PM) by approximately 50 µg m$^{-3}$ as seen by the SMPS and AMS (see Figure 19). Particle growth is negligible in the model run TMB. Therefore, more sensitivity runs have been performed as introduced in section 5.2, where uptake of further precursors of methylglyoxal





and their initial reaction steps in the aqueous phase are considered. The influence of uptake alone in the sensitivity run UPT is negligible and the concentration-time profile of the overall dry particle mass is indistinguishable from the base run TMB. Aqueous-phase chemistry has a big influence on particle growth. In the simulation RXN, where besides uptake, initial reactions are also considered assuming 100% water-soluble organic carbon (WSOC) as reaction product, model predictions of PM at

the end of the experiment are with ~130 µg m$^{-3}$ about twice as high as seen by the SMPS or AMS. Reasonable agreement between modelled and measured concentrations is reached only when the reaction product WSOC is reduced by an artificial stoichiometric index of 0.4. This rescaling has also an impact on gas-phase concentrations decreasing the concentrations of methylglyoxal and, consequently, acetic acid. To match the concentrations in the best scenario RXN$_{0.4}$ to experimental results, scaling factors for methylglyoxal in the gas phase were improved from 2.5 to 1.35 while acetic acid decreased from 0.23 to

0.14. The comparison of all gas-phase concentrations of methylglyoxal and acetic acid can be found in Figure S18 in the ESM. Hence, large oxidation products are needed to explain the particle growth seen in the experiment, despite their negligible concentrations in the gas phase. The scaling factor for WSOC of 0.4 in the scenario RXN$_{0.4}$ could be interpreted in a way that subsequent aqueous-phase chemistry produces a mixture of soluble, semi-volatile, and volatile products of which 40% will remain in the aqueous phase and 60% are released back to the gas phase. This interpretation might also be used as a justification

for the newly introduced degradation rates for alkoxy radicals. While previous CAPRAM mechanisms used fragmentation as the predominant degradation pathway, products are shifted towards carbonyl formation with a ratio of 73% carbonyl formation and 27% fragmentation under the conditions of the experiment. As carbonyl formation decreases volatility in contrast to fragmentation, the new product distribution is in better agreement with the above results from the sensitivity runs RXN and RXN$_{0.4}$ assuming that not all carbonyl compounds will remain completely in the aqueous phase.

However, even in the final re-scaled sensitivity run RXN$_{0.4}$, some discrepancies can be observed. The increase in PM is delayed by about 20 minutes in the model run. Furthermore, the model does not predict a slight decrease in PM towards the end of the model run as seen by the SMPS and AMS. The delayed increase is somewhat surprising and its reason remains unclear as an increased degradation of mesitylene was observed in the first 1.5 hours of the experiment in Figure 18, which would imply an increased particle formation in the model runs compared to the measurements. However, even in the run RXN, where 100%

WSOC formation is assumed, PM formation is slightly lower in the first 40 minutes of the experiment than observed by the AMS and SMPS. It should also be mentioned that no wall losses of particles and oxidised organic compounds were implemented in the model as SPACCIM mainly targets atmospheric applications. Missing consideration of chamber background chemistry is another source for discrepancies. Moreover, currently no non-radical chemistry, such as accretion reactions are implemented in the CAPRAM/GECKO-A protocol, albeit, no oligomers were observed in the experimental

particle analysis.

Further investigations of the microphysical parameters liquid water content (LWC) and mean particle radius in Figure 20 show more interesting aspects of the influence of aqueous chemistry on particle growth. It seems that particle chemistry is necessary to initiate particle growth. Without aqueous-phase chemistry of the larger intermediates, no particle growth is observed in the run TMB. Uptake of these compounds alone is not sufficient to overcome the curvature effect of the relatively small seed

particles with a mean radius of 55 nm. However, if chemistry is treated in the runs RXN and RXN$_{0.4}$, particle radii and, hence, LWC increase rapidly. The chemistry is needed as a trigger for the initial growth. With growing radii and decreasing curvature effect, LWC increases and uptake processes have an increased importance for the particle composition.

The increased uptake of organic compounds and, hence, increased carboxylic acid formation in the scenarios RXN and RXN$_{0.4}$ causes a decrease in pH by about 1 unit. The decrease from near neutral concentration is strongest in the first minutes of the

experiments and levels of after 1 to 2 hours as can be seen in Figure S16 of the ESM.

Attempts were made to identify concentrations of the expected aqueous-phase products by offline analysis of C-GIS and PTFE filter samples.  However, concentrations were close to the detection limit or within the noise of the expected background chemistry as can be seen from Table 4. In this table, the first sample represents background concentrations prior to the injection



of 1,3,5-TMB and ozone. Moreover, large differences are seen in the concentrations for some of the compounds determined from samples using the different sampling techniques. For example, oxalate concentrations from filter samples were with 6.8 µg m$^{-3}$ almost 10 times as high as determined from the C-GIS samples (0.7 µg m$^{-3}$). Therefore, a direct comparison of concentrations of individual species of experimental and modelled results is difficult.

It should be mentioned that large amounts of malic acid were detected in the gas and the aqueous phase throughout the experiment as shown for gas phase in Figure S17 in the ESM and for the aqueous phase in Table 4. However, there is currently no information about formation pathways of malic acid from mesitylene oxidation and concentrations are attributed to background chemistry. Consequently, the MCMv3.2 does not predict the formation of malic acid in any of the sensitivity runs. For a thorough analysis, modelled chemical fluxes from the explicit scheme in scenario RXN$_{0.4}$ have been analysed and

averages over the whole 3.5 hours of the experiments are shown in Figure 21. In addition, final aqueous-phase concentrations after 3.5 hours are shown for each species. Figure 21 shows that less than 3% of methylglyoxal is oxidised to acetic acid in the aqueous phase and gas-phase oxidation is the main sink. In the aqueous phase, 75% of the acetic acid is formed directly from methyl glyoxal while 25% is formed with pyruvic acid as intermediate, which is the major first generation aqueous-phase oxidation product. Oxidation of pyruvate leads mainly to oxo-pyruvic acid, which is subsequently degraded to mesoxalic acid

and oxalate in the model and finally to formate. The formation of these highly oxidised dicarboxylic acids is a direct consequence of the revision of the radical protocol in the CAPRAM/GECKO-A mechanism generation process. The revised monomolecular decay of alkoxy radicals after the hydroxyl radical attack of pyruvic acid leads to an increased formation of oxo-pyruvic acid in favour of fragmentation. Previous CAPRAM mechanisms would have favoured the formation of more acetic acid. Thus, the new protocol favours particle growth, as mesoxalic and oxalic acid are far less volatile than acetic acid.

Modelled concentrations in Figure 21 (in ng m$^{-3}$) are lower than seen in the filter and C-GIS samples (see Table 4), but are generally within the experimental error. An exception is methylglyoxal with concentration modelled somewhat higher (78 ng m$^{-3}$) compared to measurements (48 ng m$^{-3}$).

Despite the difficulties to conduct experiments with sufficient bulk aqueous-phase chemistry, the plausibility of the new CAPRAM/GECKO-A protocol could be proven while further validation from model simulations with larger mechanisms

under real tropospheric conditions are needed. These studies will be presented in a companion paper.

## 6 Conclusions

A new protocol for mechanism auto-generation within the expert system GECKO-A has been developed a tested. The protocol was developed to supplement the mechanism generation process of the aqueous-phase benchmark mechanism CAPRAM. Following an extensive evaluation of existing kinetic data and prediction methods for aqueous-phase mechanistic,

thermodynamic and kinetic data, Evans-Polanyi-type correlations have been further improved. The advanced correlations consider all bonds in a molecule connected to an H-atom rather than the weakest bonds only. This way, minor channels are considered in the correlation, which decreases biases especially for compounds, where measurements are typically scarce. Interesting effects were observed, especially for hydroxyl radical reactions. While previously, BDE and second order rate constants were anti-correlated due to an increased reactivity of weaker bonds, the new correlation shows a positive relationship

due to increased side attacks on larger molecules, which have a larger accumulated bond dissociation enthalpy ($\sum$BDE). Moreover, for hydroxyl radicals, quadratic relationships were seen, while nitrate radical reactions can still be described with linear relationships. The most likely explanation for this behaviour is that for larger organic compounds the diffusion limit of the reaction with OH is reached and the levelling of the rate constants can best be described with quadratic relationships while nitrate radical reactions are well below a diffusion-controlled range, where linear relationships are still applicable.

The evaluation process led to a new CAPRAM/GECKO-A protocol for mechanism auto-generation, where hydroxyl radical reactions are estimated with structure-activity relationships by Doussin and Monod (2013) supplemented by SARs from



Minakata et al. (2009). The advanced Evans-Polanyi-type correlations were used to describe nitrate radical reactions with organic compounds and estimates with fixed rate constants and branching ratios were used to predict reactions of organic radicals. Hydrations are estimated with the SAR GROMHE and dissociations using Taft parameters with the method by Perrin et al. (1981). Uptake is described with the Schwartz approach (Schwartz, 1986) using GROMHE for the estimation of Henry's

Law coefficients, the FSG method (Fuller, 1986) for the estimation of gas-phase diffusion coefficients and constant mass accommodation coefficients $\alpha = 0.1$.

Rigorous testing in sensitivity studies led to further improvements of the protocol, such as a monomolecular decay channel of polycarbonyl compounds with at least 3 carbonyl groups within 4 carbon-containing groups. Furthermore, the largest uncertainties in current estimation methods were identified and let to further improvements of the protocol. These were found

for (i) uptake processes and (ii) radical reactions. While uncertainties in uptake processes are generally high, recent research seems to additionally indicate that GROMHE is underestimating uptake. Therefore, the current protocol refrains from using GROMHE for very polar molecules with an O:C ratio of one or greater. Instead, a Henry's Law coefficient of $1 \cdot 10^9$ M atm$^{-1}$ is used. Moreover, radical reactions are based on only a very few measurements for peroxy radicals and recent estimates from previous CAPRAM studies for alkoxy radicals. The latter radicals are either fragmenting or reacting with dissolved oxygen to

form carbonyl compounds. The ratio has a large impact on particle growth as carbonyl formation leads to less volatile products than fragmentation. Therefore, the ratio was refined to best resemble experimental findings. Further sensitivity studies focused on organic nitrate formation in the aqueous phase as alkyl nitrate concentrations are at the upper end of observations or overestimated in the current sensitivity studies.

A mesitylene oxidation experiment was conducted at the aerosol chamber LEAK, which could prove the plausibility of the

current protocol for automated mechanism construction. Modelled gas-phase concentrations of mesitylene, methylglyoxal, and acetic acid are in good agreement with the PTR-MS measurements. Sensitivity studies have shown, that the uptake and aqueous-phase chemistry of larger methylglyoxal precursor species are needed to accurately describe particle growth. However, the currently very explicit protocol would lead to very large and detailed mechanisms for the considered compounds with up to 9 carbon atoms and tests are infeasible for the current version of the box model SPACCIM.

Therefore, more model studies with more comprehensive mechanisms under real atmospheric conditions are needed for the validation of the protocol. The studies are part of a companion paper with further CAPRAM mechanism development.

**Author contributions.**

P.B., C.M.V., A.T, B.A. and H.H. developed the protocol for mechanism auto-generation ]

P.B., C.M.V., A.T., R.W., B.A. and H.H. designed the model experiments.

P.B., C.M.V., and B.A. did model development on GECKO-A.

P.B. and R.W. did model development on SPACCIM.

P.B., A.T., A.M., O.B., M.R., L.P., D.v.P., and H.H. designed the aerosol chamber experiments.

M.R., A.M., O.B., L.P., and D.v.P. performed the aerosol chamber experiments and analysed the data.

PB, A.T., and H.H analysed model results.

P.B., A.T., and H.H. wrote the paper.

**Competing interests.** The authors declare that they have no competing interests.

**Acknowledgements.** This work was performed within the project ATMOCHEM funded by the German Research Foundation

(Deutsche Forschungsgemeinschaft, DFG) under the project number BO 1714/3-1. Co-funding was received from the European Commission within EUROCHAMP-2020 under the EUROCHAMP Grant Agreement No. 730997.



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





**Tables**

**Table 1 Overview of the different sensitivity runs performed to derive a suitable estimation for the monomolecular decay of poly-carbonyls.**

| Sensitivity run | Decay rate[s$^{-1}$] | Additional oxidation by OH |
|---|:---:|:---:|
| woBB | 0 (no bond breaking) | ☑ |
| BBe-2+OH | $10^{-2}$ | ☑ |
| BBe-2woOH | $10^{-2}$ | ☒ |
| BBe-1+OH | $10^{-1}$ | ☑ |
| BBe-1woOH | $10^{-1}$ | ☒ |
| BBe0+OH | $10^{0}$ | ☑ |
| BBe0woOH | $10^{0}$ | ☒ |



**Table 2 Parameterisations for the aqueous-phase chemistry of WSOC and HULIS compounds.**

| Process | $k_{298}/K/j_{max}$ [a] | Reference/comment |
|---|---|---|
| $WSOC + OH \rightarrow WSOC + HO_2$ | $3.8 \cdot 10^8$ | Estimated after Arakaki et al. (2013) |
| $HULIS + OH \rightarrow HULIS + HO_2$ | $3.8 \cdot 10^8$ | Estimated after Arakaki et al. (2013) |
| $HULIS^- + OH \rightarrow HULIS^- + HO_2$ | $3.8 \cdot 10^8$ | Estimated after Arakaki et al. (2013) |
| $WSOC + NO_3 \rightarrow WSOC + HO_2 + NO_3^- + H^+$ | $1.0 \cdot 10^7$ | Estimated in accord with subsection 3.3.1 |
| $HULIS + NO_3 \rightarrow HULIS + HO_2 + NO_3^- + H^+$ | $1.0 \cdot 10^7$ | Estimated in accord with subsection 3.3.1 |
| $HULIS^- + NO_3 \rightarrow HULIS + HO_2 + NO_3^-$ | $5.0 \cdot 10^7$ | Estimated in accord with subsection 3.3.1 |
| $HULIS \rightleftharpoons HULIS^- + H^+$ | $3.98 \cdot 10^{-4}$ [b] | Estimated with $pK_a$ of fulvic acid after Salma and Láng (2008) |
| $Fe^{3+} + HULIS^- \rightleftharpoons [FeHULIS]^{2+} + 0.5\,HO_2$ | $1.0 \cdot 10^9$ [c] | Estimated after iron-oxalato complex formation |
| $Fe^{2+} + HULIS^- \rightleftharpoons [FeHULIS]^+$ | $1.0 \cdot 10^9$ [c] | Estimated after iron-oxalato complex formation |
| $[FeHULIS]^{2+} \xrightarrow{h\nu} Fe^{2+} + HULIS + 0.5\,HO_2$ | $2.0 \cdot 10^{-2}$ | Estimated after iron-hydroxy complex photolysis (see Arakaki et al., 2010) |

[a] $k_{298}$ is given for irreversible reactions (indicated by $\rightarrow$) in $M^{-1}\,s^{-1}$, $K$ is given for equilibria (indicated by $\rightleftharpoons$) in $M^{-1}$. For photolysis (indicated by $\xrightarrow{h\nu}$), $j_{max}$ is given in $s^{-1}$. [b]$k_{backward} = 5.0 \cdot 10^9\,s^{-1}$; [c]$k_{backward} = 3.0 \cdot 10^{-3}\,s^{-1}$





**Table 3 Processes implemented in the final GECKO-A aqueous-phase protocol for automated mechanism self-generation.**

| Process | Estimation/comment |
|---|---|
| Organic compounds + OH | SAR by Doussin and Monod (2013) accompanied by SAR by Minakata et al. (2009) for carboxyl/carboxylate groups and unsaturated carbon atoms with a double bond or groups in α- and β-position to unsaturated carbon atoms. For organic nitrate functions the same group contribution factors for α- and β-effects are assumed as for OH-functions. Immediate oxygen addition is assumed to formed alkyl radicals. 3% cut-off for minor branches. |
| Neutral saturated organic compounds + $NO_3$ | Advanced Evans-Polanyi-type correlations using product determinations from the SAR of the corresponding hydroxyl radical reaction. |
| Unsaturated organic compounds + $NO_3$ | $k_{2nd} = 1 \cdot 10^7$ $M^{-1}$ $s^{-1}$ based on subsection 3.3.2. |
| Dissociated carboxylic acids and diacids + $NO_3$ | $k_{2nd}$(monocarboxylates + $NO_3$) = $2 \cdot 10^7$ $M^{-1}$ $s^{-1}$, $k_{2nd}$(DCA mono-anion + $NO_3$) = $2.5 \cdot 10^7$ $M^{-1}$ $s^{-1}$, and $k_{2nd}$(DCA dianion + $NO_3$) = $7.75 \cdot 10^7$ $M^{-1}$ $s^{-1}$ with ETR assumed as only reaction pathway and equal branching for unsymmetrical dicarboxylates. Nitrate radical addition to double bonds is assumed for unsaturated organic acids as only reaction pathway instead of ETR with $k_{2nd}$ = $1 \cdot 10^7$ $M^{-1}$ $s^{-1}$. |
| Monomolecular decay of polycarbonyl compound | Monomolecular decay of polycarbonyls with 3 carbonyl groups within 4 adjacent carbon atoms with $k_{1st}$ = 0.1 $s^{-1}$. Fragmentation is always between 2 carbonyl groups. Equal branching is assumed for unsymmetrical polycarbonyls with 3 adjacent carbonyl groups. |
| Hydrolysis of carbonyl nitrates | Hydrolysis of nitrates with an adjacent carbonyl group is assumed with $k_{2nd}$ = $7.5 \cdot 10^{-6}$ $M^{-1}$ $s^{-1}$ and $E_A/R$ = 6600 K based on PAN hydrolysis by Kames and Schurath (1995). |
| $RO_2 \xrightarrow{R'O_2} 0.4\ RO + 1.1\ R{=}O + 0.5\ ROH + 0.7\ O_2 + 0.3\ H_2O_2$ | Peroxy radical ($RO_2$) recombinations based on mechanism by von Sonntag and Schuchmann (1991) and von Sonntag et al. (1997) leading to alkoxy radicals (RO), carbonyl (R=O) and alcohol (ROH) compounds with updated kinetic and mechanistic data for the acetonyl peroxy radical by Schaefer et al. (2012) using the given fixed branching ratios and $k_{2nd}$ = $7.3 \cdot 10^8$ $M^{-1}$ $s^{-1}$. |
| Peroxy radicals with OH in α-position | $HO_2$ elimination assumed as only process with $k_{1st}$ = 200 $s^{-1}$. For a gem-diol function in α-position, an increased $k_{1st}$ of 1000 $s^{-1}$ is assumed. |
| $2\ RCH(O_2\cdot)COOH \rightarrow 2\ RCHO + 2\ CO_2 + H_2O_2$ <br><br> $2\ RCH(O_2\cdot)COO^- \xrightarrow{2H_2O} \rightarrow 2\ RCHO + 2\ CO_2 + H_2O_2 + 2\ OH^-$ | Additional reaction pathway to recombination and cross-reactions for peroxy radicals with $COOH/COO^-$ in α-position with estimated $k_{2nd}$ = 200 $s^{-1}$ leading to a carbonyl compound, $H_2O_2$, and dissolved $CO_2$. |





| | |
|---|---|
| $RCO_3 \xrightarrow{H_2O} RCOOH + HO_2$ | Immediate hydration of acylperoxy radicals ($RCO_3$) and $HO_2$ elimination with $k_{1st} = 1000$ s$^{-1}$ is assumed leading to carboxylic acid formation (RCOOH). |
| Monomolecular decay of alkoxy radicals | C–C bond braking with equal branching for unsymmetrical molecules is assumed for alkoxy radicals with and estimated $k_{1st}$ of $5 \cdot 10^2$ s$^{-1}$. |
| Alkoxy radicals + $O_2$ | Formation of $HO_2$ and carbonyls with an estimated $k_{2nd}$ of $5 \cdot 10^6$ s$^{-1}$. |
| Decomposition of acyloxy radicals | Decarboxylation by C–C bond braking of acyloxy radicals leading to dissolved $CO_2$ and a peroxy radical (as immediate oxygen addition is assumed for the alkyl radical formed). |
| Hydration of carbonyl compounds | Hydration constants calculated with GROMHE (Raventos-Duran et al., 2010). Estimated $k_b$ of $5.69 \cdot 10^{-3}$ s$^{-1}$; $k_f$ calculated from $K$ and $k_b$. 5% cut-off for minor hydration forms. |
| Dissociations of carboxylic acids | Calculated using $pK_a$ prediction method with Taft parameters by Perrin et al. (1981) and an estimated backward rate constant of $5 \cdot 10^{10}$ M$^{-1}$ s$^{-1}$; $k_f$ calculated from $pK_a$ and $k_b$. Currently no cut-off for minor dissociation states. |
| Phase transfer | HLC calculated using GROMHE; $D_g$ calculated with FSG method (Fuller, 1986); $\alpha$ assumed 0.1. |



**Table 4 Measured uncorrected concentrations in µg m⁻³ from C-GIS and the PTFE filter samples for identified products. Times given in the first column refer to measurement points in hours and minutes after the injection of ozone and the start of the UV-C photolysis.**

| Sample | [TIME] | Methyl-glyoxal | Formate | Acetate | Pyruvate | Oxalate | Malonate | Succinate | Malate |
|---|---|---|---|---|---|---|---|---|---|
| C-GIS samples (times correspond to start of experiment, sample A00 is used to correct for background chemistry) | | | | | | | | | |
| A00[a] | [-00:27] | — | 0.230 | 0.395 | — | 0.199 | 0.019 | 0.049 | 0.422 |
| A01[b] | [00:01] | — | 0.328 | 0.512 | — | 0.212 | 0.016 | 0.105 | 0.748 |
| A02 | [00:16] | — | 0.266 | 0.433 | — | 0.213 | 0.022 | 0.074 | 0.475 |
| A03 | [00:31] | — | 0.340 | 0.783 | — | 0.355 | 0.046 | 0.130 | 0.475 |
| A04 | [00:46] | — | 0.159 | 0.427 | — | 0.178 | 0.017 | 0.048 | 0.149 |
| A05 | [01:01] | — | 0.026 | 0.630 | — | 0.185 | 0.014 | 0.150 | 0.004 |
| A06 | [01:16] | — | 0.230 | 0.570 | — | 0.237 | 0.018 | 0.116 | 0.083 |
| A07 | [01:31] | — | 0.137 | 0.436 | — | 0.189 | 0.018 | 0.044 | 0.052 |
| A08 | [01:46] | — | 0.230 | 0.798 | — | 0.315 | 0.016 | 0.023 | 0.055 |
| A09 | [02:01] | — | 0.374 | 1.469 | — | 0.651 | 0.043 | 0.087 | 0.079 |
| A10 | [02:16] | — | 0.230 | 0.887 | — | 0.355 | 0.031 | 0.058 | 0.026 |
| A11 | [02:31] | — | 0.477 | 2.169 | — | 0.824 | 0.039 | 0.070 | 0.064 |
| A12 | [02:46] | — | 0.170 | 0.908 | — | 0.260 | 0.037 | 0.098 | 0.027 |
| A13 | [03:01] | — | 0.307 | 1.798 | — | 0.474 | 0.028 | 0.021 | 0.018 |
| A14 | [03:16] | — | 0.362 | 2.194 | — | 0.703 | 0.043 | 0.087 | 0.037 |
| PTFE filter samples (collected after the end of the experiment at 3:30) | | | | | | | | | |
| FS[c] | [-00:27] | 0.048 | 0.646 | 0.810 | 0.280 | 6.819 | 0.354 | — | 5.640 |

[a] Sample A00 was collected after the injection of seed particles, when conditions were stable, but prior to the injection of TMB and ozone and the start of the UV-C photolysis.

[b] Sample A01 was collected directly after start of the experiment with the injection of TMB and ozone and the start of the UV-C photolysis.

[c] The experiment was ended by switching of the UC-C lights at 03:30. Filter samples were taken at 03:36.





**Figures**

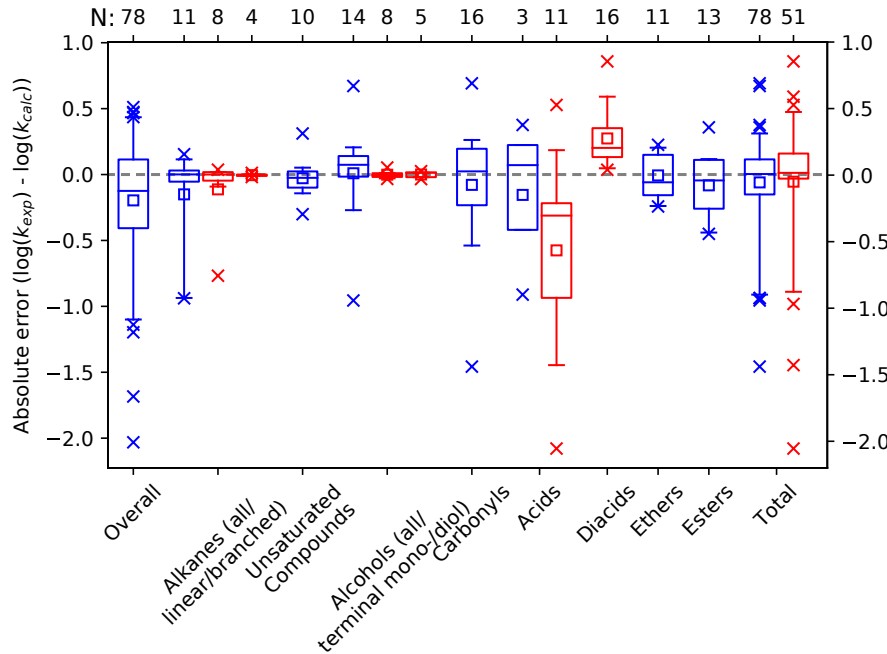

5  **Figure 1 Box plots of the absolute errors of the logarithmised experimental versus predicted OH rate constants for the prediction of OH rate constants with gas-aqueous phase correlations (blue boxes) and extrapolations of homologous series (red boxes). Boxes represent the quartiles of the absolute errors, whiskers the 95% confidence interval, data points outside this are shown explicitly. The arithmetic mean is represented by a square. The 'overall'-box represents absolute errors from a gas-aqueous phase correlation derived from a fit of all data, while the total column denotes the evaluation of all errors compiled from the analysis of the various**
10  **compound classes.**





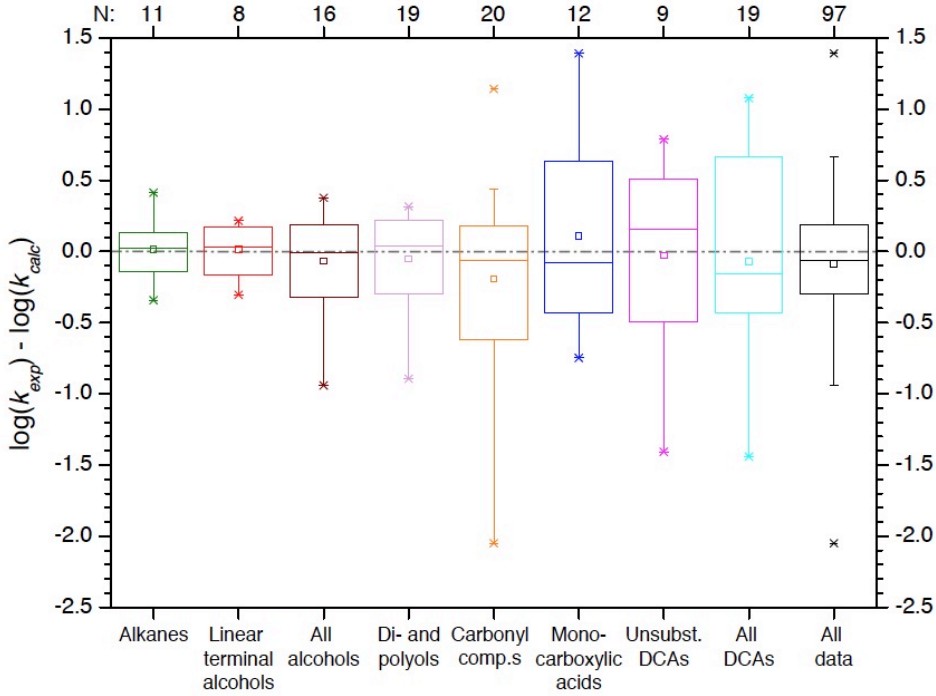

**Figure 2 Box plots of the absolute errors of Evans-Polanyi-type correlations of hydroxyl radical reactions with organic compounds distinguished by compound class.**



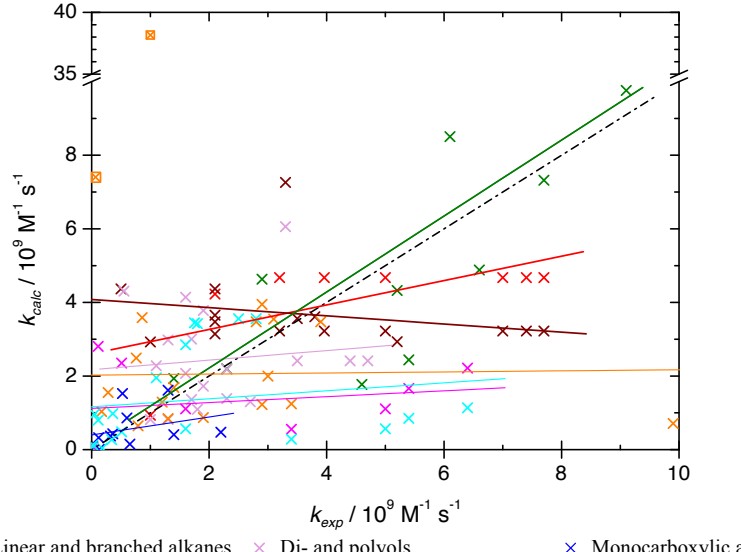

**Figure 3 Plot of predicted versus experimental data for Evans-Polanyi correlations of OH radical reactions with organic compounds differentiated by compound class. Parameters of the regression lines are given in Table S8 in the ESM. The regression lines have the same colour code as the data points. The black dashed-dotted line is the line of same reactivity.**





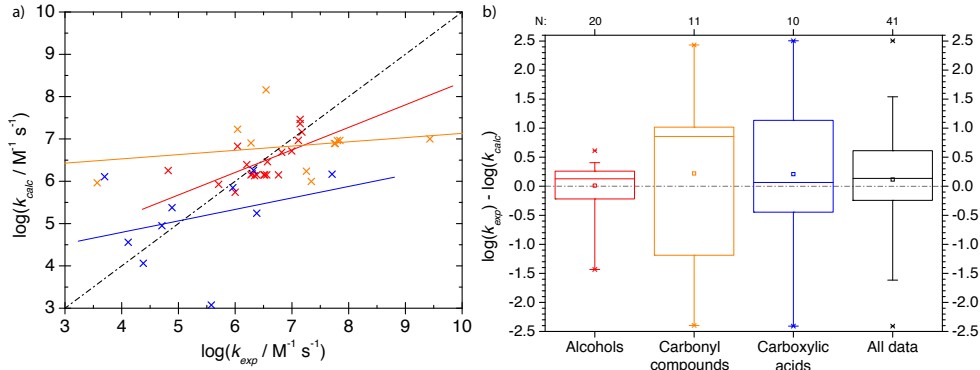

**Figure 4 Evaluation of predicted versus experimental data for Evans-Polanyi-type correlations of NO₃ radical reactions with organic compounds using scatter plots (a) and analysis of absolute errors (b) distinguished by compound class. Parameters for the linear regression lines are given in Table S9 in the ESM.**



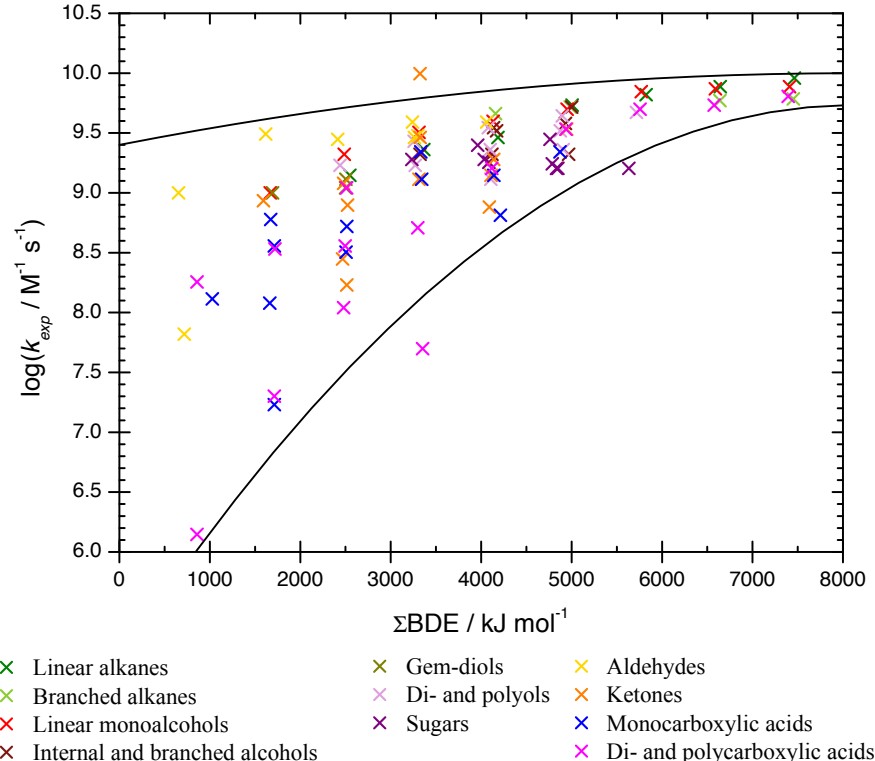

5  **Figure 5** Plot of experimentally determined second order rate constants versus accumulated bond strengths of all bonds containing hydrogen atoms in a molecule (ΣBDE) distinguished by compound classes. Black lines mark upper and lower boundary parabolas with most of the data lying within these boundaries.





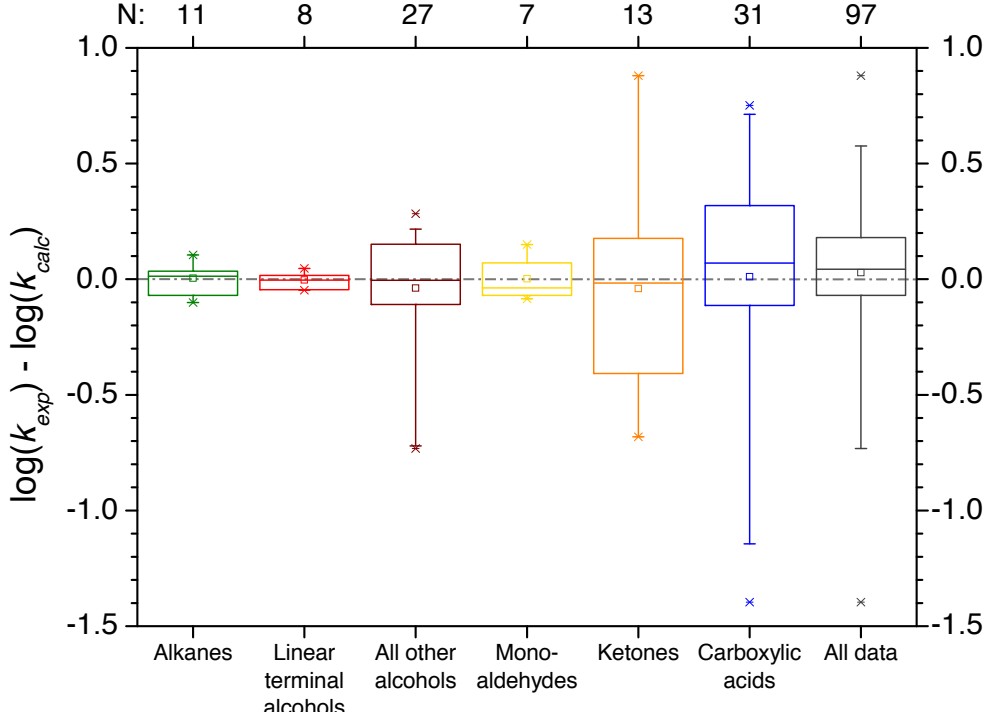

**Figure 6** Box plots of absolute errors of second order OH rate constants predicted with the advanced Evans-Polanyi-type correlations versus experimental data for the various compound classes as well as all data.



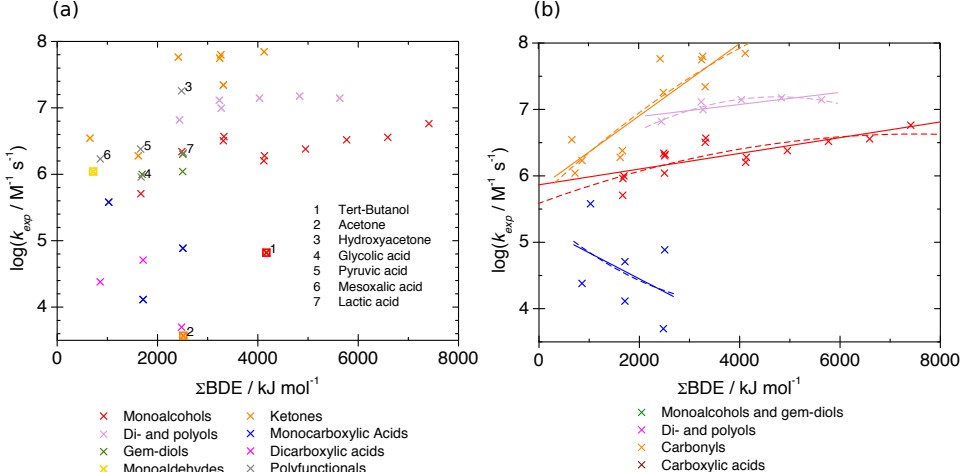

**Figure 7 Experimental second order rate constants versus accumulated bond dissociation enthalpies of all bonds in molecules containing H-atoms (ΣBDE). Subfigure a shows the raw data and subfigure b the final datasets of the combined compound classes and there linear and quadratic regressions.**





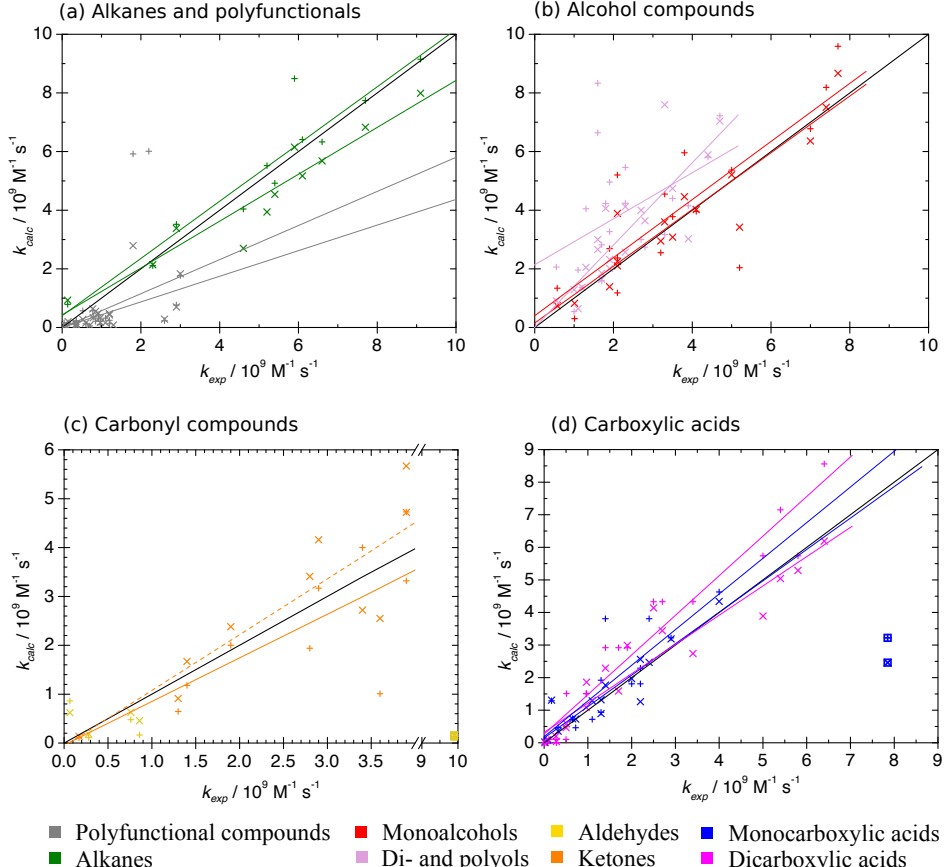

**Figure 8 Scatter plots of predicted versus experimental data for the two SARs by Doussin and Monod (2013) and Minakata et al. (2009). Oblique crosses indicate predicted data by the SAR of Doussin and Monod (2013) with solid regression lines in the same colour. Upright cross are data predicted with the SAR by Minakata et al. (2009) and dashed regression lines in the same colour. The boxed value in subfigure c has been excluded as outlier in the regressions. Regressions in subfigure c include both, aldehyde and ketone data. Parameters for regression lines are given in Table S13 and S14 in the ESM for the data by Doussin and Monod (2013) and Minakata et al. (2009), respectively.**





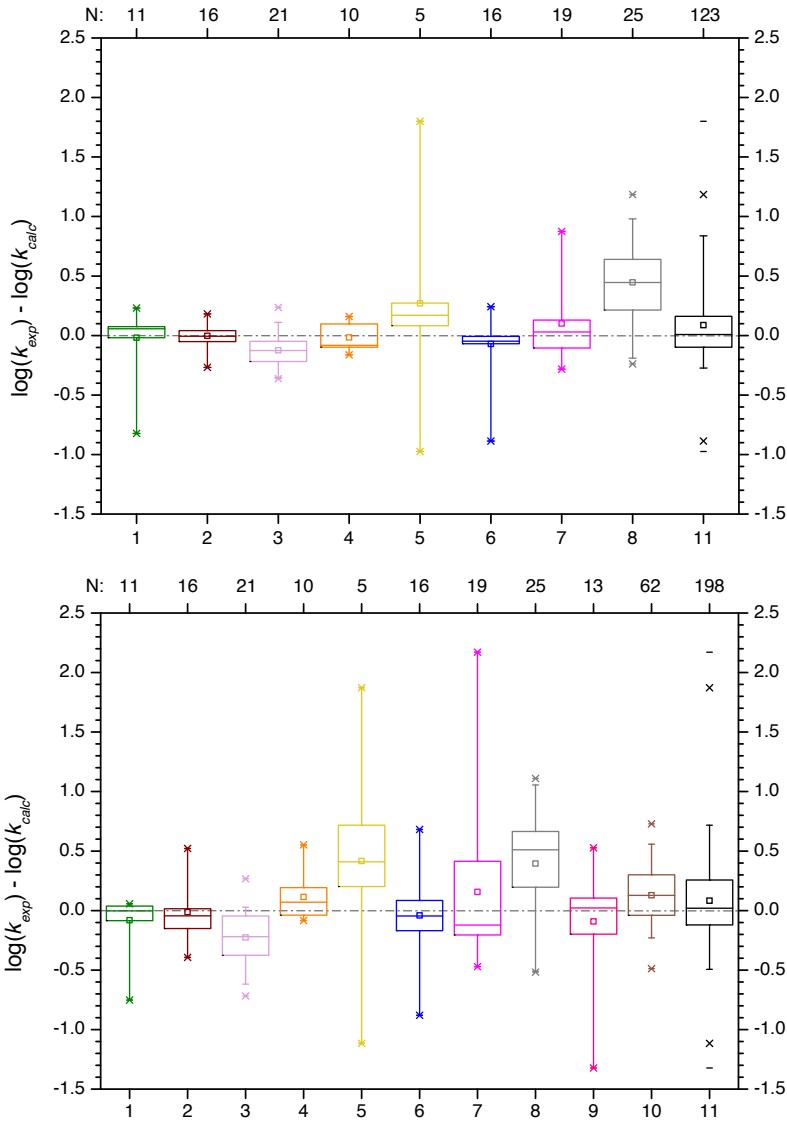

**Figure 9 Box plots of the absolute errors of the logarithmised experimental versus predicted rate constants for the prediction with the SAR of Doussin and Monod (2013 (top) and Minakata et al. (2009) (bottom). Numbers at the abscissa refer to the compound class: 1 – alkanes, 2 – mono-alcohols, 3 – di- and polyols, 4 – carbonyls, 5 – bicarbonyls, 6 – mono-carboxylic acids, 7 – di-carboxylic acids, 8 – polyfunctionals, 9 – unsaturated compounds, 10 – aromatics, 11 – all data.**



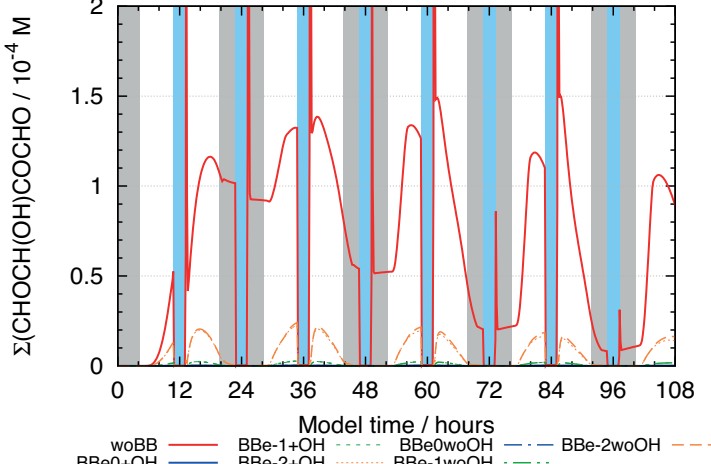

**Figure 10 Concentration-time profiles of aqueous 2-oxo-3-hydroxy-succinaldehyde in all hydration forms in the sensitivity runs investigating the decay of polycarbonyl compounds.**



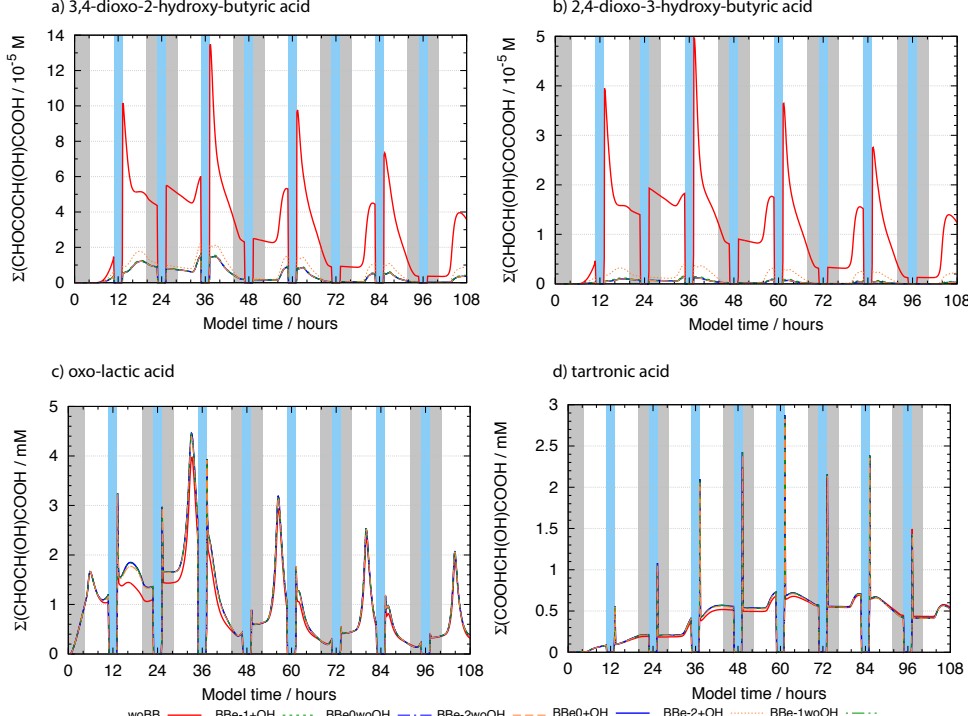

**Figure 11 Accumulated concentration-time profiles of all hydration and dissociation forms of products of 2-oxo-3-hydroxy-succinaldehyde in the sensitivity runs investigating the decay of polycarbonyl compounds.**





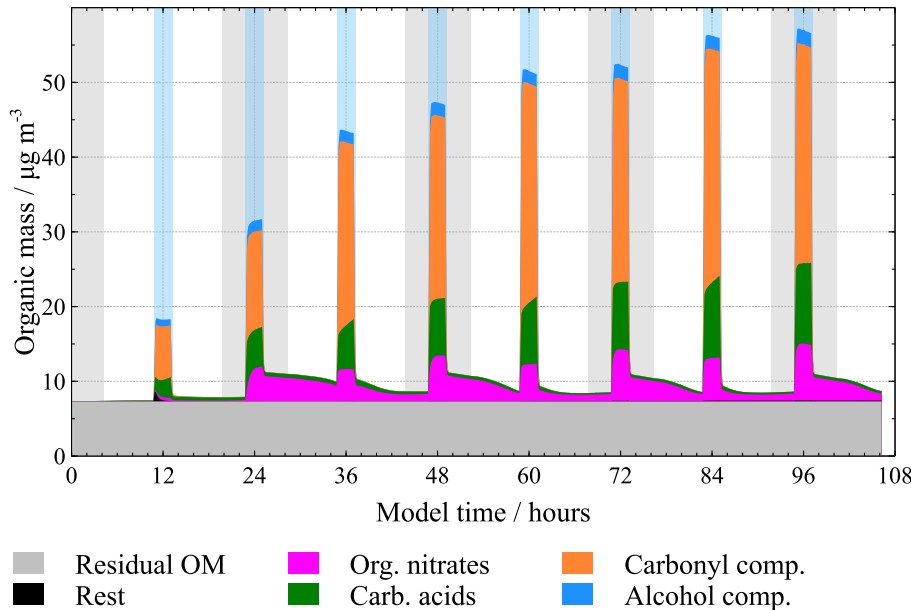

**Figure 12** Particulate organic mass fraction distinguished by constituents in the base scenario using the preliminary mechanism from Section 3.1.



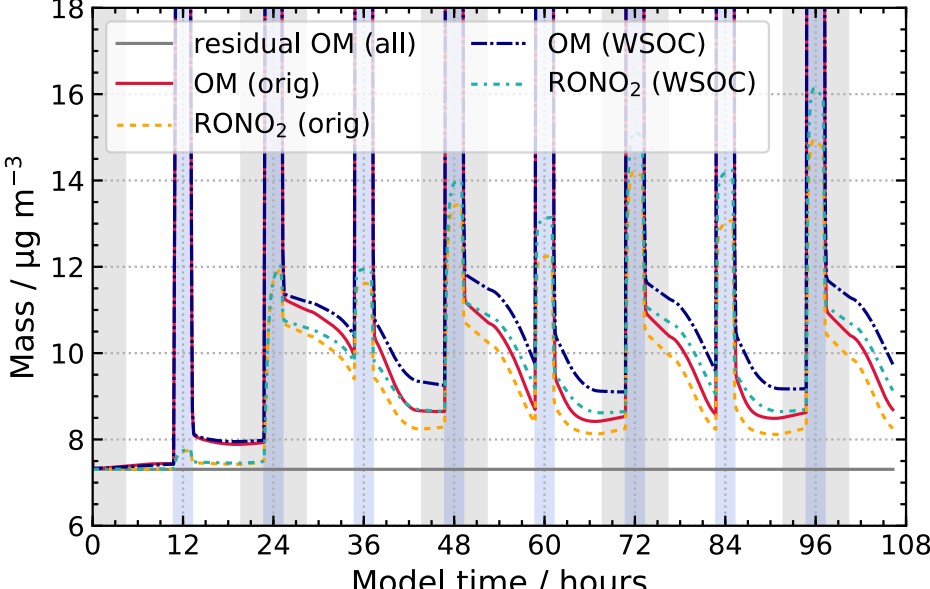

**Figure 13 Processing of the organic mass fraction in the sensitivity runs investigating the influence of parameterisations for WSOC and HULIS chemistry on aqueous-phase chemistry and composition.**



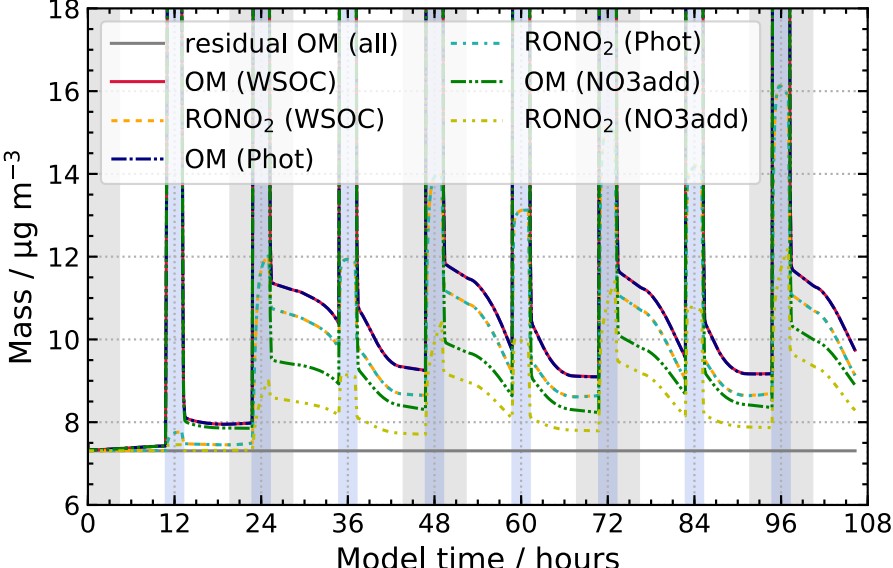

**Figure 14 Processing of the organic mass fraction in the sensitivity runs investigating the influence of sink and source reactions for organic nitrates on aqueous-phase chemistry and composition.**



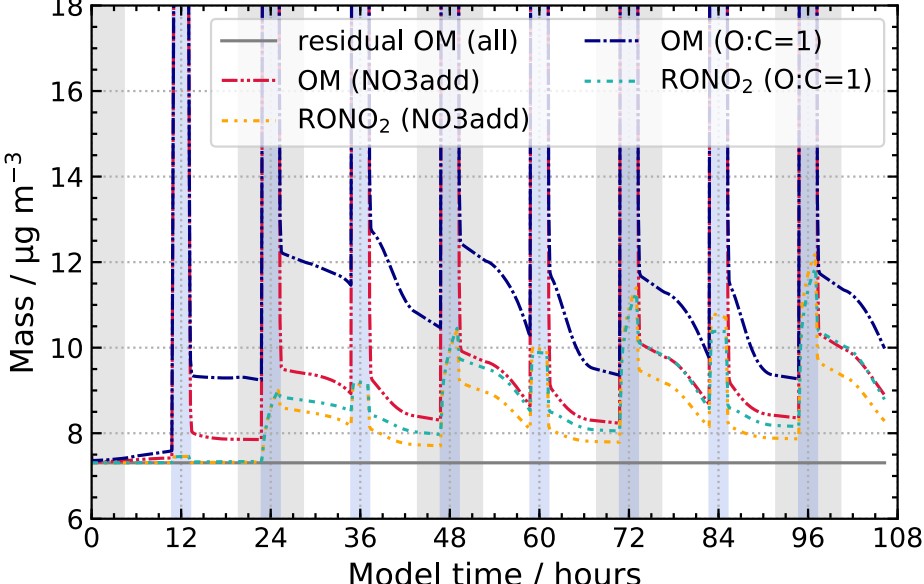

**Figure 15** Processing of the organic mass fraction in the sensitivity runs investigating the influence of phase transfer of oxygenated organic compounds on aqueous-phase chemistry and composition.





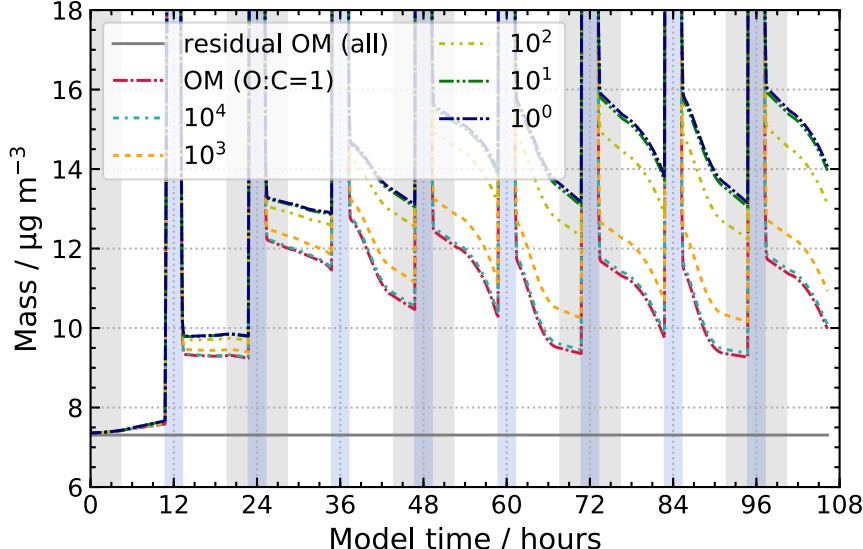

**Figure 16 Processing of the organic mass fraction in the sensitivity runs investigating the influence of alkoxy radical chemistry on aqueous phase chemistry and composition. The terms in parentheses indicate the order of magnitude of the estimate for the bimolecular reaction of alkoxy radicals with dissolved oxygen in $5 \cdot 10^x$ M$^{-1}$ s$^{-1}$.**




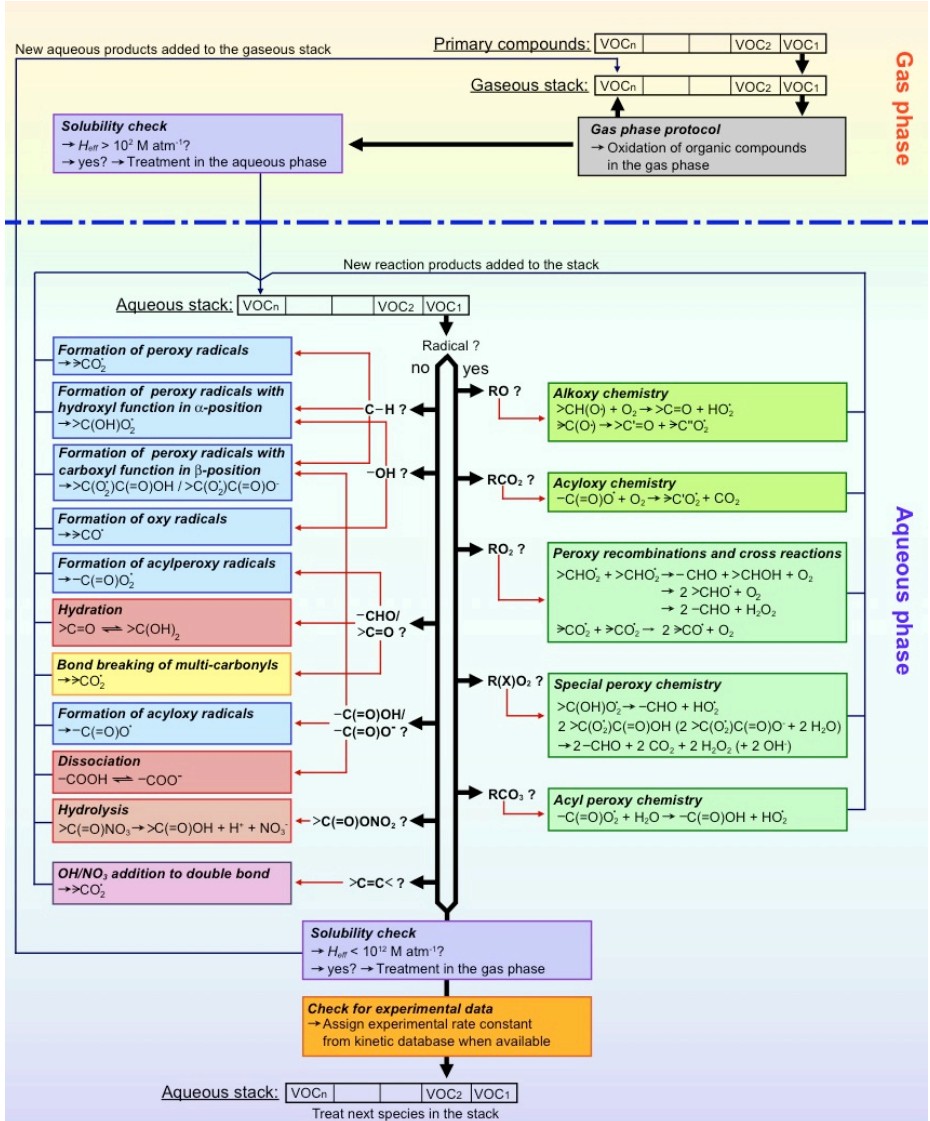

**Figure 17 Workflow of the generator GECKO-A with the processes implemented in the aqueous phase protocol. Blue boxes represent H-abstraction reactions, the pink box represents radical addition to double bonds of unsaturated organic compounds, red boxes represent equilibria, the yellow box represents C–C bond breaking of poly-carbonyls, the light brown box represents the hydrolysis of carbonyl nitrates, and green boxes represent radical chemistry (different shades differentiate between oxy and peroxy radical chemistry).**



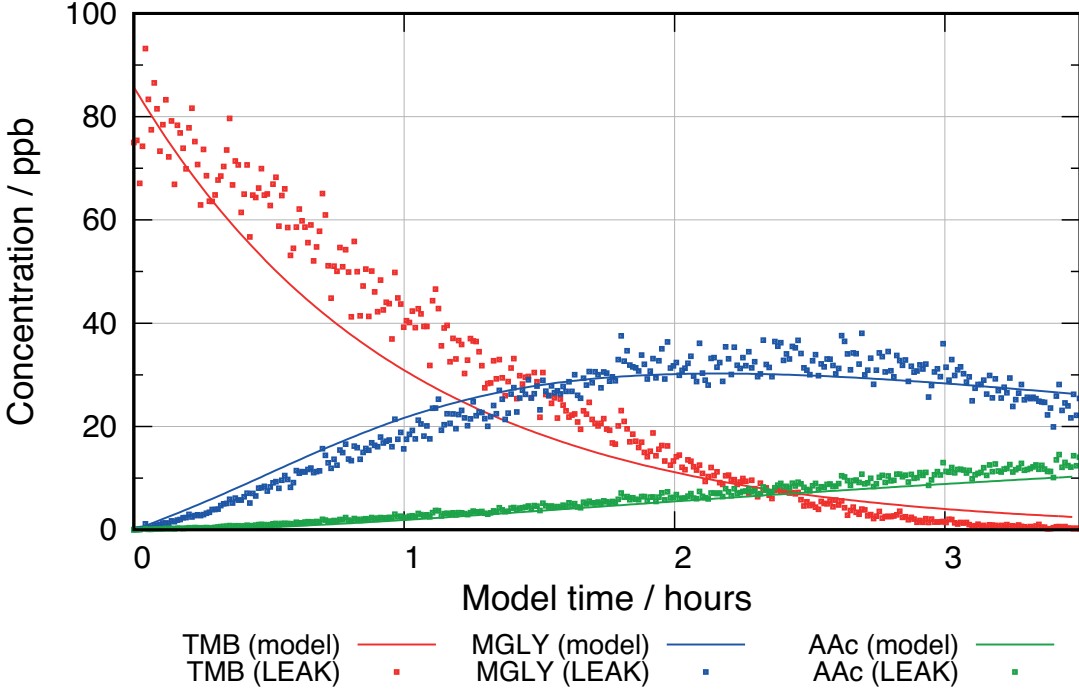

**Figure 18 Comparison of measured (squares) and modelled (lines) gas-phase volume mixing ratios for 1,3,5-trimethyl benzene (TMB, red), methylglyoxal (MGLY, blue), and acetic acid (AAc, green) in the model scenario RXN$_{0.4}$.**

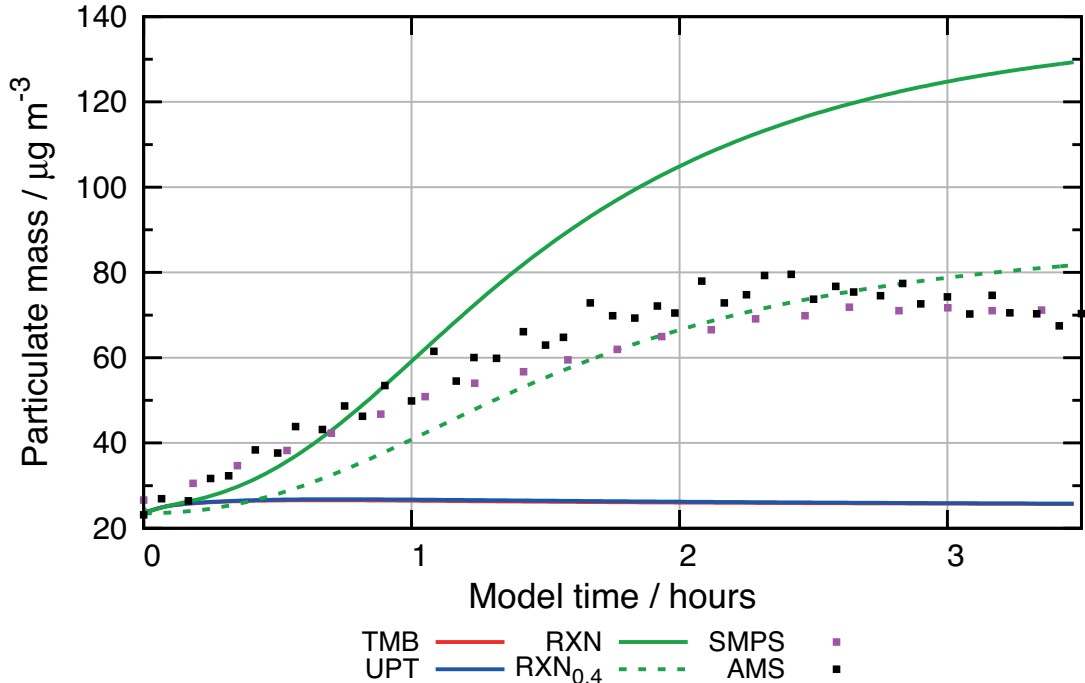

**Figure 19 Comparison of dry particle mass in the various sensitivity studies against SMPS and AMS measurements.**



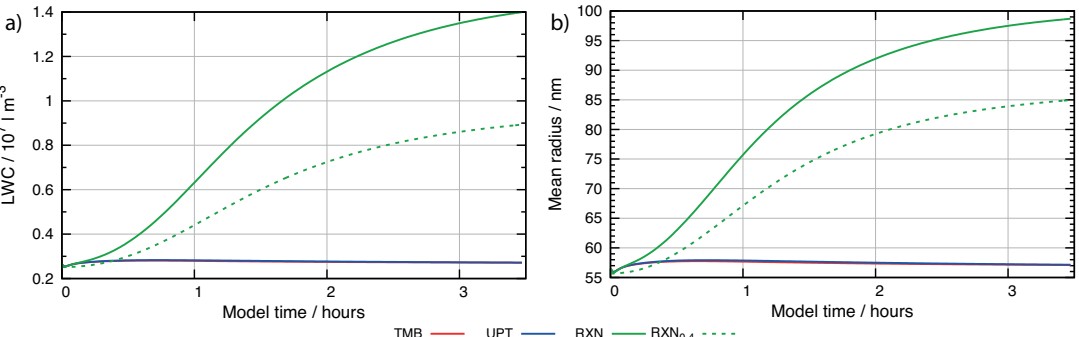

**Figure 20 Profiles of the liquid water content (LWC, subplot a) and mean particle radius (b) for the various model runs.**

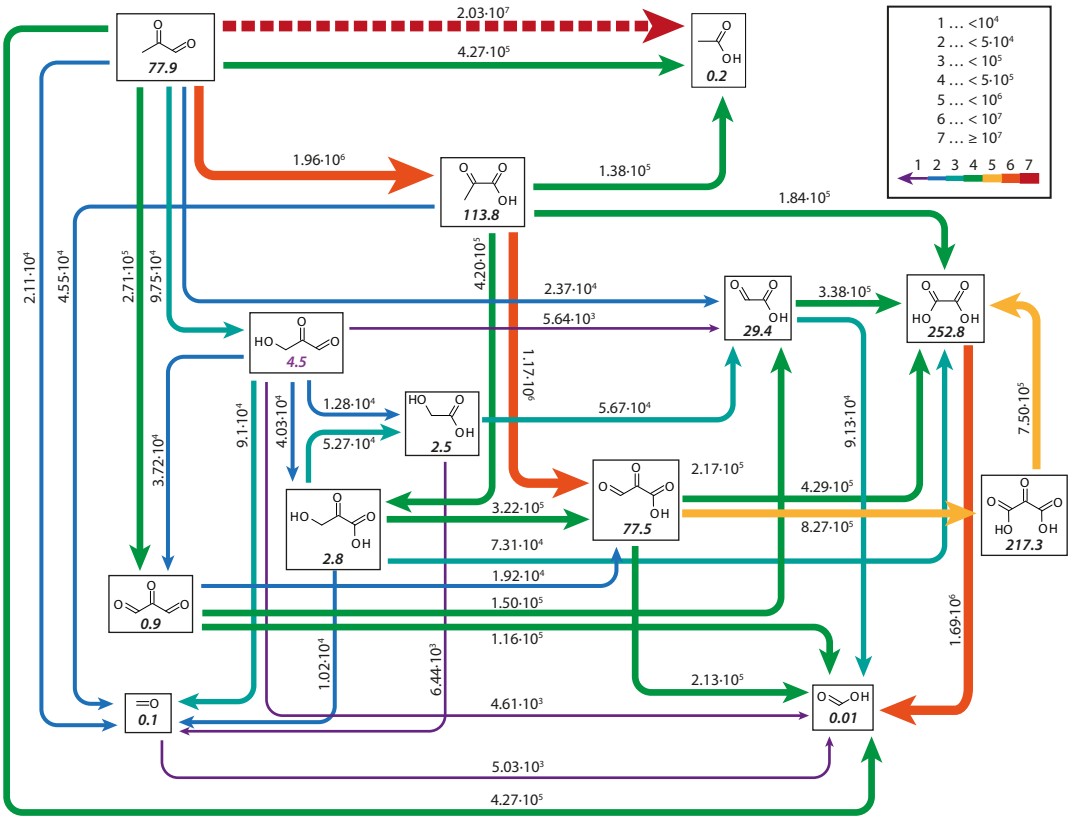

5  **Figure 21 Chemical fluxes in the mesitylene oxidation experiments modelled in the scenario RXN$_{0.4}$ averaged over the whole 3.5 hours of the experiment. Numbers above the arrows correspond to fluxes in molecules cm$^{-3}$ s$^{-1}$, bold numbers in the species boxes are concentrations in ng m$^{-3}$ at the end of the experiment.**