# Peer review of "Development of a protocol for the auto-generation of explicit aqueousphase oxidation schemes of organic compounds"

_Atmospheric Chemistry and Physics, 2018_

## Referee Comment (RC1) · Anonymous Referee #1 · 7 Feb 2019

Although the current understanding of atmospheric organic chemistry, especially for the gas phase, is much better, there is still a big gap to understand aqueous-phase mechanisms. Thus, I think the manuscript is highly suitable for publication in Atmospheric Chemistry and Physics, which I recommend after addressing the following comments:

**Specific comments**

The manuscript is very extensive with a lot of material including also ESM; I would suggest to shorten a bit if possible (especially the text/sentences, which are repeated can be deleted).

Further, I strongly suggest a separate list of abbreviation, which would be very helpful. The full name should be used at the beginning, and after abbreviation can be used throughout the manuscript.

I suggest also checking the English language.

Abstract:
Page 1, line 19: processes instead of mechanisms.
Page 1, line 33: Instead of mesitylene I would use 1,3,5-timethylbenzene or TMB later in the manuscript.

1.Introduction:

Page 2, line 38: Use "auto-generation" throughout the manuscript. "Auto-construction" is not a good choice.

2. Evaluation of kinetic data:
Page 3, lines 12-15 (Overall, a database…) can be deleted; the same is written below (lines 30-32).
Page 3, lines 9-11: If aromatics are excluded here, then this sentence can be deleted.
Page 5, line 1: Correct the sentence.

3. Sensitivity runs:
Page 10, line 12/13: The compound is missing.… "the main degradation product of the OH initiated oxidation of" ? (2-oxo-3-hydroxy-succinaldehyde)
Page 10, line 15: Figure 11c/d shows…(not show)
Page 11, lines 29-41: Please, rewrite this section to make it clearer.

Page 13: Why do you use HLC abbreviation for the Henry's Law coefficient and not as it is usually used?

4. The new multiphase mechanism generator GECKO-A
In Table3 "automated mechanism self-generation" is used; in line 17 (page 15) "automated aqueous-phase mechanism generation", but mostly "auto-generation" is used. I would suggest using the same through the whole article.

5. Validation…
Page 19, line 10: How OH radicals are formed from the photolysis of $O_3$ in the gas phase? During photolysis, O atom is formed, which with $H_2O$ forms OH (RH 75%)?

Page 19, line 30: Check the unit for Henry's law coefficient.

Page 20, line 1:…carboxyl functional groups and not "functions".

The meaning of UPT, RNX, etc. sensitivity runs should be involved in the list of abbreviations.

Page 20, line 41: From Fig. 19, the particle growth (red color) in the model run TMB cannot be seen.

Page 20, line 67/68: Correct the whole sentence: "Furthermore, uptake processes…"

Page22, line 5: Do you have an idea how malic acid is formed?

Page 22, line 27: …has been developed and tested.

Page 23, line 12: Check the unit for Henry's law coefficient.

Table 4: $^{(c)}$…by switching of the UV-C

Figure 9: ..5-dicarbonlys
Figure 13: Add conditions or at least mark cloud and non-cloud periods. This should be done also for other figures.

Figure 19 & 20: TMB cannot not be seen.

ESM
Page 19: First two lines?

Page 24, line 6: Dashed lines in subfigure d (not c).

Page 20: line 10: …plotted as box model in ? (Where?)

Page 24, line 76: The grey dashed-dotted line is not visible.

Page 30: Figure S9: Give the reference for the standard SAR.

Page 31: Correct the title for Figure S10.

Page 32: Figure S12: It is good to shortly explain alpha, beta and gamma scenarios also here.

---

## Referee Comment (RC2) · Andrea Chlebikova (Referee) · 23 Mar 2019

**1   General comments**

The paper presents an interesting extension of existing auto-generation mechanisms to include aqueous-phase reactions. The work is novel, and fits well within the scope of ACP. Thorough literature review work is presented, evaluating data and rate constant prediction methods, carefully discussing the limitations and rationalising the choices made for the new CAPRAM/GECKO-A protocol. The new mechanism is also evaluated against experimental chamber data. Overall, the paper is well-structured, though

rather long. I would recommend trying to shorten the manuscript before publication by scanning for any obviously repeated concepts and superfluous content, as well as carefully reviewing the grammar and ensuring all sentences are easy to parse (I cannot guarantee to have caught all typos, grammatical mistakes, and unclear sentences, but the ones I have are highlighted below – there are also some excess commas in places; please proofread the article again carefully). However, it is definitely suitable for publication in ACP subject to revision.

**2 Specific comments**

While I have found the written text (despite the reservations above) for the most part easy to follow, I believe some of the figures used are quite difficult to read and interpret, and would benefit from replotting.

This applies in particular to Figure 1, where I am struggling to see what the different boxplots represent. Figures 3, 4, 5, 7, S1, S3, S4, S5, S6 and S7 would benefit from using different symbols as markers rather than just different colours, as some of the colours are quite difficult to distinguish. Figure 8 could be made clearer by using more significantly different markers (rather than crosses with different orientations). The figures showing simulation data can only be understood in conjunction with careful reading of the main text, and would benefit from more detailed descriptions in their labels.

I would try to move more of the detail into the ESM, as the main text is currently very long. For example, at present, sections discussed at length in the ESM are still discussed in a lot of detail in the main text (e.g page 2 and page 7), when referencing the ESM should be enough. The text also feels unnecessarily long when sections are introduced carefully, and then go on to repeat the context mentioned previously.

The new "advanced" Evans-Polanyi correlation strikes me as unusual, in that it can no

doubt usefully scale with molecular size – but it would intuitively seem to me that we are losing most of the useful information that the individual bond enthalpies provide by lumping them into one sum. There ought to be a "better" way of accounting for more than the weakest bond than using the sum of BDEs, and perhaps this will be something that can be further improved in the future.

**3  Technical corrections**

p1, line 19: change "was" to "were" for consistency with plural usage of "data" elsewhere

p1, line 30: add "The" at start of sentence

p1, line 39: add "a" before "supplementary tool"

p1, line 40: change "understanding" to "understand"

p2, line 3-4: review sentence

p2, line 36-39: review sentence

p3, line 1-3: review sentence (singular/plural inconsistency)

p3, line 9-11: review sentence

p3, line 12: rephrase to "kinetic data available in the literature"

p3, line 13: get rid of last "reactions"

p3, line 17: add "The" at start of sentence

p3, line 29: review choice of word "actuality"

p3, line 31: add "The" at start of sentence
p4, line 11: add "The" at start of sentence

p4, line 13: review use of word "disqualify" (should be in passive voice)

p4, line 22: add "the" best results

p4, line 32: review use of word "disqualifies" (as above), and remove first "are"

p4, line 33: "functional predication"?

p5, line 1: Arrhenius expression is presented as a mixture of logarithmic and normal form, making it incorrect; also, odd to refer to the atoms as weak, rather than bonds

p5, line 23: rephrase "are resulting"

p6, line 4/5: mistakenly referring to $R^2$ value as correlation coefficient

p6, line 7: placement of brackets should include "molecules"?

p6, line 14: "are" should be "is" in both cases, or else get rid of "a" at the end of the previous line

p7, line 32: "suggest" should say "suggested"

p8, line 3/4/5: review choice of word "increment" here (appears three times in total)

p8, line 7: add "the" literature

p8, line 12: "phosphorous" should say "phosphorus" and would link with hyphen to "containing" for consistency

p8, line 28: "introduce" should say "introduced"

p8, line 34: add "The" at start of sentence

p9, line 7: change comma before "however" to a semicolon

p9, line 28: add comma after "therefore"

p10, line 15/16: change "Figure" to "Figures" or "show" to "shows"; add hyphen to "first"

p11, line 13: check bracket placement/reference towards end of line; also, correct the spelling of "submicron"

p11, line 17: following what?

p11, line 20: add "level of" detail

p12, line 15: format LHS properly

p12, line 22/23: revisit start of sentence

p13, line 8: add "The" at start of first sentence

p13, line 17/18: review sentence

p13, line 29: add "The" at start of first sentence

p13, line 34: add "the" best estimate

p14, line 28: change comma before "however" to semicolon, add comma before "especially"

p14, line 39: change "little" to "few" and add "the" before "literature"

p15, line 13/14: review sentence

p16, line 7: "define" should say "defined"

p16, line 24: either remove "a" or change "constants" to "constant"

p16, line 38: another singular/plural inconsistency at the start of the sentence

p17, line 9: change "with" to "as"

p17, line 23: "gem-dial" should say "gem-diol"

p17, line 24: change "radicals" to "radical" or "this type" to "these types"

p17, line 29: again a singular/plural inconsistency

p17, line 34: "braking" should say "breaking"

p17, line 35: maybe hyphenate "subversion" throughout, or change the word, as this has a different meaning, usually

p17, line 41/42: review sentence

p18, line 1: ditto re "subversions"

p18, line 3-9: rephrase and fix grammar issues

p18, line 10: add "a" new module, change "to"

p18, line 22: change article… In "the" next step

p19, line 39: add article before "reaction product"

p20, line 5: add article before "reaction product"

p20, line 7: remove "with"

p21, line 4: add article before "reaction product"

p21, line 5: change "with" to "at"

p21, line 7: change "has also" to "also has" and add comma before "decreasing"

Inconsistent use of hyphenation of gas-phase/aqueous-phase throughout this section – standardise

p22, line 20: add "The" at start of sentence, and "those" before seen

p22, line 25: change "are" to "is"

p22, line 27: change "a" to "and"

p22, line 38: add "off" between "levelling" and "of"

p23, line 9: "let" should say "led"

p32, in Table 3, entry for "Monomolecular decay of alkoxy radicals": "breaking" not "braking" and "an", not "and"

p32, in Table 3, entry for "Decomposition of acyloxy radicals": "breaking" not "braking"

p33, Table 4 caption: review use of articles; as stands, add "the" before "start"; correct "UC" to "UV"

p40, Figure 7 caption (line 3): "there" should say "their"

p41, Figure 8 caption (line 4): "cross" should say "crosses"

p41, Figure 8 caption (line 5): add "an" outlier

ESM, p15, p18, p20, p21, p22, p23: throughout (including the labels of Tables S3, S5, S6, S7), $R^2$ is mistakenly referred to as the correlation coefficient

ESM, p18, line 5: review "tropospheric relevant organics compounds"

ESM, p18, line 12: change "prove" to "proof"

ESM, p18, line 17: review commas

ESM, p18/19, line 21/1: review sentence

ESM, p19, Figure S2 caption (line 6): when refering to subfigure c likely should be referring to subfigure d, and when referring to subfigure d should be referring to subfigure e

ESM, p20, line 4/5 and beyond: unsure what is going on here – where is the start of the sentence? This whole paragraph repeats the same exact content already described earlier in different words

ESM, p20, line 5: insert "it"

ESM p20, line 10: Figure S4 is not the figure described – should this refer to Figure

S8?

ESM, p20, line 23: "Figure S2d" should say "Figure S2e"

ESM, p25, Table S9, p27, Table S10, p28 Tables S11 and S12: please check the units; I can understand those in Table S8, even though it would perhaps make more sense to say $C$ itself is dimensionless (etc.) unless the division by its units is also added into the equation; for these other tables, the volume part of the units does not seem to add up correctly

ESM, p26, Figure S6 caption (line 6): correct the spelling of "respective"

ESM, p30, Figure S9 caption (line 7): "give" should say "given"

ESM, p31, Figure S10 caption: expand "orig"

ESM, p33, Figure S14 caption: correct the spelling of "investigating"

ESM, p34, line 3: "these type", inconsistent use of singular/plural

ESM, p34, line 4: correct the spelling of "initialised"

ESM, p34, line 9: insert "the" only reaction pathway

---

## Author Comment (AC1) · 17 May 2019

We thank reviewer 1 for the thorough work and the positive and helpful feedback. We adjusted our manuscript accordingly.

> Page 1, line 19: processes instead of mechanisms.
> Page 5, line 1: Correct the sentence.
> Page 10, line 12/13: The compound is missing.… "the main degradation product of the OH
> initiated oxidation of" ? (2-oxo-3-hydroxy-succinaldehyde)
> Page 10, line 15: Figure 11c/d shows…(not show)
> Page 11, lines 29-41: Please, rewrite this section to make it clearer.
> Page 19, line 30: Check the unit for Henry's law coefficient.
> Page 20, line 1:…carboxyl functional groups and not "functions".
> Page 20, line 67/68: Correct the whole sentence: "Furthermore, uptake processes…"
> Page 22, line 27: …has been developed and tested.
> Page 23, line 12: Check the unit for Henry's law coefficient.
> Table 4: (c)…by switching of the UV-C
> Figure 9: ..5-dicarbonlys
> ESM:
> Page 19: First two lines?
> Page 20: line 10: …plotted as box model in ? (Where?)
> Page 24, line 6: Dashed lines in subfigure d (not c).

In particular we corrected all typos, rephrased unclear or misleading sentences, corrected wrong cross-references and units as pointed out in the specific comments listed above.

> Page 2, line 38: Use "auto-generation" throughout the manuscript. "Auto-construction" is not a good choice.
> In Table3 "automated mechanism self-generation" is used; in line 17 (page 15) "automated aqueous-phase mechanism generation", but mostly "auto-generation" is used. I would suggest using the same through the whole article.

Mechanism self-construction was consistently renamed to mechanism auto-generation in the main article and the ESM.

> The manuscript is very extensive with a lot of material including also ESM; I would suggest to shorten a bit if possible (especially the text/sentences, which are repeated can be deleted).
> I suggest also checking the English language.
> Page 3, lines 12-15 (Overall, a database…) can be deleted; the same is written below (lines 30-32).

To shorten the paper, the introductory part of section 2 was removed and partly moved to subsections 2.1 and 2.2. The article was carefully revised for the use of the English language. The analysis of the simple correlations in section 2.2 was shortened. This addresses the following comments:

> Page 3, lines 9-11: If aromatics are excluded here, then this sentence can be deleted.

However, we kept the sentence on p. 3, ll. 9-11 to make the reader aware of the additional aromatics data of the kinetics database, even though the data wasn't used for the CAPRAM/GECKO-A protocol at this stage.

> Further, I strongly suggest a separate list of abbreviation, which would be very helpful. The
> full name should be used at the beginning, and after abbreviation can be used throughout the manuscript.
> Page 1, line 33: Instead of mesitylene I would use 1,3,5-timethylbenzene or TMB later in the manuscript.

The meaning of UPT, RNX, etc. sensitivity runs should be involved in the list of abbreviations.

Furthermore, we have introduced a list of abbreviations in a new appendix A, which is split into different topics/sections. In this context, we have removed Table 2 and moved the contents to appendix A.2. The appendix includes species abbreviations with 1,3,5-trimethylbenzene as TMB as suggest to address the following comments:

Page 19, line 10: How OH radicals are formed from the photolysis of $O_3$ in the gas phase?
During photolysis, O atom is formed, which with $H_2O$ forms OH (RH 75%)?

We introduced two new reactions of

O3 + hv -> O($^1$D) + O2

O($^1$D) + $H_2$O -> 2 OH

to clarify the formation of hydroxyl radicals from ozone photolysis in section 5.

Page 13: Why do you use HLC abbreviation for the Henry's Law coefficient and not as it is usually used?

We have used HLC as abbreviation for Henry's law coefficient in the text of the article rather than the $K_H$ or $H$ as this is a it can be easily associated with Henry's Law coefficient and is a more spoken abbreviation and, hence, fits better with the text flow.

Figure 19 & 20: TMB cannot not be seen.
Page 20, line 41: From Fig. 19, the particle growth (red color) in the model run TMB cannot be seen.

In Figures 19 and 20, the concentrations in the model runs TMB and UPT were almost identical and graphs were on top of each other. Therefore, the concentrations in the run TMB were not visible. To overcome this issue, we introduced new dash-dotted line types for the scenarios UPT and RXN.

Page 24, line 76: The grey dashed-dotted line is not visible.

As the overall carbonyl correlation is identical to the ketone correlation, both graphs are on top of each other. With the previous colour scheme, the grey dash-dotted line was hard to detect. Therefore, we have changed the colour to black and adjusted the figure caption accordingly.

Page 30: Figure S9: Give the reference for the standard SAR.

In Fig. S9 in the ESM, reference to the construction method of the main SAR has been given citing all papers and cross-referencing Table 2 of the main article with the explanation of the construction method.

Page 31: Correct the title for Figure S10.

As Figure 10 has no title, we are not quite sure, what the following comment meant.
However, we detected a mistake in the unit of the organic mass and corrected the y axis label.
Furthermore, we spelled out "organic" in the legend.

Page 32: Figure S12: It is good to shortly explain alpha, beta and gamma scenarios also here.

The caption of Fig. S12 has been expanded to shortly explain the different subversions of the CAPRAM/GECKO-A protocol.

Figure 13: Add conditions or at least mark cloud and non-cloud periods. This should be done also for other figures.

We have expanded on the conditions in all figures of the sensitivity runs and explained the shading of cloud and night-time periods in the plots in the figure captions.

Page22, line 5: Do you have an idea how malic acid is formed?

As stated on page 21, ll. 18-21, the formation of malic acid from TMB cannot be explained with the current knowledge. Therefore, we attributed the formation to background chemistry,

most likely from reactions with the chamber walls and possible residue of previous experiments.

---

## Author Comment (AC3) · 17 May 2019

For a complete overview of the authors' changes, please, find attached the manuscript and ESM with tracked changes.

Please also note the supplement to this comment: https://www.atmos-chem-phys-discuss.net/acp-2018-1318/acp-2018-1318-AC3-supplement.zip

---

## Author Response (AR1)

We thank reviewer 1 for the thorough work and the positive and helpful feedback. We adjusted our manuscript accordingly.

> Page 1, line 19: processes instead of mechanisms.
> Page 5, line 1: Correct the sentence.
> Page 10, line 12/13: The compound is missing.… "the main degradation product of the OH
> initiated oxidation of" ? (2-oxo-3-hydroxy-succinaldehyde)
> Page 10, line 15: Figure 11c/d shows…(not show)
> Page 11, lines 29-41: Please, rewrite this section to make it clearer.
> Page 19, line 30: Check the unit for Henry's law coefficient.
> Page 20, line 1:…carboxyl functional groups and not "functions".
> Page 20, line 67/68: Correct the whole sentence: "Furthermore, uptake processes…"
> Page 22, line 27: …has been developed and tested.
> Page 23, line 12: Check the unit for Henry's law coefficient.
> Table 4: (c)…by switching of the UV-C
> Figure 9: ..5-dicarbonlys
> ESM:
> Page 19: First two lines?
> Page 20: line 10: …plotted as box model in ? (Where?)
> Page 24, line 6: Dashed lines in subfigure d (not c).

In particular we corrected all typos, rephrased unclear or misleading sentences, corrected wrong cross-references and units as pointed out in the specific comments listed above.

> Page 2, line 38: Use "auto-generation" throughout the manuscript. "Auto-construction" is not a good choice.
> In Table3 "automated mechanism self-generation" is used; in line 17 (page 15) "automated aqueous-phase mechanism generation", but mostly "auto-generation" is used. I would suggest using the same through the whole article.

Mechanism self-construction was consistently renamed to mechanism auto-generation in the main article and the ESM.

> The manuscript is very extensive with a lot of material including also ESM; I would suggest to shorten a bit if possible (especially the text/sentences, which are repeated can be deleted).
> I suggest also checking the English language.
> Page 3, lines 12-15 (Overall, a database…) can be deleted; the same is written below (lines 30-32).

To shorten the paper, the introductory part of section 2 was removed and partly moved to subsections 2.1 and 2.2. The article was carefully revised for the use of the English language. The analysis of the simple correlations in section 2.2 was shortened. This addresses the following comments:

> Page 3, lines 9-11: If aromatics are excluded here, then this sentence can be deleted.

However, we kept the sentence on p. 3, ll. 9-11 to make the reader aware of the additional aromatics data of the kinetics database, even though the data wasn't used for the CAPRAM/GECKO-A protocol at this stage.

> Further, I strongly suggest a separate list of abbreviation, which would be very helpful. The
> full name should be used at the beginning, and after abbreviation can be used throughout the manuscript.
> Page 1, line 33: Instead of mesitylene I would use 1,3,5-timethylbenzene or TMB later in the manuscript.

The meaning of UPT, RNX, etc. sensitivity runs should be involved in the list of abbreviations.

Furthermore, we have introduced a list of abbreviations in a new appendix A, which is split into different topics/sections. In this context, we have removed Table 2 and moved the contents to appendix A.2. The appendix includes species abbreviations with 1,3,5-trimethylbenzene as TMB as suggest to address the following comments:

Page 19, line 10: How OH radicals are formed from the photolysis of $O_3$ in the gas phase?
During photolysis, O atom is formed, which with $H_2O$ forms OH (RH 75%)?

We introduced two new reactions of

O3 + hv -> O($^1$D) + O2

O($^1$D) + $H_2$O -> 2 OH

to clarify the formation of hydroxyl radicals from ozone photolysis in section 5.

Page 13: Why do you use HLC abbreviation for the Henry's Law coefficient and not as it is usually used?

We have used HLC as abbreviation for Henry's law coefficient in the text of the article rather than the $K_H$ or $H$ as this is a it can be easily associated with Henry's Law coefficient and is a more spoken abbreviation and, hence, fits better with the text flow.

Figure 19 & 20: TMB cannot not be seen.
Page 20, line 41: From Fig. 19, the particle growth (red color) in the model run TMB cannot be seen.

In Figures 19 and 20, the concentrations in the model runs TMB and UPT were almost identical and graphs were on top of each other. Therefore, the concentrations in the run TMB were not visible. To overcome this issue, we introduced new dash-dotted line types for the scenarios UPT and RXN.

Page 24, line 76: The grey dashed-dotted line is not visible.

As the overall carbonyl correlation is identical to the ketone correlation, both graphs are on top of each other. With the previous colour scheme, the grey dash-dotted line was hard to detect. Therefore, we have changed the colour to black and adjusted the figure caption accordingly.

Page 30: Figure S9: Give the reference for the standard SAR.

In Fig. S9 in the ESM, reference to the construction method of the main SAR has been given citing all papers and cross-referencing Table 2 of the main article with the explanation of the construction method.

As Figure 10 has no title, we are not quite sure, what the following comment meant. However, we detected a mistake in the unit of the organic mass and corrected the y axis label. Furthermore, we spelled out "organic" in the legend.

Page 32: Figure S12: It is good to shortly explain alpha, beta and gamma scenarios also here.

The caption of Fig. S12 has been expanded to shortly explain the different subversions of the CAPRAM/GECKO-A protocol.

Figure 13: Add conditions or at least mark cloud and non-cloud periods. This should be done also for other figures.

We have expanded on the conditions in all figures of the sensitivity runs and explained the shading of cloud and night-time periods in the plots in the figure captions.

Page22, line 5: Do you have an idea how malic acid is formed?

As stated on page 21, ll. 18-21, the formation of malic acid from TMB cannot be explained with the current knowledge. Therefore, we attributed the formation to background chemistry,

most likely from reactions with the chamber walls and possible residue of previous experiments.

We thank Andrea Chlebikova for the very thorough work and the positive and helpful feedback. We adjusted our manuscript accordingly. In particular we corrected all typos, rephrased unclear or misleading sentences, corrected wrong cross-references and units as pointed out in the technical corrections of section 3. In the following, general and specific comments are addressed.

> p6, line 4/5: mistakenly referring to $R^2$ value as correlation coefficient
> ESM, p15, p18, p20, p21, p22, p23: throughout (including the labels of Tables S3, S5, S6, S7), $R^2$ is mistakenly referred to as the correlation coefficient

We have corrected the mis-used term 'correlation coefficient' to 'coefficient of determination'.

> p21, line 7: change "has also" to "also has" and add comma before "decreasing"
> Inconsistent use of hyphenation of gas-phase/aqueous-phase throughout this section – standardise

Concerning the seemingly inconsistent use of 'gas-phase' and 'gas phase', we have used the hyphenated form for the use as adjective as in `gas-phase compounds` and the unhyphenated form to address the gas phase as a noun. Therefore, we believe that the hyphenation is correct in this context. Mechanism self-construction was consistently renamed to mechanism auto-generation.

> While I have found the written text (despite the reservations above) for the most part easy to follow, I believe some of the figures used are quite difficult to read and interpret, and would benefit from replotting. This applies in particular to Figure 1, where I am struggling to see what the different boxplots represent. Figures 3, 4, 5, 7, S1, S3, S4, S5, S6 and S7 would benefit from using different symbols as markers rather than just different colours, as some of the colours are quite difficult to distinguish. Figure 8 could be made clearer by using more significantly different markers (rather than crosses with different orientations).

We have replotted Figures 1, 3-5, 7, S1-S7 with a revised colour and marker scheme. The marker scheme compromises between the distinctness of shapes and shapes that don't cover other data points, when many data are close together in a graph. Moreover, we have expanded on the axis labels in Fig. 1 and introduced an additional legend. Moreover, alternate grey and white shading was used to mark boxplots of corresponding compound classes.

While replotting these figures, some mistakes were correct in the old graphs. This includes some mislabelled data as well as an update of some data from an earlier version of the kinetics database. Correlations in Table S3 and S7 were updated in this context and the tables slightly reformatted.

> I would try to move more of the detail into the ESM, as the main text is currently very long. For example, at present, sections discussed at length in the ESM are still discussed in a lot of detail in the main text (e.g page 2 and page 7), when referencing the ESM should be enough. The text also feels unnecessarily long when sections are introduced carefully, and then go on to repeat the context mentioned previously.

To shorten the paper, the introductory part of section 2 was removed and partly moved to subsections 2.1 and 2.2. The analysis of the simple correlations in section 2.2 was shortened. Furthermore, we have introduced a list of abbreviations in a new appendix A, which is split into different topics/sections. In this context, we have removed Table 2 and moved the contents to appendix A.2.

> ESM, p25, Table S9, p27, Table S10, p28 Tables S11 and S12: please check the units; I can understand those in Table S8, even though it would perhaps make more sense

to say C itself is dimensionless (etc.) unless the division by its units is also added into the equation; for these other tables, the volume part of the units does not seem to add up correctly.

In Tables S8 to S12, correlations are used to derive a second order aqueous-phase rate constant in $M^{-1} s^{-1}$ from the BDE in $kJ\ mol^{-1}$. Therefore, constants in the regression have the same unit as $k_{2nd}$ and are in in $M^{-1} s^{-1}$. For terms with the BDE, the units are $M^{-1} s^{-1} kJ^{-1}$ mol to yield the unit of $k_{2nd}$. With $M = mol\ l^{-1}$, the unit can be simplified to $l\ mol^{-1} s^{-1} kJ^{-1}$ mol = $l\ kJ^{-1} s^{-1}$. For terms with a quadratic BDE, $kJ\ mol^{-1}$ has to be squared, which yields $mol\ l\ kJ^{-2} s^{-1}$ after simplification. To unify all notations, units in Table S8 have been changed to the simplified form. In Tables S10 and S12, $l^2$ was correct to l in the units of the quadratic term.

The figures showing simulation data can only be understood in conjunction with careful reading of the main text, and would benefit from more detailed descriptions in their labels.

We have expanded on the conditions in all figures of the sensitivity runs and explained the shading of cloud and night-time periods in the plots in the figure captions.

The new "advanced" Evans-Polanyi correlation strikes me as unusual, in that it can no doubt usefully scale with molecular size – but it would intuitively seem to me that we are losing most of the useful information that the individual bond enthalpies provide by lumping them into one sum. There ought to be a "better" way of accounting for more than the weakest bond than using the sum of BDEs, and perhaps this will be something that can be further improved in the future.

We thank Andrea Chlebikova for the useful feedback provided. Clearly, there is potential for further improvement of chemical mechanism development on both the experimental and theoretical side. To illustrate this fact, we added the below sentences in section 4.1 about the CAPRAM/GECKO-A protocol on page 15, ll. 28–30:

„There is a need for further development of a prediction method for kinetic and mechanistic data of nitrate radical reactions with organic compounds. However, for more advanced predication methods such as SARs, a more comprehensive experimental database is needed."

Moreover, we have re-phrased the following sentence

"A certain disadvantage of this method is that information about branching ratios and reaction products is lost."

to „… cannot be provided." The previous description implied that the original Evans-Polanyi method was able to provide branching ratios, however, it can only suggest the major reaction products. This information is still available from the individual BDEs, but a quantitative analysis of the branching ratios is not straight-forward.

[revised manuscript text omitted]

**S1 Kinetic data used to design the protocol for mechanism auto-generation**

and show recommended values of reactions of hydroxyl and nitrate radicals, respectively, with organic compounds. The data have been used to validate existing prediction methods for reactions of OH/NO$_3$ radicals with organic compounds, improve or develop new prediction methods and are used directly in the new protocol for mechanism auto-generation as preferred values over theoretically predicted values.

**Table S1 Recommended kinetic data for OH reactions with organic compounds.**

| Reactant | SMILES | GECKO-A | k298 [M⁻¹s⁻¹] | A [M⁻¹s⁻¹] | Ea / R [K] | Reference/remarks |
|---|---|---|---|---|---|---|
| | | | | | | n = 467 (282 aliphatic/185 aromatic) |
| **Alkanes** | | | | | | n = 12 |
| Methane | C | CH4 | $1.1 \cdot 10^8$ | | | Buxton et al. (1988) |
| Ethane | CC | CH3CH3 | $1.4 \cdot 10^9$ | | | Getoff (1989)[1] |
| Propane | CCC | CH3CH2CH3 | $2.3 \cdot 10^9$ | | | Getoff (1989)[1] |
| Butane | CCCC | CH3CH2CH2CH3 | $2.9 \cdot 10^9$ | | | Getoff (1989)[1] |
| Pentane | CCCCC | CH3CH2CH2CH2CH3 | $5.4 \cdot 10^9$ | | | Rudakov et al. (1981)[2] |
| Hexane | CCCCCC | CH3CH2CH2CH2CH2CH3 | $6.6 \cdot 10^9$ | | | Rudakov et al. (1981)[2] |
| Heptane | CCCCCCC | CH3CH2CH2CH2CH2CH2CH3 | $7.7 \cdot 10^9$ | | | Rudakov et al. (1981)[2] |
| Octane | CCCCCCCC | CH3CH2CH2CH2CH2CH2CH2CH3 | $9.1 \cdot 10^9$ | | | Rudakov et al. (1981)[2] |
| Iso-butane | CC(C)C | CH3CH(CH3)CH3 | $4.6 \cdot 10^9$ | | | Rudakov et al. (1981)[2] |
| 2-methyl butane | CCC(C)C | CH3CH2CH(CH3)CH3 | $5.2 \cdot 10^9$ | | | Rudakov et al. (1981)[2] |
| 3-ethyl pentane | CCC(CC)CC | CH3CH2CH(CH2CH3)CH2CH3 | $5.9 \cdot 10^9$ | | | Rudakov et al. (1981)[2] |
| 2,2,4-trimethyl pentane | CC(C)CC(C)(C)C | CH3CH(CH3)CH2C(CH3)(CH3)CH3 | $6.1 \cdot 10^9$ | | | Rudakov et al. (1981)[2] |
| **Alkenes and alkynes** | | | | | | n = 7 |
| Ethylene | C=C | CdH2=CdH2 | $3.1 \cdot 10^9$ | | | Thomas (1967) with updated $k_{ref}$ from NIST database |
| Propylene | CC=C | CH3CdH=CdH2 | $7.0 \cdot 10^9$ | | | Thomas (1967) with updated $k_{ref}$ from NIST database |
| 1-butene | CCC=C | CH3CH2CdH=CdH2 | $7.0 \cdot 10^9$ | | | Thomas (1967) with updated $k_{ref}$ from NIST database |
| Butadien | C=CC=C | CdH2=CdHCdH=CdH2 | $7.0 \cdot 10^9$ | | | Thomas (1967) with updated $k_{ref}$ from NIST database |
| 2,3-dimethyl butadiene | C=C(C)C(C)=C | CdH2=Cd(CH3)Cd(CH3)=CdH2 | $3.1 \cdot 10^{10}$ | | | Moise et al. (2005) |
| Isobutylene | CC(=C)C | CH3Cd(CH3)=CdH2 | $5.4 \cdot 10^9$ | | | Thomas (1967) with updated $k_{ref}$ from NIST database |
| Acetylene | C#C | | $4.7 \cdot 10^9$ | | | Average of Anderson and Schulte-Frohlinde (1978)[1] values |
| **N- and S-substituted alkanes and alkenes** | | | | | | n = 16 |
| Methyl nitrite | CN(=O)(=O) | CH3(NO2) | $1.85 \cdot 10^8$ | | | Anbar et al. (1966a) |
| Methylene nitrite | C=N(=O)(=O) | | $8.5 \cdot 10^9$ | | | Asmus and Taub (1968)[2] |
| Nitroso tert-butane | CC(C)(C)N=O | CH3C(NO)(CH3)CH3 | $4.0 \cdot 10^8$ | | | Bakalik and Thomas (1977)[2] |
| Methanesulfinate | CS(=O)[O-] | | $5.3 \cdot 10^9$ | | | Flyunt et al. (2001) |
| Methanesulfonate | CS(=O)(=O)[O-] | | $1.3 \cdot 10^9$ | | | Lind and Eriksen (1977)[2] |
| Methylsulfate | COS(=O)(=O)[O-] | | $5.0 \cdot 10^7$ | | | Almgren et al. (1979)[2] |
| Hydroxymethylsulfate (HMS⁻) | OCS(=O)(=O)[O-] | | $3.0 \cdot 10^8$ | | | Barlow et al. (1997)[3] |
| Methyldisulfonate monoanion | OS(=O)(=O)CS(=O)(=O)[O-] | | $2.5 \cdot 10^7$ | | | Lind and Eriksen (1977)[2] |
| Methyldisulfonate dianion | [O-]S(=O)(=O)CS(=O)(=O)[O-] | | $4.3 \cdot 10^7$ | | | Lind and Eriksen (1977)[2] |
| Ethyl sulfonate | CCS(=O)(=O)[O-] | | $1.0 \cdot 10^8$ | | | Balazs et al. (1968)[2] |
| Ethyl sulphate | CCOS(=O)(=O)[O-] | | $3.5 \cdot 10^8$ | | | Almgren et al. (1979)[2] |
| Butyl sulphate | CCCCOS(=O)(=O)[O-] | | $1.0 \cdot 10^9$ | | | Almgren et al. (1979)[2] |
| Hexyl sulphate | CCCCCCOS(=O)(=O)[O-] | | $2.5 \cdot 10^9$ | | | Almgren et al. (1979)[2] |
| Octyl sulphate | CCCCCCCCOS(=O)(=O)[O-] | | $6.5 \cdot 10^9$ | | | Almgren et al. (1979)[2] |
| Decyl sulphate | CCCCCCCCCCOS(=O)(=O)[O-] | | $8.2 \cdot 10^9$ | | | Buxton et al. (1988) |
| Methylene sulfonate | C=CS(=O)(=O)[O-] | | $3.5 \cdot 10^9$ | | | Behar et al. (1982)[2] |
| **Monoalcohols** | | | | | | n = 19 |
| Methanol | CO | CH3OH | $1.0 \cdot 10^9$ | $7.0 \cdot 10^9$ | 580 | Average of George et al. (2003); Alam et al. (2003); and Janata (2002) with E/R value of Elliot and McCracken (1989) |
| Ethanol | CCO | CH3CH2OH | $2.1 \cdot 10^9$ | $1.02 \cdot 10^{11}$ | 1200 | Ervens et al. (2003) |
| Propanol | CCCO | CH3CH2CH2OH | $3.2 \cdot 10^9$ | $5.6 \cdot 10^{10}$ | 1000 | Ervens et al. (2003) |
| Butanol | CCCCO | CH3CH2CH2CH2OH | $3.96 \cdot 10^9$ | $1.0 \cdot 10^{11}$ | 1000 | Hesper (2003)[4] |
| Pentanol | CCCCCO | CH3CH2CH2CH2CH2OH | $5.0 \cdot 10^9$ | | | Stemmler and von Gunten (2000) |
| Hexanol | CCCCCCO | CH3CH2CH2CH2CH2CH2OH | $7.0 \cdot 10^9$ | | | Scholes and Willson (1967)[2] |
| Heptanol | CCCCCCCO | CH3CH2CH2CH2CH2CH2CH2OH | $7.4 \cdot 10^9$ | | | Scholes and Willson (1967)[2] |
| octanol | CCCCCCCCO | CH3CH2CH2CH2CH2CH2CH2CH2OH | $7.7 \cdot 10^9$ | | | Scholes and Willson (1967)[2] |
| Iso-propanol | CC(O)C | CH3CH(OH)CH3 | $2.1 \cdot 10^9$ | $6.1 \cdot 10^{10}$ | 962 | Hesper (2003)[4] |
| 2-butanol | CCC(O)C | CH3CH2CH(OH)CH3 | $3.5 \cdot 10^9$ | $7.4 \cdot 10^{10}$ | 910 | Hesper (2003)[4] |
| 3-pentanol | CCC(O)CC | CH3CH2CH(OH)CH2CH3 | $2.1 \cdot 10^9$ | | | Snook and Hamilton (1974) |
| Iso-butanol | CC(C)CO | CH3CH(CH3)CH2(OH) | $3.3 \cdot 10^9$ | | | Buxton et al. (1988) |
| Tert-butanol | CC(O)(C)C | CH3C(OH)(CH3)CH3 | $5.0 \cdot 10^8$ | $3.3 \cdot 10^{10}$ | 1200 | Ervens et al. (2003) |
| 2,2-dimethyl propanol | CC(C)(C)CO | CH3C(CH3)(CH3)CH2(OH) | $5.2 \cdot 10^9$ | | | Walling (1975) with updated $k_{ref}$ from NIST database |
| 1,1-dimethyl propanol | CCC(C)(C)O | CH3CH2C(OH)(CH3)CH3 | $1.9 \cdot 10^9$ | | | Anbar et al. (1966a) |
| Isoamyl alcohol | CC(C)CCO | CH3CH(CH3)CH2CH2(OH) | $3.8 \cdot 10^9$ | | | Buxton et al. (1988) |
| Allyl alcohol | C=CCO | CH2(OH)CdH=CdH2 | $5.9 \cdot 10^9$ | | | Maruthamuthu (1980)[1, 2] |

| Reactant | SMILES | GECKO-A | k298 [M$^{-1}$s$^{-1}$] | A [M$^{-1}$s$^{-1}$] | E$_a$ / R [K] | Reference/remarks | n = 467 (282 aliphatic/185 aromatic) |
|---|---|---|---|---|---|---|---|
| 3-hydroxy-1,4-pentadien | C=CC(O)C=C | CdH2=CdHCH(OH)CdH=CdH2 | 1.0·10$^{10}$ | | | Simic et al. (1973b)[2] | |
| 2,4-hexadienol | CC=CC=CCO | CH3CdH=CdHCdH=CdHCH2(OH) | 9.8·10$^9$ | | | Simic et al. (1973b)[2] | |
| Diols | | | | | | | n = 13 |
| ethylene glycol | OCCO | CH2(OH)CH2(OH) | 1.7·10$^9$ | 9.0·10$^{10}$ | 1191 | Hoffmann et al. (2009) | |
| 1,3-propanediol | OCCCO | CH2(OH)CH2CH2(OH) | 2.7·10$^9$ | 2.5·10$^{11}$ | 1371 | Hoffmann et al. (2009) | |
| 1,4-butanediol | OCCCCO | CH2(OH)CH2CH2CH2(OH) | 3.5·10$^9$ | 2.0·10$^{11}$ | 1203 | Hoffmann et al. (2009) | |
| 1,5-pentanediol | OCCCCCO | CH2(OH)CH2CH2CH2CH2(OH) | 4.4·10$^9$ | 3.1·10$^{11}$ | 1275 | Hoffmann et al. (2009) | |
| 1,6-hexanediol | OCCCCCCO | CH2(OH)CH2CH2CH2CH2CH2(OH) | 4.7·10$^9$ | | | Anbar et al. (1966a) | |
| 1,2-propanediol | CC(O)CO | CH3CH(OH)CH2(OH) | 1.7·10$^9$ | 1.9·10$^{11}$ | 1383 | Hoffmann et al. (2009) | |
| 1,2-butanediol | CCC(O)CO | CH3CH2CH(OH)CH2(OH) | 2.3·10$^9$ | 5.2·10$^{11}$ | 1600 | Hoffmann et al. (2009) | |
| 1,3-butanediol | CC(O)CCO | CH3CH(OH)CH2CH2(OH) | 1.8·10$^9$ | | | Average of Anbar et al. (1966a) and Adams et al. (1965a)[2] | |
| 2,3-butanediol | CC(O)C(O)C | CH3CH(OH)CH(OH)CH3 | 1.3·10$^9$ | | | Adams et al. (1965a)[2] | |
| 2,4-pentanediol | CC(O)CC(O)C | CH3CH(OH)CH2CH(OH)CH3 | 2.3·10$^9$ | | | Ulanski et al. (1994)[1] | |
| 2,3-dimethyl- 2,3-butanediol | OC(C)(C)C(C)(C)O | CH3C(OH)(CH3)C(OH)(CH3)CH3 | 5.5·10$^8$ | | | Anbar et al. (1966a) | |
| 2-butyne-1,4-diol | OCC#CCO | | 7.2·10$^9$ | | | Gilbert and Whitwood (1989)[8]; 78% H-Abs, 22% OH addition | |
| 2,5-dimethyl-3-hexyne-2,5-diol | OC(C)(C)C#CC(C)(C)O | | 3.3·10$^9$ | | | Walling and El-Taliawi (1973a)[2] | |
| Aldehydes and gem-diols | | | | | | | n = 13 |
| Formaldehyde | C=O | CH2O | 1.0·10$^9$ | | | Witter and Neta (1973)[5] | |
| Hydrated formaldehyde | OCO | CH2(OH)(OH) | 1.0·10$^9$ | 3.1·10$^{10}$ | 1022 | Hart et al. (1964)[5]; Chin and Wine (1992) | |
| Acetaldehyde | CC=O | CH3CHO | 3.1·10$^9$ | | | Average of overall rates of Schuchmann and von Sonntag (1988) and Monod et al. (2005); 2.7% H-Abs on methyl group | |
| Hydrated acetaldehyde | CC(O)O | CH3CH(OH)(OH) | 1.3·10$^9$ | | | Schuchmann and von Sonntag (1988); 7,7% H-Abs on methyl group | |
| Propionaldehyde | CCC=O | CH3CH2CHO | 2.8·10$^9$ | 2.6·10$^{11}$ | 1300 | Overall rate constant of Hesper (2003) | |
| Butyraldehyde | CCCC=O | CH3CH2CH2CHO | 3.9·10$^9$ | 8.1·10$^{10}$ | 900 | Overall rate constant of Hesper (2003) | |
| Valeraldehyde | CCCCC=O | CH3CH2CH2CH2CHO | 3.9·10$^9$ | | | Overall rate constant of Monod et al. (2005) | |
| Isobutyraldehyde | CC(C)C=O | CH3CH(CH3)CHO | 2.9·10$^9$ | 3.0·10$^{10}$ | 700 | Gligorovski and Herrmann (2004) | |
| Acrolein | C=CC=O | CHOCdH=CdH2 | 7.0·10$^9$ | | | Lilie and Henglein (1970)[1,2] | |
| Crotonaldehyde | CC=CC=O | CH3CdH=CdHCHO | 5.8·10$^9$ | | | Lilie and Henglein (1970)[2] | |
| Methacrolein | C=C(C)C=O | CH3Cd(CHO)=CdH2 | 9.4·10$^9$ | 5.6·10$^{11}$ | 1200 | Schöne et al. (2014) | |
| Glyoxal | O=CC=O | CHOCHO | 6.6·10$^7$ | | | Draganic and Marcovic, unpublished data[1,2] | |
| Hydrated glyoxal | OC(O)C(O)O | CH(OH)(OH)CH(OH)(OH) | 1.1·10$^9$ | 1.8·10$^{11}$ | 1516 | Buxton et al (1997) | |
| Ketones | | | | | | | n = 12 |
| Acetone | CC(=O)C | CH3COCH3 | 1.7·10$^8$ | 6.9·10$^{10}$ | 1788 | Average of measurements within MOST[6] | |
| Methyl ethyl ketone | CC(=O)CC | CH3CH2COCH3 | 1.3·10$^9$ | 1.7·10$^{11}$ | 1451 | Average of measurement within the MOST project[6]; 10% CH2COCH2CH3 and 90% CH3COCHCH3[6] | |
| Methyl propyl ketone | CC(=O)CCC | CH3CH2CH2COCH3 | 1.9·10$^9$ | | | Adams et al. (1965b)[2] | |
| Diethyl ketone | CCC(=O)CC | CH3CH2COCH2CH3 | 1.4·10$^9$ | | | Adams et al. (1965b)[2] | |
| Methyl vinyl ketone | CC(=O)C=C | CH3COCdH=CdH2 | 7.3·10$^9$ | 9.0·10$^{11}$ | 1443 | Schöne et al. (2014) | |
| Methyl iso-butyl ketone | CC(C)CC(=O)C | CH3COCH2CH(CH3)CH3 | 3.4·10$^9$ | | | Average of measurements within the MOST project[6]; 30% CH3C(CH3)CH2C(=O)CH3 and 70% CH3CH(CH3)CHC(=O)CH3[6] | |
| Methylglyoxal | CC(=O)C=O | CH3COCHO | 8.6·10$^8$ | 8.0·10$^{10}$ | 1350 | Average of Monod et al. (2005); Park et al. (2003)[7]; Stefan and Bolton (1999); and TROPOS measurements | |
| Hydrated methylglyoxal | CC(=O)C(O)O | CH3COCH(OH)(OH) | 7.9·10$^8$ | 1.6·10$^{11}$ | 1589 | Average of measurement within the MOST project[6] | |
| Diacetyl | CC(=O)C(=O)C | CH3COCOCH3 | 2.8·10$^8$ | 4.3·10$^{12}$ | 2880 | Gligorovski and Herrmann (2004) | |
| Acetylacetone | CC(=O)CC(=O)C | CH3COCH2COCH3 | 9.9·10$^9$ | | | Broszkiewicz et al. (1982)[2] | |
| Acetylacetone, conjugate base | CC([O-])=CC(=O)C | | 7.4·10$^9$ | | | Broszkiewicz et al. (1982)[2] | |
| Acetonyl acetone | CC(=O)CCC(=O)C | CH3COCH2CH2COCH3 | 7.6·10$^8$ | 1.1·10$^{11}$ | 1485 | Gligorovski and Herrmann (2004) | |
| Monocarboxylic acids (undissociated and dissociated) | | | | | | | n = 28 |
| Formic acid | C(=O)O | CHO(OH) | 1.3·10$^8$ | 3.7·10$^9$ | 1000 | Buxton et al. (1988) with rounded E/R values of Chin and Wine (1992)/ Adams et al. (1965a) | |
| Formate | C(=O)[O-] | CHO(Om) | 3.2·10$^9$ | 9.2·10$^{10}$ | 1000 | Buxton et al. (1988); Elliot and Simsons (1984) | |
| Acetic acid | CC(=O)O | CH3CO(OH) | 1.7·10$^7$ | 1.5·10$^9$ | 1330 | Chin and Wine (1992) | |
| Acetate | CC(=O)[O-] | CH3CO(Om) | 7.3·10$^7$ | 2.8·10$^{10}$ | 1770 | Chin and Wine (1992) | |
| Propionic acid | CCC(=O)O | CH3CH2CO(OH) | 3.2·10$^8$ | 7.6·10$^{11}$ | 2300 | Ervens et al. (2003) | |
| Propionate | CCC(=O)[O-] | CH3CH2CO(Om) | 7.2·10$^8$ | 3.2·10$^{11}$ | 1800 | Ervens et al. (2003) | |
| Butyric acid | CCCC(=O)O | CH3CH2CH2CO(OH) | 2.2·10$^9$ | | | Scholes and Willson (1967)[2] | |
| Butyrate | CCCC(=O)[O-] | CH3CH2CH2CO(Om) | 2.0·10$^9$ | | | Anbar et al. (1966a) | |
| N-valerate | CCCCC(=O)[O-] | CH3CH2CH2CH2CO(Om) | 2.9·10$^9$ | | | Anbar et al. (1966a) | |

| Reactant | SMILES | GECKO-A | k298 [$M^{-1}s^{-1}$] | A [$M^{-1}s^{-1}$] | $E_a$ / R [K] | Reference/remarks $\qquad$ n = 467 (282 aliphatic/185 aromatic) |
|---|---|---|---|---|---|---|
| Hexanate | CCCCC(=O)[O-] | CH3CH2CH2CH2CH2CO(Om) | $4.0 \cdot 10^9$ | | | Anbar et al. (1966a) |
| Iso-butyrate | CC(C)C(=O)[O-] | CH3CH(CH3) CO(Om) | $1.3 \cdot 10^9$ | | | Anbar et al. (1966a) |
| Iso-valeric acid | CC(C)CC(=O)O | CH3CH(CH3)CH2CO(OH) | $1.4 \cdot 10^9$ | | | Merz and Waters (1949)[1] |
| Iso-valerate | CC(C)CC(=O)[O-] | CH3CH(CH3)CH2CO(Om) | $2.4 \cdot 10^9$ | | | Anbar et al. (1966a) |
| 2-methyl-butyrate | CCC(C)C(=O)[O-] | CH3CH2CH(CH3) CO(Om) | $2.2 \cdot 10^9$ | | | Anbar et al. (1966a) |
| Pivalic acid | CC(C)(C)C(=O)O | CH3C(CH3)(CH3)CO(OH) | $6.5 \cdot 10^8$ | | | Nauser and Bühler (1994) |
| Pivalate | CC(C)(C)C(=O)[O-] | CH3C(CH3)(CH3)CO(Om) | $1.1 \cdot 10^9$ | | | Average of Nauser and Bühler (1994) and Anbar et al. (1966a) |
| 2,2-dimethyl butyrate | CC(C)(C)CC(=O)[O-] | CH3C(CH3)(CH3)CH2CO(Om) | $1.7 \cdot 10^8$ | | | Anbar et al. (1966a) |
| Acrylic acid | C=CC(=O)O | CO(OH)CdH=CdH2 | $5.1 \cdot 10^9$ | $9.4 \cdot 10^{10}$ | 842 | Schöne et al. (2014) |
| Acrylate | C=CC(=O)[O-] | CO(Om)CdH=CdH2 | $5.9 \cdot 10^9$ | $1.8 \cdot 10^{10}$ | 360 | Schöne et al. (2014) |
| Crotonic acid | CC=CC(=O)O | CH3CdH=CdHCO(OH) | $2.9 \cdot 10^9$ | | | Walling and El-Taliawi (1973b)[2] |
| Crotonate | CC=CC(=O)[O-] | CH3CdH=CdHCO(Om) | $5.0 \cdot 10^9$ | | | Lilie and Henglein (1970); Maruthamuthu and Dhandavel (1980)[2] |
| Methacrylic acid | C=C(C)C(=O)O | CH3Cd(CO(OH))=CdH2 | $1.1 \cdot 10^{10}$ | $1.0 \cdot 10^{12}$ | 1320 | Schöne et al. (2014) |
| Methacrylate | C=C(C)C(=O)[O-] | CH3Cd(CO(Om))=CdH2 | $1.1 \cdot 10^{10}$ | $8.0 \cdot 10^{12}$ | 1924 | Schöne et al. (2014) |
| Methylcrotonic acid | CC(C)=CC(=O)O | CH3Cd(CH3)=CdHCO(OH) | $9.0 \cdot 10^9$ | | | Kumar and Rao (1991)[1] |
| Methylcrotonate | CC(C)=CC(=O)[O-] | CH3Cd(CH3)=CdHCO(Om) | $5.9 \cdot 10^9$ | | | Average of different measurements of Kumar and Rao (1991)[1] |
| Sorbic acid | CC=CC=CC(=O)O | CH3CdH=CdHCdH=CdHCO(OH) | $9.8 \cdot 10^9$ | | | Simic et al. (1973b)[1] |
| Linoleate | CCCCCC=CCC=CCCCCCCCC(=O)O | CH3CH2CH2CH2CH2CdH=CdHCH2CdH=CdH... ...CH2CH2CH2CH2CH2CH2CO(OH) | $1.0 \cdot 10^{10}$ | | | Al-Sheikhly et al. (2004) |
| Linolenic acid | CCC=CCC=CCC=CCCCCCCCC(=O)[O-] | CH3CH3CdH=CdHCH2CdH=CdHCH2CdH=CdH... ...CH2CH2CH2CH2CH2CH2CH2CO(Om) | $9.9 \cdot 10^9$ | | | Patterson and Hasegawa (1978)[2]; value scaled by a factor of 1.35 based on the ratio of the values of Patterson and Hasegawa (1978) and Al-Sheikhly et al. (2004) for linoleate |
| **Dicarboxylic acids (undissociated and dissociated)** | | | | | | **n = 26** |
| Oxalic acid | OC(=O)C(=O)O | CO(OH)CO(OH) | $1.4 \cdot 10^6$ | | | Getoff et al. (1971) |
| Oxalate monoanion | OC(=O)C(=O)[O-] | CO(Om)CO(OH) | $1.9 \cdot 10^8$ | $2.5 \cdot 10^{12}$ | 2800 | Ervens et al. (2003) |
| Oxalate dianion | [O-]C(=O)C(=O)[O-] | CO(Om)CO(Om) | $1.6 \cdot 10^8$ | $4.6 \cdot 10^{14}$ | 4300 | Ervens et al. (2003) |
| Malonic acid | OC(=O)CC(=O)O | CO(OH)CH2CO(OH) | $2.0 \cdot 10^7$ | | | Buxton et al. (1988) |
| Malonate monoanion | OC(=O)CC(=O)[O-] | CO(Om)CH2CO(OH) | $6.0 \cdot 10^7$ | $3.2 \cdot 10^9$ | 1300 | Ervens et al. (2003) |
| Malonate dianion | [O-]C(=O)CC(=O)[O-] | CO(Om)CH2CO(Om) | $2.7 \cdot 10^8$ | | | Average of Logan (1989) and Adams et al. (1965a) |
| Succinic acid | OC(=O)CCC(=O)O | CO(OH)CH2CH2CO(OH) | $1.1 \cdot 10^8$ | $8.1 \cdot 10^9$ | 1278 | Ervens et al. (2003) |
| Succinate monoanion | OC(=O)CCC(=O)[O-] | CO(Om)CH2CH2CO(OH) | $2.6 \cdot 10^8$ | $1.2 \cdot 10^{11}$ | 1808 | TROPOS measurements |
| Succinate dianion | [O-]C(=O)CCC(=O)[O-] | CO(Om)CH2CH2CO(Om) | $5.0 \cdot 10^8$ | $5.4 \cdot 10^{10}$ | 1413 | Ervens et al. (2003) |
| Glutaric acid | OC(=O)CCCC(=O)O | CO(OH)CH2CH2CH2CO(OH) | $5.1 \cdot 10^8$ | $2.5 \cdot 10^{10}$ | 1164 | TROPOS measurements |
| Glutarate dianion | [O-]C(=O)CCCC(=O)[O-] | CO(Om)CH2CH2CH2CO(Om) | $8.2 \cdot 10^8$ | $2.5 \cdot 10^{12}$ | 2355 | TROPOS measurements |
| Adipic acid | OC(=O)CCCCC(=O)O | CO(OH)CH2CH2CH2CH2CO(OH) | $1.6 \cdot 10^9$ | $3.2 \cdot 10^{11}$ | 1479 | TROPOS measurements |
| Adipate monoanion | OC(=O)CCCCC(=O)[O-] | CO(Om)CH2CH2CH2CH2CO(OH) | $1.4 \cdot 10^9$ | $9.7 \cdot 10^{12}$ | 2641 | TROPOS measurements |
| Adipate dianion | [O-]C(=O)CCCCC(=O)[O-] | CO(Om)CH2CH2CH2CH2CO(Om) | $1.4 \cdot 10^9$ | $1.2 \cdot 10^{12}$ | 1985 | TROPOS measurements |
| Pimelic acid | OC(=O)CCCCCC(=O)O | CO(OH)CH2CH2CH2CH2CH2CO(OH) | $3.4 \cdot 10^9$ | $3.0 \cdot 10^{11}$ | 1335 | TROPOS measurements |
| Pimelate monoanion | OC(=O)CCCCCC(=O)[O-] | CO(Om)CH2CH2CH2CH2CH2CO(OH) | $1.8 \cdot 10^9$ | $9.7 \cdot 10^{12}$ | 2641 | TROPOS measurements |
| Pimelate dianion | [O-]C(=O)CCCCCC(=O)[O-] | CO(Om)CH2CH2CH2CH2CH2CO(Om) | $2.1 \cdot 10^9$ | $1.7 \cdot 10^{13}$ | 2514 | TROPOS measurements |
| Suberic acid | OC(=O)CCCCCCC(=O)O | CO(OH)CH2CH2CH2CH2CH2CH2CO(OH) | $5.0 \cdot 10^9$ | | | Hesper (2003) |
| Suberate dianion | [O-]C(=O)CCCCCCC(=O)[O-] | CO(Om)CH2CH2CH2CH2CH2CH2CO(Om) | $5.8 \cdot 10^9$ | | | Hesper (2003) |
| Azelaic acid | OC(=O)CCCCCCCC(=O)O | CO(OH)CH2CH2CH2CH2CH2CH2CH2CO(OH) | $5.4 \cdot 10^9$ | | | Scholes and Willson (1967)[2] |
| Sebacic acid | OC(=O)CCCCCCCCC(=O)O | CO(OH)CH2CH2CH2CH2CH2CH2CH2CH2CO(OH) | $6.4 \cdot 10^9$ | | | Scholes and Willson (1967)[2] |
| Fumaric acid | OC(=O)/C=C/C(=O)O | CO(OH)CdH=CdHCO(OH) | $6.0 \cdot 10^9$ | | | Cabelli and Bielski (1985)[2] |
| Maleic acid | OC(=O)/C=C\C(=O)O | CO(OH)CdH=CdHCO(OH) | $6.0 \cdot 10^9$ | | | Cabelli and Bielski (1985)[2] |
| 3-hexene-1,6-dioate dianion | [O-]C(=O)CC=CCC(=O)[O-] | CO(Om)CH2CdH=CdHCH2CO(Om) | $5.9 \cdot 10^9$ | $6.3 \cdot 10^{11}$ | 1395 | Elliot and McCracken (1989)[1] |
| Crocetin | OC(=O)C(C)=CC=CC(C)=CC=C... ...C=C(C)C=CC=C(C)C(=O)O | CH3Cd(CO(OH))=CdHCdH=CdHCd(CH3)=CdH... ...CdH=CdHCdH=Cd(CH3)CdH=CdH... ...CdH=Cd(CH3)CO(OH) | $2.3 \cdot 10^{10}$ | | | Bors et al. (1982)[2] |
| Acetylenedicarboxylate dianion | [O-]C(=O)C#CC(=O)[O-] | | $2.6 \cdot 10^9$ | | | Simhon et al. (1988)[1] |
| **Organic peroxides** | | | | | | **n = 3** |
| Methyl hydrogen peroxide | COO | CH3(OOH) | $3.2 \cdot 10^8$ | | | Average of Graedel and Weschler (1981) and Monod et al. (2007) |
| Ethyl hydrogen peroxide | CCOO | CH3CH2(OOH) | $5.8 \cdot 10^8$ | | | Monod et al. (2007) |
| Tert-butyl hydrogen peroxide | CC(C)(C)OO | CH3C(OOH)(CH3)CH3 | $8.0 \cdot 10^7$ | | | Phulkar et al. (1990)[1] |
| **Ethers** | | | | | | **n = 17** |
| Dimethyl ether | COC | CH3-O-CH3 | $1.0 \cdot 10^9$ | | | Eibenberger (1980)[2] |

| Reactant | SMILES | GECKO-A | $k_{298}$ [M$^{-1}$s$^{-1}$] | A [M$^{-1}$s$^{-1}$] | $E_a$ / R [K] | Reference/remarks | n = 467 (282 aliphatic/185 aromatic) |
|---|---|---|---|---|---|---|---|
| Diethyl ether | CCOCC | CH3CH2-O-CH2CH3 | $3.6 \cdot 10^9$ | | | Buxton et al. (1988) | |
| Methylal | COCOC | CH3-O-CH2-O-CH3 | $1.2 \cdot 10^9$ | | | Eibenberger (1980)[2] | |
| Diethoxy methane | CCOCOCC | CH3CH2-O-CH2-O-CH2CH3 | $1.6 \cdot 10^9$ | | | Anbar et al. (1966a) | |
| Ethylene glycol dimethyl ether | COCCOC | CH3-O-CH2CH2-O-CH3 | $1.6 \cdot 10^9$ | | | Anbar et al. (1966a) | |
| Ethylene glycol diethyl ether | CCOCCOCC | CH3CH2-O-CH2CH2-O-CH2CH3 | $2.3 \cdot 10^9$ | | | Anbar et al. (1966a) | |
| Diethylene glycol diethyl ether | CCOCCOCCOCC | CH3CH2-O-CH2CH2-O-CH2CH2-O-CH2CH3 | $3.2 \cdot 10^9$ | | | Anbar et al. (1966a) | |
| Hydroxybutyl vinyl ether | OCCCCOC=C | CH2(OH)CH2CH2CH2-O-CdH=CdH2 | $1.9 \cdot 10^9$ | | | Moise et al. (2005) | |
| di(ethylene glycol) divinyl ether | C=COCCOCCOC=C | CdH2=CdH-O-CH2CH2-O-CH2CH2-O-CdH=CdH2 | $2.3 \cdot 10^{10}$ | | | Moise et al. (2005) | |
| tri(ethylene glycol) divinyl ether | C=COCCOCCOCCOC=C | CdH2=CdH-O-CH2CH2-O-CH2CH2-O-CH2...
...CH2-O-CdH=CdH2 | $1.54 \cdot 10^{10}$ | $2.3 \cdot 10^{13}$ | 2179 | Gligorovski et al. (2009) | |
| Dimethyl acetal | COC(C)OC | CH3-O-CH(CH3)-O-CH3 | $2.2 \cdot 10^9$ | | | Eibenberger (1980)[2] | |
| Methyl tert-butyl ether | CC(C)(C)OC | CH3-O-C(CH3)(CH3)CH3 | $2.1 \cdot 10^9$ | | | Average of Eibenberger (1980)[2,5], Chang and Young (2000)[5], Adams et al. (1965a)[5], Mitani et al. (2002)[5], and Garoma and Gurol (2005) | |
| Ethyl tert-butyl ether | CC(C)(C)OCC | CH3CH2-O-C(CH3)(CH3)CH3 | $1.5 \cdot 10^9$ | $1.2 \cdot 10^{10}$ | 580 | Monod et al. (2005) | |
| Di(iso-propyl) ether | CC(C)OC(C)C | CH3CH(CH3)-O-CH(CH3)CH3 | $2.49 \cdot 10^9$ | | | Mezyk et al. (2001)[3] | |
| Di(tert-butyl) ether | CC(C)(C)OC(C)(C)C | CH3C(CH3)(CH3)-O-C(CH3)(CH3)CH3 | $1.81 \cdot 10^9$ | | | Mezyk et al. (2001)[3] | |
| 2-methyl-2-methoxy-butane | CCC(C)(C)OC | CH3CH2C(CH3)(CH3)-O-CH3 | $2.37 \cdot 10^9$ | | | Mezyk et al. (2001)[3] | |
| 2,4-dimethoxy pentane | COC(C)CC(C)OC | CH3-O-CH(CH3)CH2CH(CH3)-O-CH3 | $3.7 \cdot 10^9$ | | | Janik et al. (2000)[3] | |

| Esters | | | | | | | n = 26 |
|---|---|---|---|---|---|---|---|
| Ethyl formate | CCOC=O | CH3CH2-O-CHO | $7.9 \cdot 10^8$ | $9.3 \cdot 10^{11}$ | 2106 | Average of measurements within the MOST project[6]; 42.3% C$_2$H$_5$COCO and 57.7% CH$_3$C(H)OCHO[6] | |
| Butyl formate | CCCCOC=O | CH3CH2CH2CH2-O-CHO | $2.8 \cdot 10^9$ | | | Stemmler and von Gunten (2000) | |
| Tert-butyl formate | CC(C)(C)OC=O | CH3C(CH3)(CH3)-O-CHO | $7.0 \cdot 10^8$ | | | Acero et al. (2001)[5] | |
| Methyl acetate | CC(=O)OC | CH3CO-O-CH3 | $1.5 \cdot 10^8$ | | | Average of Anbar et al. (1966a) and Adams et al. (1965a)[2] | |
| Methyl propionate | CCC(=O)OC | CH3CH2CO-O-CH3 | $4.5 \cdot 10^8$ | | | Adams et al. (1965a)[2] | |
| Ethyl acetate | CC(=O)OCC | CH3CO-O-CH2CH3 | $4.0 \cdot 10^8$ | | | Adams et al. (1965a)[2] | |
| Ethyl propionate | CCC(=O)OCC | CH3CH2CO-O-CH2CH3 | $5.2 \cdot 10^8$ | | | Average of Adams et al. (1965a)[8] and Bíró and Wojnárovits (1992)[1] | |
| Propyl acetate | CCCOC(=O)C | CH3CH2CH2-O-COCH3 | $1.4 \cdot 10^9$ | | | Adams et al. (1965a)[2] | |
| Methyl butyrate | CCCC(=O)OC | CH3CH2CH2CO-O-CH3 | $1.7 \cdot 10^9$ | | | Adams et al. (1965a)[1] | |
| Butyl acetate | CCCCOC(=O)C | CH3CH2CH2CH2-O-COCH3 | $1.8 \cdot 10^9$ | $5.33 \cdot 10^{10}$ | 1000 | Monod et al. (2005) | |
| Ethyl butyrate | CCCC(=O)OCC | CH3CH2CH2CO-O-CH2CH3 | $1.6 \cdot 10^9$ | | | Adams et al. (1965a)[2] | |
| Butyl propionate | CCCCOC(=O)CC | CH3CH2CH2CH2-O-COCH2CH3 | $1.6 \cdot 10^9$ | | | Adams et al. (1965a)[2] | |
| Iso-propyl acetate | CC(C)OC(=O)C | CH3CO-O-CH(CH3)CH3 | $5.3 \cdot 10^8$ | | | Average of Hardison et al. (2002)[5,7] and Adams et al. (1965a)[1] | |
| Methyl acrylate | C=CC(=O)OC | CH3-O-COCdH=CdH2 | $5.3 \cdot 10^9$ | | | Kumar et al. (1988)[1] | |
| Ethyl acrylate | C=CC(=O)OCC | CH3CH2-O-COCdH=CdH2 | $5.8 \cdot 10^9$ | | | Kumar et al. (1988)[1] | |
| Butyl acrylate | C=CC(=O)OCCCC | CH3CH2CH2CH2-O-COCdH=CdH2 | $5.5 \cdot 10^9$ | | | Kumar et al. (1988)[1] | |
| Methyl methacrylate | C=C(C)C(=O)OC | CH3-O-COCd(CH3)=CdH2 | $1.2 \cdot 10^{10}$ | | | Average of Maruthamuthu (1980)[2] and Kumar et al. (1988)[1] | |
| Butyl methacrylate | C=C(C)C(=O)OCCCC | CH3CH2CH2CH2-O-COCd(CH3)=CdH2 | $1.2 \cdot 10^{10}$ | | | Kumar et al. (1988)[1] | |
| Dimethyl malonate | COC(=O)CC(=O)OC | CH3-O-COCH2CO-O-CH3 | $2.7 \cdot 10^8$ | | | George et al. (2003) | |
| Diethyl malonate | CCOC(=O)CC(=O)OCC | CH3CH2-O-COCH2CO-O-CH2CH3 | $6.5 \cdot 10^8$ | | | Adams et al. (1965a)[1] | |
| Dimethyl succinate | COC(=O)CCC(=O)OC | CH3-O-COCH2CH2CO-O-CH3 | $5.3 \cdot 10^8$ | | | George et al. (2003) | |
| Diethyl succinate | CCOC(=O)CCC(=O)OCC | CH3CH2-O-COCH2CH2CO-O-CH2CH3 | $7.8 \cdot 10^8$ | | | Adams et al. (1965a)[2] | |
| Diethyl meleate | CCOC(=O)/C=C\C(=O)OCC | CH3CH2-O-CHO | $5.9 \cdot 10^9$ | | | Bíró and Wojnárovits (1996) | |
| Diethyl fumarate | CCOC(=O)/C=C/C(=O)OCC | CH3CH2-O-CHO | $5.9 \cdot 10^9$ | | | Bíró and Wojnárovits (1996) | |
| Dimethyl carbonate | COC(=O)OC | CH3-O-CO-O-CH3 | $5.1 \cdot 10^7$ | | | George et al. (2003) | |
| Diethyl carbonate | CCOC(=O)OCC | CH3CH2-O-CO-O-CH2CH3 | $7.9 \cdot 10^8$ | | | George et al. (2003) | |

| Aliphatic polyfunctional compounds | | | | | | | n = 43 |
|---|---|---|---|---|---|---|---|
| 2-hydroxy-2-methyl propanal | CC(C)(O)C=O | CH3C(OH)(CH3)CHO | $3.0 \cdot 10^9$ | | | Acero et al. (2001)[5] | |
| 2,3-dihydroxy-2-propenal | OC=C(O)C=O | CHOCd(OH)=CdH(OH) | $9.9 \cdot 10^9$ | | | Horii et al. (1986)[2] | |
| 2,3-dihydroxy-2-propenal, conjugate base | [O-]C=C(O)C=O | | $1.6 \cdot 10^{10}$ | | | Horii et al. (1986)[2] | |
| Hydroxy acetone | CC(=O)CO | CH3COCH2(OH) | $1.2 \cdot 10^9$ | $4.0 \cdot 10^{10}$ | 1082 | Gligorovski (2005)[7] | |
| Acetoin | CC(O)C(=O)C | CH3CH(OH)COCH3 | $2.9 \cdot 10^9$ | $2.9 \cdot 10^{11}$ | 1323 | Hesper (2003) | |
| Glycolic acid | OCC(=O)O | CH2(OH)CO(OH) | $6.0 \cdot 10^8$ | | | Buxton et al. (1988) | |
| Glycolate | OCC(=O)[O-] | CH2(OH)CO(Om) | $1.2 \cdot 10^9$ | | | Ross et al. (1998) | |
| Hydrated glyoxylic acid | OC(O)C(=O)O | CH(OH)(OH)CO(OH) | $3.6 \cdot 10^8$ | $8.1 \cdot 10^9$ | 1000 | Ervens et al. (2003) | |
| Hydrated glyoxylate | OC(O)C(=O)[O-] | CH(OH)(OH)CO(Om) | $2.6 \cdot 10^9$ | $6.0 \cdot 10^{15}$ | 4330 | Ervens et al. (2003) | |
| Lactic acid | CC(O)C(=O)O | CH3CH(OH)CO(OH) | $5.24 \cdot 10^8$ | $2.28 \cdot 10^{10}$ | 1120 | Martin et al. (2009) | |
| Lactate | CC(O)C(=O)[O-] | CH3CH(OH)CO(Om) | $7.77 \cdot 10^8$ | $6.07 \cdot 10^{10}$ | 1295 | Martin et al. (2009) | |
| 2-hydroxy-butyric acid | CCC(O)C(=O)O | CH3CH2CH(OH)CO(OH) | $1.3 \cdot 10^9$ | | | Merz and Waters (1949)[1] | |

| Reactant | SMILES | GECKO-A | k298 [M$^{-1}$s$^{-1}$] | A [M$^{-1}$s$^{-1}$] | E$_a$ / R [K] | Reference/remarks n = 467 (282 aliphatic/185 aromatic) |
|---|---|---|---|---|---|---|
| Pyruvic acid | CC(=O)C(=O)O | CH3COCO(OH) | 1.2·10$^8$ | 1.0·10$^{12}$ | 2800 | Ervens et al. (2003) |
| Pyruvate | CC(=O)C(=O)[O-] | CH3COCO(Om) | 7.0·10$^8$ | 1.3·10$^{12}$ | 2300 | Ervens et al. (2003) |
| Glucuronic acid | OCC(O)C(O)C(O)C(O)C(=O)O | CH2(OH)CH(OH)CH(OH)CH(OH)CH(OH)CO(OH) | 2.2·10$^9$ | | | Average of different measurements of Phillips and Worthington (1970)[1] |
| Tartronic acid | OC(=O)C(O)C(=O)O | CO(OH)CH(OH)CO(OH) | 3.4·10$^8$ | 1.5·10$^{10}$ | 1112 | TROPOS measurements |
| Tartronate monoanion | OC(=O)C(O)C(=O)[O-] | CO(Om)CH(OH)CO(OH) | 3.6·10$^8$ | | | Schuchmann et al. (1995) |
| Tartronate dianion | [O-]C(=O)C(O)C(=O)[O-] | CO(Om)CH(OH)CO(Om) | 4.4·10$^8$ | | | Schuchmann et al. (1995) |
| Mesoxalic acid | OC(=O)C(=O)C(=O)O | CO(OH)COCO(OH) | 1.8·10$^8$ | 3.8·10$^{10}$ | 1588 | Gligorovski et al. (2009) |
| Mesoxalate monoanion | OC(=O)C(=O)C(=O)[O-] | CO(Om)COCO(OH) | 5.7·10$^7$ | | | Schuchmann et al. (1991) |
| Mesoxalate dianion | [O-]C(=O)C(=O)C(=O)[O-] | CO(Om)COCO(Om) | 1.0·10$^8$ | | | Schuchmann et al. (1991) |
| Malic acid | OC(=O)CC(O)C(=O)O | CO(OH)CH2CH(OH)CO(OH) | 3.6·10$^8$ | 7.9·10$^{10}$ | 1575 | Gligorovski et al. (2009) |
| Maleate monoanion | OC(=O)CC(O)C(=O)[O-] | CO(Om)CH2CH(OH)CO(OH) | 9.7·10$^8$ | 9.7·10$^8$ | 1701 | Gligorovski et al. (2009) |
| Maleate dianion | [O-]C(=O)CC(O)C(=O)[O-] | CO(Om)CH2CH(OH)CO(Om) | 8.5·10$^8$ | 1.2·10$^{11}$ | 1449 | Gligorovski et al. (2009) |
| Tartaric acid | OC(=O)C(O)C(O)C(=O)O | CO(OH)CH(OH)CH(OH)CO(OH) | 1.1·10$^9$ | | | Average of Scholes and Willson (1967)[2] and Moore et al. (1979)[1] |
| Tartrate dianion | [O-]C(=O)C(O)C(O)C(=O)[O-] | CO(Om)CH(OH)CH(OH)CO(Om) | 1.0·10$^8$ | | | Average of Kraljic (1967)[1,2] and Logan (1989) |
| Citric acid | OC(=O)CC(O)(C(=O)O)CC(=O)O | CO(OH)CH2C(OH)(CO(OH))CH2CO(OH) | 5.0·10$^7$ | | | Adams et al. (1965a)[2] |
| Citrate | [O-]C(=O)C... ...C(O)(C(=O)[O-])CC(=O)[O-] | CO(Om)CH2C(OH)(CO(Om))CH2CO(Om) | 1.5·10$^8$ | | | Zepp et al. (1992)[1] |
| Ethylene glycol methyl ether | COCCO | CH3-O-CH2CH2(OH) | 1.3·10$^9$ | | | Anbar et al. (1966a) |
| Ethylene glycol ethyl ether | CCOCCO | CH3CH2-O-CH2CH2(OH) | 1.7·10$^9$ | | | Anbar et al. (1966a) |
| Ethylene glycol butyl ether | CCCCOCCO | CH3CH2CH2CH2-O-CH2CH2(OH) | 5.1·10$^9$ | | | Stemmler and von Gunten (2000) |
| Diethylene glycol | OCCOCCO | CH2(OH)CH2-O-CH2CH2(OH) | 2.1·10$^9$ | | | Anbar et al. (1966a) |
| di(ethylene glycol) vinyl ether | C=COCCOCCO | CH2(OH)CH2-O-CH2CH2-O-CdH=CH2 | 4.2·10$^{10}$ | | | Moise et al. (2005) |
| Methoxyacetate | COCC(=O)[O-] | CH3-O-CH2CO(Om) | 6.1·10$^8$ | | | Anbar et al. (1966a) |
| 2-methyl-2-methoxy propanol | OCC(C)(C)OC | CH3-O-C(CH3)(CH3)CH2(OH) | 8.02·10$^8$ | | | Mezyk et al. (2009) |
| 2-methyl-2-methoxy propanal | O=CC(C)(C)OC | CH3-O-C(CH3)(CH3)CHO | 3.5·10$^9$ | | | Mezyk et al. (2009) |
| 2-methyl-2-methoxy propanoic acid | O=C(O)C(C)(C)OC | CH3-O-C(CH3)(CH3)CO(OH) | 7.73·10$^8$ | | | Mezyk et al. (2009) |
| 2-hydroxyethyl acetate | OCCOC(=O)C | CH3CO-O-CH2CH2(OH) | 9.1·10$^8$ | | | Matsushige et al. (1975)[2] |
| Butyl hydroxyacetate | CCCCOC(=O)CO | CH3CH2CH2CH2-O-COCH2(OH) | 3.2·10$^9$ | | | Stemmler and von Gunten (2000) |
| 1-hydroxy ethyl butyrate | CCCC(=O)OCCO | CH3CH2CH2CO-O-CH2CH2(OH) | 2.1·10$^9$ | | | Stemmler and von Gunten (2000) |
| Methyl methoxy acetate | COCC(=O)OC | CH3-O-CH2CO-O-CH3 | 1.8·10$^9$ | | | Massaut et al. (1988)[2] |
| Monoethyl adipate | CCOC(=O)CCCCC(=O)O | CH3CH2-O-COCH2CH2CH2CH2CO(OH) | 3.0·10$^9$ | | | Haag and Yao (1992)[1] |
| 1-hydroxy ethyl acrylate | C=CC(=O)OCCO | CH2(OH)CH2-O-COCdH=CH2 | 5.8·10$^9$ | | | Sáfrány and Wojnárovits (1993)[1] |

| Polyols and Sugars | | | | | | n = 14 |
|---|---|---|---|---|---|---|
| Glycerol | OCC(O)CO | CH2(OH)CH(OH)CH2(OH) | 1.9·10$^9$ | 2.8·10$^{11}$ | 1479 | Hoffmann et al. (2009) |
| Erythritol | OCC(O)C(O)CO | CH2(OH)CH(OH)CH(OH)CH2(OH) | 1.9·10$^9$ | 1.3·10$^{12}$ | 1948 | Hoffmann et al. (2009) |
| Arabitol | OCC(O)C(O)C(O)CO | CH2(OH)CH(OH)CH(OH)CH(OH)CH2(OH) | 1.6·10$^9$ | 2.5·10$^{10}$ | 794 | Hoffmann et al. (2009) |
| Mannitol | OCC(O)C(O)C(O)C(O)CO | CH2(OH)CH(OH)CH(OH)CH(OH)CH(OH)CH2(OH) | 1.6·10$^9$ | 1.8·10$^{10}$ | 734 | Hoffmann et al. (2009) |
| Pentaerythritol | OCC(CO)(CO)CO | CH2(OH)C(CH2(OH))(CH2(OH))CH2(OH) | 3.3·10$^9$ | | | Anbar et al. (1966a) |
| Arabinose | OCC(O)C(O)C(O)C=O | CH2(OH)CH(OH)CH(OH)CH(OH)CHO | 1.8·10$^9$ | | | Edwards et al. (1979)[2] |
| D-fructose | OCC1C(O)C(O)C(CO)(O)O1/ OCC(O)C(O)C(O)C(=O)CO | C1H(OH)CH(OH)... ...C(OH)(CH2(OH))-O-C1HCH2(OH)/ CH2(OH)CH(OH)CH(OH)CH(OH)COCH2(OH) | 1.6·10$^9$ | | | Moore et al. (1979)[2] |
| Ribose | O1C(O)C(O)C(O)C1CO/ OCC(O)C(O)C(O)C=O | C1H(OH)CH(OH)CH(OH)-O-C1HCH2(OH)/ CH2(OH)CH(OH)CH(OH)CH(OH)CHO | 1.5·10$^9$ | | | Buxton et al. (1988) |
| 2-Deoxy-D-ribose | O1C(O)CC(O)C1CO/ OCC(O)C(O)CC=O | C1H(OH)CH2CH(OH)-O-C1HCH2(OH)/ CH2(OH)CH(OH)CH(OH)CH2CHO | 2.5·10$^9$ | | | Baker et al. (1982)[2] |
| Inositol | C1(O)C(O)C(O)C(O)C(O)C1(O) | C1H(OH)CH(OH)CH(OH)CH(OH)CH(OH)C1H(OH) | 1.7·10$^9$ | | | Buxton et al. (1988) |
| Levoglucosan | OC1C(O)C(O)C2OC1OC2 | C1H2-O-CH(-O-C12H)CH(OH)CH(OH)C2H(OH) | 2.2·10$^9$ | 1.0·10$^{11}$ | 1100 | TROPOS measurements |
| Glucose | O1C(O)C(O)C(O)C(O)C1CO/ OCC(O)C(O)C(O)C(O)C=O | C1H(OH)CH(OH)CH(OH)CH(OH)-O-C1HCH2(OH)/ CH2(OH)CH(OH)CH(OH)CH(OH)CH(OH)CHO | 1.5·10$^9$ | | | Buxton et al. (1988) |
| Galactose | O1C(O)C(O)C(O)C(O)C1CO/ OCC(O)C(O)C(O)C(O)C=O | C1H(OH)CH(OH)CH(OH)CH(OH)-O-C1HCH2(OH)/ CH2(OH)CH(OH)CH(OH)CH(OH)CH(OH)CHO | 2.0·10$^9$ | | | Bucknall et al. (1978)[2] |
| 2-Deoxy-D-glucose | O1C(O)CC(O)C(O)C1CO/ OCC(O)C(O)C(O)CC=O | C1H(OH)CH(OH)CH2CH(OH)-O-C1HCH2(OH)/ CH2(OH)CH(OH)CH(OH)CH(OH)CH2CHO | 2.8·10$^9$ | | | Baker et al. (1982)[2] |

Formatted Table

| Reactant | SMILES | GECKO-A | $k298$ [M$^{-1}$s$^{-1}$] | A [M$^{-1}$s$^{-1}$] | $E_a$ / R [K] | Reference/remarks | n = 467 (282 aliphatic/185 aromatic) |
|---|---|---|---|---|---|---|---|
| **Cyclic aliphatic compounds** | | | | | | | n = 27 |
| Cyclobutane carboxylate | C1CCC1C(=O)[O-] | C1H2CH2CH2C1HCO(Om) | 3.0·10$^9$ | | | Anbar et al. (1966a) | |
| Cyclopentane | C1CCCC1 | C1H2CH2CH2CH2C1H2 | 3.7·10$^9$ | | | Buxton et al. (1988) | |
| Methyl cyclopentane | C1CCCC1C | C1H2CH2CH2CH2C1HCH3 | 7.0·10$^9$ | | | Rudakov et al. (1981)[2] | |
| Cyclopentane carboxylate | C1CCCC1C(=O)[O-] | C1H2CH2CH2CH2C1HCO(Om) | 4.2·10$^9$ | | | Anbar et al. (1966a) | |
| Cyclohexane | C1CCCCC1 | C1H2CH2CH2CH2CH2C1H2 | 6.1·10$^9$ | | | Rudakov et al. (1981)[2] | |
| Methyl cyclohexane | C1CCCCC1C | C1H2CH2CH2CH2CH2C1HCH3 | 7.1·10$^9$ | | | Rudakov et al. (1981)[2] | |
| Cyclohexane carboxylate | C1CCCCC1C(=O)[O-] | C1H2CH2CH2CH2CH2C1HCO(Om) | 5.5·10$^9$ | | | Anbar et al. (1966a) | |
| Cycloheptane | C1CCCCCC1 | C1H2CH2CH2CH2CH2CH2C1H2 | 7.7·10$^9$ | | | Rudakov et al. (1981)[2] | |
| Cyclopentene | C1CCC=C1 | C1H2CH2CH2CdH=CdHC1H2 | 7.0·10$^9$ | | | Soylemez and Schuler (1974)[2] | |
| Cyclohexene | C1CCC=C1 | C1H2CH2CH2CdH=CdHCH2C1H2 | 8.8·10$^9$ | | | Michael and Hart (1970)[2] | |
| 1,3-cyclohexadiene | C1CC=CC=C1 | C1H2CdH=CdHCdH=CdHC1H2 | 9.9·10$^9$ | | | Michael and Hart (1970)[2] | |
| 1,4-cyclohexadiene | C1C=CCC=C1 | C1H2CdH=CdHCH2CdH=Cd1H | 7.7·10$^9$ | | | Michael and Hart (1970)[2] | |
| cycloheptatriene | C1C=CC=CC=C1 | C1H2CdH=CdHCdH=CdHCdH=Cd1H | 1.0·10$^{10}$ | | | Schöneshöfer (1971)[2] | |
| Quinine | O=C1C=CC(=O)C=C1 | | 1.2·10$^9$ | | | Adams and Michael (1967)[2] | |
| Oxirane | C1OC1 | C1H2-O-C1H2 | 6.8·10$^7$ | | | Anbar et al. (1966a) | |
| Methyl oxirane | C1OC1C | C1H2-O-C1HCH3 | 2.5·10$^8$ | | | Anbar et al. (1966a) | |
| Ethyl oxirane | C1OC1CC | C1H2-O-C1HCH2CH3 | 7.8·10$^8$ | | | Anbar et al. (1966a) | |
| Glycidol | C1OC1CO | C1H2-O-C1HCH2(OH) | 4.7·10$^8$ | | | Anbar et al. (1966a) | |
| Tetrahydrofuran | C1COCC1 | C1H2CH2-O-CH2C1H2 | 3.8·10$^9$ | | | George et al. (2003) | |
| Dioxolane | C1OCOC1 | C1H2-O-CH2-O-C1H2 | 4.0·10$^9$ | | | Eibenberger (1980)[2] | |
| 2-methyl dioxolane | C1OC(C)OC1 | C1H2-O-CH(CH3)-O-C1H2 | 3.5·10$^9$ | | | Eibenberger (1980)[2] | |
| 2,2-dimethyl dioxolane | C1OC(C)(C)OC1 | C1H2-O-C(CH3)(CH3)-O-C1H2 | 2.1·10$^9$ | | | Eibenberger (1980)[2] | |
| Tetrahydropyran | C1CCCCO1 | C1H2CH2CH2-O-CH2C1H2 | 1.5·10$^9$ | | | Walling et al. (1974) | |
| 1,3-dioxane | C1COCOC1 | C1H2CH2-O-CH2-O-C1H2 | 4.0·10$^9$ | | | Eibenberger (1980)[2] | |
| 1,4-dioxane | C1OCCOC1 | C1H2-O-CH2CH2-O-C1H2 | 2.8·10$^9$ | | | Buxton et al. (1988) | |
| Trioxane | C1OCOCO1 | C1H2-O-CH2-O-CH2-O1- | 1.0·10$^9$ | | | Average of Anbar et al. (1966a) and Eibenberger (1980)[2] | |
| paraldehyde | CC1OC(C)OC(C)O1 | -O1-CH(CH3)-O-CH(CH3)-O-C1HCH3 | 1.6·10$^9$ | | | Average of Anbar et al. (1966a) and Eibenberger (1980)[2] | |
| **Terpenes and terpene oxidation products** | | | | | | | n = 4 |
| R(+)-limonene | CC1=CCC(CC1)C(C)=C | C1H2CH2Cd(CH3)=CdHCH2C1HCd(CH3)=CdH2 | 2.9·10$^9$ | | | TROPOS measurements | |
| α-pinene | CC2(C)C1CC=C(C)C2C1 | C12HCH2CH(C1(CH3)CH3)CH2CdH=Cd2CH3 | 1.4·10$^9$ | | | Raabe (1996) | |
| Cis-verbenol | CC2(C)C1CC2C(O)C=C1C | C12HCH2CH(C1(CH3)CH3)CH(OH)CdH=Cd2CH3 | 6.8·10$^9$ | | | Buxton et al. (2000)[3] | |
| (-)-Myrtenal | CC2(C)C1CC=C(C=O)C2C1 | CH3C1(CH3)CH(CH2C12H)CH2CdH=Cd2CHO | 1.7·10$^9$ | 1.0·10$^{13}$ | 2600 | TROPOS measurements | |
| **Unclassified aliphatic compounds** | | | | | | | n = 2 |
| Disuccinyl peroxide | [O-]C(=O)CCC(=O)OO… …C(=O)CCC(=O)[O-] | | 8.0·10$^7$ | | | Graedel and Weschler (1981)[1] | |
| Diacetyl peroxide | CC(=O)OOC(=O)C | | 6.0·10$^7$ | | | Graedel and Weschler (1981)[1] | |
| **Furans (including ascorbic acid)** | | | | | | | n = 21 |
| Furan | c1cocc1 | -O1-CdH=CdHCdH=Cd1H | 3.9·10$^9$ | | | Lilie (1971)[2] | |
| 2-methylfuran | c1coc(C)c1 | -O1-CdH=CdHCdH=Cd1CH3 | 1.9·10$^{10}$ | | | Vysotskaya et al. (1983)[2] | |
| Furfuryl alcohol | c1coc(CO)c1 | -O1-CdH=CdHCdH=Cd1CH2(OH) | 1.5·10$^{10}$ | | | Saveleva et al. (1973); Vysotskaya et al. (1983)[2] | |
| Furfural | c1coc(C=O)c1 | -O1-CdH=CdHCdH=Cd1CHO | 6.3·10$^9$ | | | Average of Vysotskaya et al. (1983)[2] and D'Angelantonio et al. (1999) | |
| 5-methyl furfural | c1c(C)oc(C=O)c1 | CH3Cd1=CdHCdH=Cd(-O1-)CHO | 7.2·10$^9$ | | | Vysotskaya et al. (1983)[2] | |
| 5-hydroxymethylfurfural | c1c(CO)oc(C=O)c1 | CH2(OH)Cd1=CdHCdH=Cd(-O1-)CHO | 5.8·10$^9$ | | | Vysotskaya et al. (1983)[2] | |
| 5-nitro furfural | c1c(N(=O)=O)oc(C=O)c1 | -O1-Cd(NO2)=CdHCdH=Cd1CHO | 5.5·10$^9$ | | | Greenstock and Dunlop (1973); Chapman et al. (1973)[2] | |
| Acetylfuran | c1coc(C(=O)C)c1 | -O1-CdH=CdHCdH=Cd1COCH3 | 4.5·10$^9$ | | | Vysotskaya et al. (1983)[1] | |
| Furan-2-carboxylate | c1coc(C(=O)[O-])c1 | C1HcHcH-O-c1CO(Om) | 1.2·10$^{10}$ | | | Saveleva et al. (1973)[2] | |
| 5-carboxylate furfural | c1c(C=O)oc(C(=O)[O-])c1 | CO(Om)c1-O-c(CHO)cHc1H | 3.8·10$^9$ | | | Vysotskaya et al. (1983)[2] | |
| 2-carboxylate-5-nitro-furan | c1c(N(=O)=O)oc(C(=O)[O-])c1 | C1HcHc(NO2)-O-c1CO(Om) | 5.3·10$^9$ | | | Greenstock and Dunlop (1973); Chapman et al. (1973)[2] | |
| Tetronate | O=C1C=C([O-])CO1 | | 9.2·10$^9$ | | | Schuler et al. (1974)[1] | |
| α-Hydroxytetronate | O=C1C([O-])=C(O)CO1 | | 4.7·10$^9$ | | | Schuler et al. (1974)[1] | |
| Ascorbic acid | O1C(=O)C(O)=C(O)C1C(O)CO | -O1-COCd(OH)=Cd(OH)C1HCH(OH)CH2(OH) | 1.0·10$^{10}$ | | | Buxton et al. (1988) | |
| Ascorbate | O1C(=O)C([O-])=C(O)C1C(O)CO | | 3.4·10$^9$ | | | Average of Schuler (1977)[2], Redpath and Willson (1973)[2], Schöneshöfer (1972)[2], Bonifačić et al. (1994)[1], Bonifačić et al. (1994)[1], and Ye and Schuler (1990)[1] | |
| L-ascorbate-2-sulfate | O1C(=O)… …C(OS(=O)(=O)[O-])=C(O)C1C(O)CO | | 4.2·10$^9$ | | | Cabelli et al. (1983)[2] | |

| Reactant | SMILES | GECKO-A | $k298$ [M$^{-1}$s$^{-1}$] | A [M$^{-1}$s$^{-1}$] | $E_a$ / R [K] | Reference/remarks | n = 467 (282 aliphatic/185 aromatic) |
|---|---|---|---|---|---|---|---|
| 1-O-methyl ascorbic acid | O1C(=O)C(OC)=C(O)C1C(O)CO | -O1-COCd(-O-CH3)=Cd(OH)C1HCH(OH)CH2(OH) | 2.5·10[9] | | | Cabelli et al. (1983)[2] | |
| 2-O-methyl ascorbic acid | O1C(=O)C(O)=C(OC)C1C(O)CO | CH3-O-Cd1=Cd(OH)CO-O-C1HCH(OH)CH2(OH) | 2.7·10[9] | | | Cabelli et al. (1983)[2] | |
| 3-O-methyl ascorbic acid | O1C(=O)C(=O)C(OC)C1C(OC)CO | -O1-COCd(OH)=Cd(OH)C1HCH(CH2(OH))-O-CH3 | 3.0·10[9] | | | Cabelli et al. (1983)[2] | |
| 3-O-methyl ascorbate | O1C(=O)C([O-])=C(O)C1C(OC)CO | | 4.8·10[9] | | | Cabelli et al. (1983)[2] | |
| O-dimethyl ascorbic acid | O1C(=O)C(OC)=C(OC)C1C(O)CO | CH3-O-Cd1=Cd(-O-CH3)CO-O-C1HCH(OH)CH2(OH) | 4.2·10[9] | | | Cabelli et al. (1983)[2] | |

| Benzols | | | | | | | n = 17 |
|---|---|---|---|---|---|---|---|
| Benzol | c1ccccc1 | c1HcHcHcHcHc1H | 7.8·10[9] | | | Buxton et al. (1988) | |
| Toluol | c1ccccc1C | c1HcHcHcHcHc1CH3 | 8.1·10[9] | | | Albarran et al. (2003); branching rations o:m:p = 0.84:0.41:1 (i<0.2) | |
| Ethyl benzol | c1ccccc1CC | c1HcHcHcHcHc1CH2CH3 | 7.5·10[9] | | | Sehested and Holcman (1979)[2] | |
| o-xylene | Cc1ccccc1C | CH3c1c1cHcHcHcHc1CH3 | 6.7·10[9] | | | Sehested et al. (1975)[2] | |
| m-xylene | c1c(C)cccc1C | c1Hc(CH3)cHcHcHc1CH3 | 7.5·10[9] | | | Sehested et al. (1975)[2] | |
| p-xylene | c1cc(C)ccc1C | c1Hc Hc(CH3)cHcHc1CH3 | 7.0·10[9] | | | Sehested et al. (1975)[2] | |
| 1,2,3-trimethyl benzol | Cc1cccc(C)c1C | CH3c1c1cHcHcHc(CH3)c1CH3 | 7.0·10[9] | | | Sehested et al. (1975)[2] | |
| 1,2,4-trimethyl benzol | Cc1ccc(C)cc1C | CH3c1cHc(CH3)cHc1CH3 | 6.2·10[9] | | | Sehested et al. (1975)[2] | |
| Mesitylene | c1c(C)cc(C)cc1C | c1Hc(CH3)cHc(CH3)cHc1CH3 | 6.4·10[9] | | | Sehested et al. (1975)[2] | |
| 1,2,3,4-tetramethyl benzol | Cc1cc(C)c(C)c1C | CH3c1c(CH3)cHcHc(CH3)c1CH3 | 7.2·10[9] | | | Sehested et al. (1975)[1] | |
| 1,2,3,5-tetramethyl benzol | Cc1cc(C)cc(C)c1C | CH3c1cc(CH3)cHc(CH3)c1CH3 | 7.1·10[9] | | | Sehested et al. (1975)[1] | |
| 1,2,4,5-tetramethyl benzol | Cc1cc(C)c(C)cc1C | CH3c1cHc(CH3)c(CH3)cHc1CH3 | 7.0·10[9] | | | Sehested et al. (1975)[1] | |
| Pentamethyl benzol | Cc1cc(C)c(C)c(C)c1C | CH3c1c(CH3)cHc(CH3)c(CH3)c1CH3 | 7.5·10[9] | | | Sehested et al. (1975)[2] | |
| Hexamethyl benzol | Cc1c(C)c(C)c(C)c(C)c1C | CH3c1c(CH3)c(CH3)c(CH3)c(CH3)c1CH3 | 7.2·10[9] | | | Sehested et al. (1975)[2] | |
| Cumene | c1ccccc1C(C)C | c1HcHcHcHcHc1CH(CH3)CH3 | 7.5·10[9] | | | Sehested and Holcman (1979)[2] | |
| Styrene | c1ccccc1C=C | c1HcHcHcHcHc1CdH=CdH2 | 6.0·10[9] | | | Brede et al. (1977)[2] | |
| α-methylstyrene | c1ccccc1C(C)=C | c1HcHcHcHcHc1Cd(CH3)=CdH2 | 9.7·10[9] | | | Brede et al. (1977)[2] | |

| N/S-substituted benzols | | | | | | | n = 7 |
|---|---|---|---|---|---|---|---|
| Nitrosobenzol | c1ccccc1N=O | c1HcHcHc(NO)cHc1H | 1.8·10[10] | | | Asmus et al. (1966)[2] | |
| Nitrobenzol | c1ccccc1N(=O)=O | c1HcHcHc(NO2)cHc1H | 3.9·10[9] | | | Buxton et al. (1988) | |
| Benzolsulfonic acid | c1ccccc1S(=O)(=O)O | | 1.6·10[9] | | | Merz and Waters (1949)[2] | |
| Phenyl sulfonate | c1ccccc1S(=O)(=O)[O-] | | 3.9·10[9] | | | Average of Anbar et al. (1966b) and Neta and Dorfman (1968)[2] | |
| 2,4-dinitro-toluol | c1c(N(=O)=O)cc(N(=O)=O)cc1C | c1Hc(NO2)cHc(NO2)cHc1CH3 | 9.0·10[8] | | | Makarov et al. (2008) | |
| 2,4,6-trinitro-toluol | c1c(N(=O)=O)cc(N(=O)=O)cc1 ...(N(=O)=O) | c1Hc(NO2)cHc(NO2)cHc1(NO2) | 7.4·10[8] | | | Makarov et al. (2008) | |
| 1-methyl-3-nitro phenyl sulfonate | c1(C)cc(N(=O)=O)ccc1S(=O)(=O)[O-] | | 1.6·10[9] | | | Basinski and Lerke (1972)[2] | |

| Phenols and other aromatic alcohols | | | | | | | n = 25 |
|---|---|---|---|---|---|---|---|
| Phenol | c1ccccc1O | c1HcHcHc(OH)cHc1H | 1.01·10[10] | | | Chen and Schuler (1993); partial $k$ @ o/m/p position: 2.37·10[9]/1.3·10[8]/3.56·10[9] | |
| Phenolate | c1ccccc1[O-] | | 9.6·10[9] | | | Bonin et al. (2007) | |
| Catechol | c1cccc(O)c1O | c1HcHc(OH)c(OH)cHc1H | 1.1·10[10] | | | Saveleva et al. (1972)[2] | |
| Resorcinol | c1ccc(O)cc1O | c1HcHc(OH)cHc(OH)c1H | 1.2·10[10] | | | Saveleva et al. (1972)[2] | |
| Hydroquinone | c1cc(O)ccc1O | c1Hc(OH)cHcHc(OH)c1H | 1.2·10[10] | | | Average of Adams et al. (1965a)[2]; *Al-Suhybani and Hughes*, 1986[2]; and *Heckel et al.*, 1966[2] | |
| phloroglucinol | c1c(O)cc(O)cc1O | c1Hc(OH)cHc(OH)cHc1(OH) | 1.0·10[10] | | | *Wang et al.*, 1994[1] | |
| p-tert-butyl phenol | c1cc(O)ccc1C(C)(C)C | c1HcHc(OH)cHcHc1C(CH3)(CH3)CH3 | 1.9·10[10] | | | Saveleva et al. (1972) | |
| 4-tert-butyl catechol | c1c(O)c(O)ccc1C(C)(C)C | c1Hc(OH)c(OH)cHcHc1C(CH3)(CH3)CH3 | 7.6·10[9] | | | *Richter*, 1979[2] | |
| 4-(1,2-dihydroxyethyl)-catechol | c1c(O)c(O)ccc1C(O)CO | c1Hc(OH)c(OH)cHcHc1CH(OH)CH2(OH) | 1.6·10[10] | | | Average of different measurements of *Ek et al.*, 1989 (8) | |
| Benzyl alcohol | c1ccccc1CO | c1HcHcHcHcHc1CH2(OH) | 8.4·10[9] | | | Neta and Dorfman (1968)[2] | |
| Phenetyl alcohol | c1ccccc1CCO | c1HcHcHcHcHc1CH2CH2(OH) | 6.4·10[9] | | | Buxton et al. (1988) | |
| o-cresol | c1(O)ccccc1C | c1HcHcHcHc(OH)c1CH3 | 1.1·10[10] | | | Saveleva et al. (1972)[1] | |
| p-cresol | c1cc(O)ccc1C | c1HcHc(OH)cHcHc1CH3 | 1.2·10[10] | | | Fisher and Hamill (1973) | |
| 2,3-dihydroxy toluol | c1(O)c(O)cccc1C | c1HcHcHc(OH)c(OH)c1CH3 | 1.6·10[10] | | | Gohn and Getoff (1977)[1] | |
| 3,4-dihydroxy toluol | c1cc(O)c(O)cc1C | c1HcHc(OH)c(OH)cHc1CH3 | 1.6·10[10] | | | Gohn and Getoff (1977)[2] | |
| 1-phenylethanol | c1ccccc1C(O)C | c1HcHcHcHcHc1CH(OH)CH3 | 1.1·10[10] | | | Snook and Hamilton (1974) | |
| 1-(p-ethylphenyl)ethanol | c1cc(CC)ccc1C(O)C | c1HcHc(CH(OH)CH3)cHcHc1CH2CH3 | 1.3·10[10] | | | Snook and Hamilton (1974) | |
| 1-phenyl propanol | c1ccccc1C(O)CC | c1HcHcHcHcHc1CH(OH)CH2CH3 | 1.0·10[10] | | | Snook and Hamilton (1974) | |
| 1-phenyl-2-methyl propanol | c1ccccc1C(O)C(C)C | c1HcHcHcHcHc1CH(OH)CH(CH3)CH3 | 9.5·10[9] | | | Snook and Hamilton (1974) | |
| 1-phenyl-2,2-dimethyl propanol | c1ccccc1C(O)C(C)(C)C | c1HcHcHcHcHc1CH(OH)C(CH3)(CH3)CH3 | 9.9·10[9] | | | Snook and Hamilton (1974) | |

| Reactant | SMILES | GECKO-A | k298 [M$^{-1}$s$^{-1}$] | A [M$^{-1}$s$^{-1}$] | E$_a$ / R [K] | Reference/remarks | n = 467 (282 aliphatic/185 aromatic) |
|---|---|---|---|---|---|---|---|
| 1-phenyl-2-propanol | c1ccccc1CC(O)C | c1HcHcHcHcHc1CH2CH(OH)CH3 | 2.1·10$^{10}$ | | | Snook and Hamilton (1974) | |
| 2-phenyl-2-propanol | c1ccccc1C(C)(O)C | c1HcHcHcHcHc1C(OH)(CH3)CH3 | 4.6·10$^9$ | | | Snook and Hamilton (1974) | |
| Phenyl tert-butyl alcohol | c1ccccc1CC(C)(O)C | c1HcHcHcHcHc1CH2C(OH)(CH3)CH3 | 1.7·10$^{10}$ | | | Snook and Hamilton (1974) | |
| 4-phenyl-2-butanol | c1ccccc1CCC(O)C | c1HcHcHcHcHc1CH2CH2CH(OH)CH3 | 2.0·10$^{10}$ | | | Snook and Hamilton (1974) | |
| 1,1-dimethyl-3-phenyl butanol | c1ccccc1CCC(C)(O)C | c1HcHcHcHcHc1CH2CH2C(OH)(CH3)CH3 | 5.9·10$^9$ | | | Snook and Hamilton (1974) | |

| Aromatic carbonyls, acids and diacids | | | | | | | n = 8 |
|---|---|---|---|---|---|---|---|
| Benzaldehyde | c1ccccc1C=O | c1HcHcHcHcHc1CHO | 3.5·10$^9$ | | | Average of Geeta et al. (2001) and Shevchuk et al. (1969)[2] | |
| Phenyl acetone | c1ccccc1C(=O)C | c1HcHcHcHcHc1COCH3 | 5.3·10$^9$ | | | Average of Geeta et al. (2001), Anbar et al. (1966b); Willson et al. (1971)[1]; and Neta and Dorfman (1968)[1] | |
| Benzoic acid | c1ccccc1C(=O)O | c1HcHcHcHcHc1CO(OH) | 3.1·10$^9$ | | | Average of Wander et al. (1968)[2] and Ashton et al. (1995)[1] | |
| Benzoate | c1ccccc1C(=O)[O-] | c1HcHcHcHcHc1CO(Om) | 5.9·10$^9$ | | | Buxton et al. (1988) | |
| o-phthalate | c1(C(=O)[O-])ccccc1C(=O)[O-] | CO(Om)c1cHcHcHcHc1CO(Om) | 5.9·10$^9$ | | | Simic et al. (1973a)[2] | |
| p-phthalate | c1cc(C(=O)[O-])ccc1C(=O)[O-] | c1HcHc(CO(Om))cHcHc1CO(Om) | 3.3·10$^9$ | | | Anbar et al. (1966b) | |
| Phenyl acetate | c1ccccc1CC(=O)[O-] | c1HcHcHcHcHc1CH2CO(Om) | 7.9·10$^9$ | | | Neta and Dorfman (1968)[2] | |
| Cinnamate | c1ccccc1C=CC(=O)[O-] | c1HcHcHcHcHc1CdH=CdHCO(Om) | 8.1·10$^9$ | | | Bobrowski and Raghavan (1982)[2] | |

| Anisoles, aromatic ethers and esters | | | | | | | n = 9 |
|---|---|---|---|---|---|---|---|
| Anisole | c1ccccc1OC | c1HcHcHcHcHc1-O-CH3 | 5.8·10$^9$ | | | Average of Anbar et al. (1966b) and O'Neill et al. (1975)[2] | |
| o-dimethoxy benzol | COc1ccccc1OC | CH3-O-c1cHcHcHcHc1-O-CH3 | 5.2·10$^9$ | | | O'Neill et al. (1975)[2] | |
| m-dimethoxy benzol | c1c(OC)cccc1OC | CH3-O-c(c1H)cHcHcHc1-O-CH3 | 7.2·10$^9$ | | | O'Neill et al. (1975)[2] | |
| p-dimethoxy benzol | c1cc(OC)ccc1OC | c1HcHc(-O-CH3)cHcHc1-O-CH3 | 7.0·10$^9$ | | | O'Neill et al. (1975)[2] | |
| 1,2,3-trimethoxy benzol | COc1cccc(OC)c1OC | CH3-O-c1cHcHcHc(-O-CH3)c1-O-CH3 | 8.0·10$^9$ | | | O'Neill et al. (1975)[2] | |
| 1,2,4-trimethoxy benzol | COc1cc(OC)ccc1OC | CH3-O-c1cHcHc(-O-CH3)cHc1-O-CH3 | 8.1·10$^9$ | | | O'Neill et al. (1975)[2] | |
| benzyl methyl ether | c1ccccc1COC | c1HcHcHcHcHc1CH2-O-CH3 | 1.0·10$^{10}$ | | | Snook and Hamilton (1974) | |
| 1-methoxy-2-methyl-1-phenylpropane | c1ccccc1C(OC)C(C)C | c1HcHcHcHcHc1CH(-O-CH3)CH(CH3)CH3 | 7.4·10$^9$ | | | Snook and Hamilton (1974) | |
| Phenyl acetate | c1ccccc1OC(=O)C | c1HcHcHcHcHc1-O-COCH3 | 5.2·10$^9$ | | | Anbar et al. (1966b) | |

| Polyfunctional aromatic compounds | | | | | | | n = 64 |
|---|---|---|---|---|---|---|---|
| o-nitro phenol | Oc1ccccc1N(=O)=O | c1HcHc(OH)c(NO2)cHc1H | 9.2·10$^9$ | | | Saveleva et al. (1972)[9] | |
| o-nitro phenolate | [O-]c1ccccc1N(=O)=O | | 9.2·10$^9$ | | | Saveleva et al. (1972)[2] | |
| m-nitro phenolate | c1c([O-])cccc1N(=O)=O | | 7.1·10$^9$ | | | Saveleva et al. (1972)[2] | |
| p-nitro phenol | c1cc(O)ccc1N(=O)=O | c1Hc(OH)cHcHc(NO2)c1H | 3.8·10$^9$ | | | Cercek and Ebert (1968)[2] | |
| p-nitro phenolate | c1cc([O-])ccc1N(=O)=O | | 7.6·10$^9$ | | | Saveleva et al. (1972)[2] | |
| 2-hydroxy benzaldehyde | c1(O)ccccc1C=O | c1HcHcHcHc(OH)c1CHO | 5.2·10$^9$ | | | Geeta et al. (2001) | |
| 2-hydroxy benzaldehyde, conjugate base | c1([O-])ccccc1C=O | | 5.2·10$^9$ | | | Saveleva et al. (1972)[2] | |
| 3-hydroxy benzaldehyde | c1c(O)cccc1C=O | c1HcHcHc(OH)cHc1CHO | 7.7·10$^9$ | | | Geeta et al. (2001) | |
| 4-hydroxy benzaldehyde | c1cc(O)ccc1C=O | c1HcHc(OH)cHcHc1CHO | 1.21·10$^{10}$ | | | Geeta et al. (2001) | |
| 4-hydroxy benzaldehyde, conjugate base | c1cc([O-])ccc1C=O | | 1.0·10$^{10}$ | | | Saveleva et al. (1972)[2] | |
| 3,4-dihydroxy benzaldehyde | c1c(O)c(O)ccc1C=O | c1HcHc(OH)c(OH)cHc1CHO | 8.3·10$^9$ | | | Bors et al. (1982) | |
| 2-hydroxyphenyl acetone | c1(O)ccccc1C(=O)C | c1HcHcHcHc(OH)c1COCH3 | 2.7·10$^9$ | | | Geeta et al. (2001) | |
| 3-hydroxyphenyl acetone | c1c(O)cccc1C(=O)C | c1HcHcHc(OH)cHc1COCH3 | 2.6·10$^9$ | | | Geeta et al. (2001) | |
| 4-hydroxyphenyl acetone | c1cc(O)ccc1C(=O)C | c1HcHc(OH)cHcHc1COCH3 | 5.1·10$^9$ | | | Geeta et al. (2001) | |
| 3,4-dihydroxyphenyl acetone | c1c(O)c(O)ccc1C(=O)C | c1HcHc(OH)c(OH)cHc1COCH3 | 1.0·10$^{10}$ | | | Bors et al. (1982)[1] | |
| 2,4-dihydroxyphenyl acetone | c1(O)cc(O)ccc1C(=O)C | c1HcHc(OH)cHc(OH)c1COCH3 | 3.0·10$^{10}$ | | | Bors et al. (1982)[1] | |
| 2,5-dihydroxyphenyl acetone | c1(O)ccc(O)cc1C(=O)C | c1Hc(OH)cHcHc(OH)c1COCH3 | 8.0·10$^9$ | | | Bors et al. (1982)[1] | |
| 2,6-dihydroxyphenyl acetone | c1(O)cccc(O)c1C(=O)C | c1HcHc(OH)c(c(OH)c1H)COCH3 | 8.0·10$^9$ | | | Bors et al. (1982)[2] | |
| 2-hydroxy benzoic acid | c1(O)ccccc1C(=O)O | c1HcHcHcHc(OH)c1CO(OH) | 2.2·10$^{10}$ | | | Buxton et al. (1988) | |
| 2-hydroxy benzoate | c1(O)ccccc1C(=O)[O-] | | 1.6·10$^{10}$ | | | Buxton et al. (1988) | |
| 4-hydroxy benzoic acid | c1cc(O)ccc1C(=O)O | c1HcHc(OH)cHcHc1CO(OH) | 7.7·10$^9$ | | | Average of Neta and Dorfman (1968)[1]; Anderson et al. (1987)[1]; and Shetiya et al. (1976)[1] | |
| 4-hydroxy benzoate | c1cc(O)ccc1C(=O)[O-] | c1HcHc(OH)cHcHc1CO(Om) | 8.5·10$^9$ | | | Buxton et al. (1988) | |
| 2,3-hydroxy benzoate | c1(O)c(O)cccc1C(=O)[O-] | c1HcHcHc(OH)c(O)c1CO(Om) | 1.0·10$^{10}$ | | | Oturan et al. (1992)[1] | |
| 2,4-hydroxy benzoate | c1(O)cc(O)ccc1C(=O)[O-] | c1HcHc(OH)cHc(O)c1CO(Om) | 1.6·10$^{10}$ | | | Oturan et al. (1992)[1] | |
| 2,5-hydroxy benzoate | c1(O)ccc(O)cc1C(=O)[O-] | c1Hc(OH)cHcHc(O)c1CO(Om) | 1.7·10$^{10}$ | | | Oturan et al. (1992)[1] | |
| 2,6-hydroxy benzoate | c1(O)cccc(O)c1C(=O)[O-] | c1HcHc(OH)c(CO(Om))c(OH)c1H | 1.0·10$^{10}$ | | | Oturan et al. (1992)[1] | |
| 3,4,5-hydroxy benzoate | c1c(O)c(O)c(O)cc1C(=O)[O-] | c1Hc(OH)c(OH)c(OH)cHc1CO(Om) | 4.0·10$^{10}$ | | | Oturan et al. (1992)[1] | |

| Reactant | SMILES | GECKO-A | k298 [M⁻¹s⁻¹] | A [M⁻¹s⁻¹] | $E_a$ / R [K] | Reference/remarks $n = 467$ (282 aliphatic/185 aromatic) |
|---|---|---|---|---|---|---|
| 2,3,4-hydroxy benzoate | c1(O)c(O)c(O)ccc1C(=O)[O-] | c1HcHc(OH)c(OH)c(OH)c1CO(Om) | $1.9 \cdot 10^{10}$ | | | Oturan et al. (1992)[1] |
| 2,4,6-hydroxy benzoate | c1(O)cc(O)cc(O)c1C(=O)[O-] | c1(OH)cHc(OH)cHc(OH)c1CO(Om) | $4.8 \cdot 10^{10}$ | | | Oturan et al. (1992)[1] |
| 4-methyl benzoate | c1cc(C)ccc1C(=O)[O-] | CH3c1cHcHc(CO(Om))cHc1H | $8.0 \cdot 10^{9}$ | | | Neta et al. (1972)[1] |
| 4-nitrobenzoate | c1cc(N(=O)=O)ccc1C(=O)[O-] | c1HcHc(NO2)cHcHc1CO(Om) | $2.6 \cdot 10^{9}$ | | | Neta and Dorfman (1968)[2] Buxton et al. (1988) |
| (3-(4-hydroxyphenyl) propionate monoanion) | c1cc(O)ccc1CCC(=O)[O-] | c1HcHc(OH)cHcHc1CH2CH2CO(Om) | $1.2 \cdot 10^{10}$ | | | Buxton et al. (1988) |
| (3-(4-hydroxyphenyl) propionate dianion) | c1cc([O-])ccc1CCC(=O)[O-] | | $1.9 \cdot 10^{10}$ | | | Buxton et al. (1988) |
| p-hydroxy cinnamate | c1cc(O)ccc1C=CC(=O)[O-] | c1HcHc(OH)cHcHc1CdH=CdHCO(Om) | $8.2 \cdot 10^{9}$ | | | Bobrowski (1984)[2] |
| 3,4-dihydroxy cinnamate | c1c(O)c(O)ccc1C=CC(=O)[O-] | c1HcHc(OH)c(OH)cHc1CdH=CdHCO(Om) | $2.8 \cdot 10^{10}$ | | | Bors et al. (1982)[2] |
| o- methoxy phenol | Oc1ccccc1OC | c1HcHcHcHc(OH)c1-O-CH3 | $2.0 \cdot 10^{10}$ | | | O'Neill and Steenken (1977)[2] |
| m- methoxy phenol | c1c(O)cccc1OC | c1HcHc(OH)cHc1-O-CH3 | $3.2 \cdot 10^{10}$ | | | O'Neill and Steenken (1977)[2] |
| p- methoxy phenol | c1cc(O)ccc1OC | c1HcHc(OH)cHcHc1-O-CH3 | $2.6 \cdot 10^{10}$ | | | O'Neill and Steenken (1977)[9] |
| 2,3-dimethoxy phenol | COc1c(OC)cccc1O | CH3-O-c1cHcHcHc(OH)c1-O-CH3 | $2.0 \cdot 10^{10}$ | | | O'Neill and Steenken (1977)[2] |
| 2,6-dimethoxy phenol | c1(OC)cccc(OC)c1O | c1HcHcHc(-O-CH3)c(OH)c1-O-CH3 | $2.6 \cdot 10^{10}$ | | | O'Neill and Steenken (1977)[2] |
| 3,5-dimethoxy phenol | c1c(OC)cc(OC)cc1O | CH3-O-c(c1H)cHc(OH)cHc1-O-CH3 | $2.0 \cdot 10^{10}$ | | | O'Neill and Steenken (1977)[2] |
| 2-methoxy benzoate | c1(OC)ccccc1C(=O)[O-] | CH3-O-c1cHcHcHcHc1CO(Om) | $5.4 \cdot 10^{9}$ | | | O'Neill et al. (1977)[2] |
| 3-methoxy benzoate | c1c(OC)cccc1C(=O)[O-] | CH3-O-c1cHcHcHc(c1H)CO(Om) | $6.6 \cdot 10^{9}$ | | | O'Neill et al. (1977)[2] |
| 4-methoxy benzoate | c1cc(OC)ccc1C(=O)[O-] | CH3-O-c1cHcHc(CO(Om))cHc1H | $7.2 \cdot 10^{9}$ | | | O'Neill et al. (1977)[2] |
| 2-hydroxy-5-methoxy benzoate | c1(O)ccc(OC)cc1C(=O)[O-] | CH3-O-c1cHcHc(OH)c(c1H)CO(Om) | $1.8 \cdot 10^{10}$ | $9.0 \cdot 10^{10}$ | 1323 | O'Neill et al. (1977)[2] |
| 3-methoxy-4-hydroxy benzoate | c1c(OC)c(O)ccc1C(=O)[O-] | CH3-O-c1c(OH)cHcHc(c1H)CO(Om) | $1.4 \cdot 10^{10}$ | | | O'Neill et al. (1977)[2] |
| 2,3-dimethoxy benzoate | c1(OC)c(OC)cccc1C(=O)[O-] | CH3-O-c1cHcHcHc(CO(Om))c1-O-CH3 | $1.0 \cdot 10^{10}$ | | | O'Neill et al. (1977)[2] |
| 2,4-dimethoxy benzoate | c1(OC)cc(OC)ccc1C(=O)[O-] | CH3-O-c1cHc(-O-CH3)cHcHc1CO(Om) | $1.0 \cdot 10^{10}$ | | | O'Neill et al. (1977)[2] |
| 3,4-dimethoxy benzoate | c1(OC)c(OC)ccc1C(=O)[O-] | CH3-O-c1cHcHc(CO(Om))cHc1-O-CH3 | $1.2 \cdot 10^{10}$ | | | O'Neill et al. (1977)[2] |
| 2,6-dimethoxy benzoate | c1(OC)cccc(OC)c1C(=O)[O-] | CH3-O-c1cHcHcHc(-O-CH3)c1CO(Om) | $6.6 \cdot 10^{9}$ | | | O'Neill et al. (1977)[2] |
| 3,5-dimethoxy benzoate | c1c(OC)cc(OC)c1C(=O)[O-] | CH3-O-c(c1H)cHc(CO(Om))cHc1-O-CH3 | $7.0 \cdot 10^{9}$ | | | O'Neill et al. (1977)[2] |
| 3,5-dimethoxy-4-hydroxy benzoate | c1c(OC)c(O)c(OC)cc1C(=O)[O-] | CH3-O-c1c(OH)c(-O-CH3)cHc(c1H)CO(Om) | $1.6 \cdot 10^{10}$ | | | O'Neill et al. (1977)[2] |
| 2,3,4-trimethoxy benzoate | c1(OC)c(OC)c(OC)ccc1C(=O)[O-] | CH3-O-c1c(-O-CH3)cHcHc(CO(Om))c1-O-CH3 | $1.0 \cdot 10^{10}$ | | | O'Neill et al. (1977)[2] |
| 2,3,5-trimethoxy benzoate | c1(OC)c(OC)cc(OC)c1C(=O)[O-] | CH3-O-c1cHc(-O-CH3)cHc(CO(Om))c1-O-CH3 | $7.0 \cdot 10^{9}$ | | | O'Neill et al. (1977)[2] |
| 2,4,6-trimethoxy benzoate | c1(OC)cc(OC)cc(OC)c1C(=O)[O-] | CH3-O-c1cHc(-O-CH3)cHc(-O-CH3)c1CO(Om) | $1.2 \cdot 10^{10}$ | | | O'Neill et al. (1977)[2] |
| 3,4,5-trimethoxy benzoate | c1c(OC)c(OC)c(OC)cc1C(=O)[O-] | CH3-O-c1c(-O-CH3)Hc(CO(Om))cHc1-O-CH3 | $1.3 \cdot 10^{10}$ | | | O'Neill et al. (1977)[2] |
| 3-methoxy-4-hydroxy cinnamate | c1c(OC)c(O)ccc1C=CC(=O)[O-] | CH3-O-c1c(OH)cHcHc(c1H)CdH=CdHCO(Om) | $1.0 \cdot 10^{10}$ | | | Bors et al. (1982)[2] |
| 3,5-dimethoxy-4-hydroxy cinnamate | c1c(OC)c(O)c(OC)cc1C=CC(=O)[O-] | CH3-O-c1c(OH)c(-O-CH3)cHc(c1H)CdH=CdHCO(Om) | $2.2 \cdot 10^{10}$ | | | Buxton et al. (1988) |
| phenoxy acetic acid | c1ccccc1OCC(=O)O | c1HcHcHcHcHc1-O-CH2CO(OH) | $1.0 \cdot 10^{10}$ | | | Zona et al. (2002)[3] |
| 1-(p-ethoxyphenyl)ethanol | c1cc(OCC)ccc1C(O)C | CH3CH(OH)c1cHcHc(cHc1H)-O-CH2CH3 | $2.7 \cdot 10^{9}$ | | | Snook and Hamilton (1974) |
| 1-(p-methoxyphenyl)-2,2-dimethyl propanol | c1cc(OC)ccc1C(O)C(C)(C)C | c1HcHc(-O-CH3)cHcHc1CH(OH)C(CH3)(CH3)CH3 | $7.6 \cdot 10^{9}$ | | | Snook and Hamilton (1974) |
| 3,5-dinitro anisole | c1c(N(=O)=O)cc(N(=O)=O)cc1OC | c1Hc(NO2)cHc(NO2)cHc1-O-CH3 | $4.0 \cdot 10^{9}$ | | | Tamminga et al. (1979)[2] |
| Isoeugenol | c1c(OC)c(O)ccc1C=CC | CH3-O-c1c(OH)cHcHc(c1H)CdH=CdHCH3 | $3.9 \cdot 10^{10}$ | | | Buxton et al. (1988) |
| Propyl 3,4,5-trihydroxybenzoate | c1c(O)c(O)c(O)cc1C(=O)OCCC | c1Hc(OH)c(OH)c(OH)cHc1CO-O-CH2CH2CH3 | $1.1 \cdot 10^{10}$ | | | Buxton et al. (1988) |

| Polycyclic aromatic compounds (including sugars) | | | | | | $n = 34$ |
|---|---|---|---|---|---|---|
| Endothall | OC(=O)C2C(C1CCC2O1)C(=O)O | CO(OH)C1HCH(CO(OH))C2H-O-C1HCH2C2H2 | $1.5 \cdot 10^{9}$ | | | Haag and Yao (1992)[8] |
| Camphor | CC2(C)C1CC(=O)C2(C)CC1 | C2H2CH2CH(C1(CH3)CH3)CH2COC12CH3 | $4.1 \cdot 10^{9}$ | | | Land and Swallow (1979) |
| 2-phenyl furan | c1ccccc1c2ccco2 | -O2-CdH=CdHCdH=Cd2c1cHcHcHcHcHc1H | $1.6 \cdot 10^{10}$ | | | Vysotskaya et al. (1983)[2] |
| 5-phenyl furfural | c1ccccc1c2ccc(C=O)o2 | -O2-Cd(CHO)=CdHCdH=Cd2c1cHcHcHcHcHc1H | $5.9 \cdot 10^{9}$ | | | Vysotskaya et al. (1983)[2] |
| Biphenyl | c1ccccc1c2ccccc2 | c1HcHcHcHcHc1c2cHcHcHcHcHc2H | $1.04 \cdot 10^{10}$ | | | Chen and Schuler (1993); partial $k$ @ o/m/p position: $1.07 \cdot 10^{9}/5.5 \cdot 10^{8}/1.52 \cdot 10^{9}$ |
| Biphenyl-4-carboxylate | c1ccccc1c2ccc(C(=O)[O-])cc2 | c1HcHc(CO(Om))cHcHc1c2cHcHcHcHc2H | $6.8 \cdot 10^{9}$ | | | Simic et al. (1973a)[2] |
| Biphenyl-2,2'-dicarboxylate | [O-]C(=O)c1ccccc1c2ccccc2C(=O)[O-] | CO(Om)c1cHcHcHcHc1c2cHcHcHcHcHc2CO(Om) | $7.0 \cdot 10^{9}$ | | | Simic et al. (1973a)[2] |
| Biphenyl-4,4'-dicarboxylate | c1cc(C(=O)[O-])ccc1c2ccc(C(=O)[O-])cc2 | c1HcHc(CO(Om))cHcHc1c2cHcHc(CO(Om))cHc2H | $8.3 \cdot 10^{9}$ | | | Simic et al. (1973a)[2] |
| Benzophenone | c1ccccc1C(=O)c2ccccc2 | c1HcHcHcHcHc1COc2cHcHcHcHcHc2H | $8.8 \cdot 10^{9}$ | | | Buxton et al. (1988) |
| Diphenyl acetate | c1ccccc1C(C(=O)[O-])c2ccccc2 | | $4.0 \cdot 10^{9}$ | | | Neta et al. (1972)[2] |
| 4-Phenoxybenzoate | c1ccccc1Oc2ccc(C(=O)[O-])cc2 | | $7.0 \cdot 10^{9}$ | | | Neta and Schuler (1975)[2] |
| 1,4-(3,4-dihydroxyphenyl)-2,3-dimethyl butane | c1cc(O)c(O)cc1CC(C)C(C)Cc2cc(O)c(O)cc2 | c1HcHc(OH)c(OH)cHc1CH2CH(CH3)CH(CH3)CH2c2cHc(OH)c(OH)cHc2H | $1.5 \cdot 10^{10}$ | | | Bors et al. (1982)[2] |
| 1,2,3,4-tetrahydro-1-naphthol/ α-tetralol | OC2CCCc1ccccc12 | c2HcHcHcHc1CH(OH)CH2CH2CH2c12 | $7.0 \cdot 10^{9}$ | | | Snook and Hamilton (1974)[2] |

| Reactant | SMILES | GECKO-A | $k_{298}$ [M$^{-1}$s$^{-1}$] | A [M$^{-1}$s$^{-1}$] | $E_a$ / R [K] | Reference/remarks | n = 467 (282 aliphatic/185 aromatic) |
|---|---|---|---|---|---|---|---|
| Coumarin | O=C1C=Cc2ccccc2O1 | | $2.0 \cdot 10^9$ | | | Gopakumar et al. (1977) | |
| β-Benzylglucoside | c1cccc1COC2C(O)C(O)C(O)C(CO)O2 | C2H(OH)CH(OH)CH(OH)CH(CH2(OH))-O-C2H-O-CH2c1cHcHcHcHc1H | $4.2 \cdot 10^{10}$ | | | Mittal and Mittal (1986)[2] | |
| 2,4dimethylphenyl-β-D-glucopyranoside | Cc1cc(C)ccc1COC2C(O)C(O)C(O)C(CO)O2 | C2H(OH)CH(OH)CH(OH)CH(CH2(OH))-O-C2H-O-CH2c1cHcHc(CH3)cHcHc1CH3 | $3.9 \cdot 10^9$ | | | Phillips et al. (1971)[2] | |
| 3,4dimethylphenyl-β-D-glucopyranoside | c1c(C)c(C)ccc1COC2C(O)C(O)C(O)C(CO)O2 | C2H(OH)CH(OH)CH(OH)CH(CH2(OH))-O-C2H-O-CH2c(cHc1CH3)cHcHc1CH3 | $4.2 \cdot 10^9$ | | | Phillips et al. (1971)[2] | |
| Naphthalene | c1cccc2ccccc12 | c12cHcHcHcHc1cHcHcHc2H | $9.6 \cdot 10^9$ | | | Average of Evers et al. (1980)[2]; Zevos and Sehested (1978)[2]; Roder et al. (1990)[1]; and Kanodia et al. (1988)[1] | |
| 1-naphthol | Oc2cccc1ccccc12 | c2HcHcHcHc1c(OH)cHcHcHc12 | $1.2 \cdot 10^{10}$ | | | Average of Doherty et al. (1986)[8] and Kanodia et al. (1988)[8] | |
| 2-naphthol | Oc1ccc2ccccc2c1 | c2HcHcHcHc1cHc(OH)cHcHc12 | $1.2 \cdot 10^{10}$ | | | Kanodia et al. (1988)[1] | |
| 1-naphthoate | [O-]C(=O)c2cccc1ccccc12 | CO(Om)c2cHcHcHc1cHcHcHcHc12 | $7.9 \cdot 10^9$ | | | Simic et al. (1973a)[2] | |
| 2-naphthoate | [O-]C(=O)c1ccc2ccccc2c1 | CO(Om)c(c2H)cHcHc1cHcHcHcHc12 | $7.6 \cdot 10^9$ | | | Simic et al. (1973a)[2] | |
| 1-naphthylacetate | [O-]C(=O)Cc2cccc1ccccc12 | CO(Om)CH2c2cHcHcHc1cHcHcHcHc12 | $8.7 \cdot 10^9$ | | | Shetiya et al. (1972)[2] | |
| Di-tert-butylnaphtalenesulfonate | CC(C)(C)c1ccc2c(c1)cc(cc2S([O-])(=O)=O)C(C)(C)C | | $1.1 \cdot 10^{10}$ | | | Barber and Thomas (1978)[2] | |
| chromotropic acid | O=S(=O)(O)c2cc1cc(cc(O)c1c(O)c2)S(=O)(=O)O | | $1.2 \cdot 10^8$ | | | Ahrens (1967)[2] | |
| 9,10-Anthraquinone-1-sulfonate | [O-]S(=O)(=O)c3cccc2C(=O)c1ccccc1C(=O)c23 | | $7.2 \cdot 10^9$ | | | Hulme et al. (1972)[2] | |
| 9,10-Anthraquinone-2-sulfonate | [O-]S(=O)(=O)c2ccc3C(=O)c1ccccc1C(=O)c3c2 | | $5.6 \cdot 10^9$ | | | Hulme et al. (1972)[2] | |
| 8-Methoxypsoralen | COc1c3occc3cc2C=CC(=O)Oc12 | | $1.1 \cdot 10^{10}$ | | | Redpath et al. (1978)[2] | |
| Flourescein | [O-]C(=O)c4ccccc4C=1c3ccc([O-])cc3OC2=CC(=O)C=CC=12 | | $1.2 \cdot 10^{10}$ | | | Prütz (1973)[2] | |
| Pyrene butyrate | [O-]C(=O)CCCc1cc2ccc3cccc4ccc(c1)c2c34 | | $1.3 \cdot 10^{10}$ | | | Barber and Thomas (1978)[2] | |
| D-cellobiose | OCC1OC(O)C(O)C(O)C1OC2C(O)C(O)C(O)C(CO)O2 | C2H(OH)CH(OH)CH(OH)CH(CH2(OH))-O-C1HCH(OH)CH(OH)CH(OH)-O-C1HCH2(OH) | $3.6 \cdot 10^9$ | | | Zakatova et al. (1969)[2] | |
| crocin | O1C(CO)C(O)C(O)C(O)C1OCC2C(O)C(O)C(O)C(O2)OC(=O)C(C)=CC=CC(C)=CC=CC=C(C)C=CC=C(C)C(=O)OC3C(O)C(O)C(O)C(O3)COC4C(O)(O)C(O)C(O4)CO | | $3.1 \cdot 10^9$ | | | Buxton et al. (1988) | |
| carmine | OC1C(O)C(O)C(OC1CO)c4c(O)c3C(=O)c2c(C)c(C(=O)O)c(O)cc2C(=O)c3c(O)c4O | | $1.3 \cdot 10^{10}$ | | | Sychev et al. (1979)[2] | |
| *9-anthroate ion* | | | $8.0 \cdot 10^9$ | | | Simic et al. (1973b)[2] | |

**Remarks:** References taken from:    [1]NIST database: Ross et al. (1998); [2]Buxton et al. (1988); [3]Herrmann (2003); [4] Gligorovski et al. (2009); [5](Cooper et al. (2009); [6]CAPRAM database available at http://projects.tropos.de/capram/; [7] de Sémainville et al. (2007); [8]Warneck (2005); [9]Barzaghi and Herrmann (2004)

**Table S2 Recommend kinetic data for NO₃ reactions with organic compounds.**

| Reactant | SMILES | GECKO-A | k298 [M⁻¹s⁻¹] | A [M⁻¹s⁻¹] | Eₐ / R [K] | Reference/remarks | n = 130 (81 aliphatic/49 aromatic) |
|---|---|---|---|---|---|---|---|
| **Alkenes** | | | | | | | n = 1 |
| Isoprene | C=C(C)C=C | CdH2=CdHCd(CH3)=CdH2 | $1.0 \cdot 10^9$ | | | TROPOS measurements | |
| | | | | | | | |
| **Monoalcohols** | | | | | | | n = 14 |
| Methanol | CO | CH3OH | $5.1 \cdot 10^5$ | $9.4 \cdot 10^{11}$ | 4300 | Average of Exner et al. (1993) and Rousse and George (2004) | |
| Ethanol | CCO | CH3CH2OH | $2.2 \cdot 10^6$ | $1.4 \cdot 10^{11}$ | 3300 | Herrmann and Zellner (1998) | |
| Propanol | CCCO | CH3CH2CH2OH | $3.2 \cdot 10^6$ | | | Herrmann et al. (1994) | |
| Butanol | CCCCO | CH3CH2CH2CH2OH | $1.9 \cdot 10^6$ | | | Shastri and Huie (1990) | |
| Pentanol | CCCCCO | CH3CH2CH2CH2CH2OH | $2.4 \cdot 10^6$ | | | Shastri and Huie (1990) | |
| Hexanol | CCCCCCO | CH3CH2CH2CH2CH2CH2OH | $3.3 \cdot 10^6$ | | | Shastri and Huie (1990) | |
| Heptanol | CCCCCCCO | CH3CH2CH2CH2CH2CH2CH2OH | $3.6 \cdot 10^6$ | | | Shastri and Huie (1990) | |
| octanol | CCCCCCCCO | CH3CH2CH2CH2CH2CH2CH2CH2OH | $5.8 \cdot 10^6$ | | | Shastri and Huie (1990) | |
| Iso-propanol | CC(O)C | CH3CH(OH)CH3 | $3.7 \cdot 10^6$ | $3.1 \cdot 10^8$ | 1323 | Herrmann et al. (1994) with Eₐ/R of Ito et al. (1989b) | |
| Iso-butanol | CC(C)CO | CH3CH(CH3)CH2(OH) | $1.6 \cdot 10^6$ | | | Shastri and Huie (1990) | |
| Tert-butanol | CC(C)(C)O | CH3C(OH)(CH3)CH3 | $6.6 \cdot 10^4$ | | | Herrmann et al. (1994) | |
| Allyl alcohol | C=CCO | CH2(OH)CdH=CdH2 | $2.2 \cdot 10^8$ | | | Average of Alfassi et al. (1993) and Ito et al. (1989b) | |
| 2-butenol | CC=CCO | CH3CdH=CdHCH2(OH) | $2.1 \cdot 10^9$ | | | Alfassi et al. (1993) | |
| 3-methyl 3-buten-1-ol | C=C(C)CCO | CH2(OH)CH2Cd(CH3)=CdH2 | $2.4 \cdot 10^9$ | | | Alfassi et al. (1993) | |
| | | | | | | | |
| **Diols and Polyols** | | | | | | | n = 2 |
| ethylene glycol | OCCO | CH2(OH)CH2(OH) | $6.6 \cdot 10^6$ | $7.1 \cdot 10^9$ | 2117 | Hoffmann et al. (2009) | |
| 1,2-propanediol | CC(O)CO | CH3CH(OH)CH2(OH) | $9.9 \cdot 10^6$ | $6.8 \cdot 10^{10}$ | 2622 | Hoffmann et al. (2009) | |
| | | | | | | | |
| **Aldehydes and gem-diols** | | | | | | | n = 11 |
| Formaldehyde | C=O | CH2O | $3.5 \cdot 10^6$ | $3.36 \cdot 10^6$ | 674 | Average of (Ito et al. (1989a)[6] and Wayne et al. (1991) with Eₐ/R of Wayne et al. (1991) | |
| Hydrated formaldehyde | OCO | CH2(OH)(OH) | $1.0 \cdot 10^6$ | $3.6 \cdot 10^{12}$ | 4500 | Exner et al. (1993)[1] | |
| Acetaldehyde | CC=O | CH3CHO | $1.9 \cdot 10^6$ | | | Zellner et al. (1996) | |
| Hydrated acetaldehyde | CC(O)O | CH3CH(OH)(OH) | $2.0 \cdot 10^6$ | | | Average of Zellner et al. (1996)[2] and Rousse and George (2004) | |
| Propionaldehyde | CCC=O | CH3CH2CHO | $5.8 \cdot 10^7$ | $3.2 \cdot 10^{11}$ | 2646 | de Sémainville et al. (2007) | |
| Butyraldehyde | CCCC=O | CH3CH2CH2CHO | $5.6 \cdot 10^7$ | $4.9 \cdot 10^{10}$ | 2045 | de Sémainville et al. (2007) | |
| Iso-butyraldehyde | CC(C)C=O | CH3CH(CH3) CHO | $6.3 \cdot 10^7$ | $3.7 \cdot 10^8$ | 529 | Wayne et al. (1991) | |
| 2,2-dimethyl propanal | CC(C)(C)C=O | CH3C(CH3)(CH3)CHO | $7.0 \cdot 10^7$ | $3.8 \cdot 10^8$ | 505 | Wayne et al. (1991) | |
| Methacrolein | C=C(C)C=O | CH3Cd(CHO)=CdH2 | $4.0 \cdot 10^7$ | $5.8 \cdot 10^8$ | 842 | Schöne et al. (2014) | |
| Glyoxal | O=CC=O | CHOCHO | $1.1 \cdot 10^6$ | $9.9 \cdot 10^{10}$ | 3400 | TROPOS measurement | |
| Hydrated glyoxal | OC(O)C(O)O | CH(OH)(OH)CH(OH)(OH) | $1.1 \cdot 10^6$ | $8.9 \cdot 10^{10}$ | 3368 | Herrmann et al. (1995b)[2] | |
| | | | | | | | |
| **Ketones** | | | | | | | n = 3 |
| Acetone | CC(=O)C | CH3COCH3 | $3.7 \cdot 10^3$ | $7.6 \cdot 10^9$ | 4330 | Herrmann and Zellner (1998) | |
| Methyl ethyl ketone | CC(=O)CC | CH3CH2COCH3 | $2.2 \cdot 10^7$ | $3.9 \cdot 10^{11}$ | 2887 | de Sémainville et al. (2007) | |
| Methyl vinyl ketone | CC(=O)C=C | CH3COCdH=CdH2 | $9.7 \cdot 10^6$ | $6.2 \cdot 10^8$ | 1200 | Schöne et al. (2014) | |
| | | | | | | | |
| **Monocarboxylic acids (undissociated and dissociated)** | | | | | | | n = 10 |
| Formic acid | C(=O)O | CHO(OH) | $3.8 \cdot 10^5$ | $3.4 \cdot 10^{10}$ | 3400 | Exner et al. (1994) | |
| Formate | C(=O)[O-] | CHO(Om) | $5.1 \cdot 10^7$ | $8.2 \cdot 10^{10}$ | 2200 | Exner et al. (1994)[3] | |
| Acetic acid | CC(=O)O | CH3CO(OH) | $1.3 \cdot 10^4$ | $4.9 \cdot 10^9$ | 3800 | Exner et al. (1994)[4] | |
| Acetate | CC(=O)[O-] | CH3CO(Om) | $2.9 \cdot 10^6$ | $1.0 \cdot 10^{12}$ | 3800 | Exner et al. (1994)[4] | |
| Propionic acid | CCC(=O)O | CH3CH2CO(OH) | $7.7 \cdot 10^4$ | | | Rousse and George (2004) | |
| Acrylic acid | C=CC(=O)O | CO(OH)CdH=CdH2 | $6.9 \cdot 10^6$ | $2.2 \cdot 10^{13}$ | 4450 | Schöne et al. (2014) | |
| Acrylate | C=CC(=O)[O-] | CO(Om)CdH=CdH2 | $4.4 \cdot 10^7$ | $2.2 \cdot 10^9$ | 1200 | Schöne et al. (2014) | |
| Crotonic acid | CC=CC(=O)O | CH3CdH=CdHCO(OH) | $5.1 \cdot 10^7$ | | | Neta and Huie (1986) | |
| Methacrylic acid | C=C(C)C(=O)O | CH3Cd(CO(OH))=CdH2 | $9.2 \cdot 10^7$ | | | Schöne et al. (2014) | |
| Methacrylate | C=C(C)C(=O)[O-] | CH3Cd(CO(Om))=CdH2 | $1.7 \cdot 10^8$ | | | Schöne et al. (2014) | |
| | | | | | | | |
| **Dicarboxylic acids (undissociated and dissociated)** | | | | | | | n = 9 |
| Oxalic acid | OC(=O)C(=O)O | CO(OH)CO(OH) | $2.4 \cdot 10^4$ | | | Yang et al. (2004) | |
| Oxalate monoanion | OC(=O)C(=O)[O-] | CO(OH)CO(Om) | $6.1 \cdot 10^7$ | $8.4 \cdot 10^9$ | -2180 | Average of Yang et al. (2004) and de Sémainville et al. (2010) with Eₐ/R of Raabe (1996) | |

Formatted Table

| Reactant | SMILES | GECKO-A | $k298$ [M$^{-1}$s$^{-1}$] | A [M$^{-1}$s$^{-1}$] | $E_a/R$ [K] | Reference/remarks      n = 130 (81 aliphatic/49 aromatic) |
|---|---|---|---|---|---|---|
| Oxalate dianion | [O-]C(=O)C(=O)[O-] | CO(Om)CO(Om) | $2.2 \cdot 10^8$ | $2.2 \cdot 10^{12}$ | 2766 | Average of Yang et al. (2004) and de Sémainville et al. (2010) with $E_a/R$ of de Sémainville et al. (2010) |
| Malonic acid | OC(=O)CC(=O)O | CO(OH)CH2CO(OH) | $5.1 \cdot 10^4$ | | | de Sémainville et al. (2010) |
| Malonate monoanion | OC(=O)CC(=O)[O-] | CO(OH)CH2CO(Om) | $5.6 \cdot 10^6$ | $5.0 \cdot 10^{11}$ | 3368 | de Sémainville et al. (2010) |
| Malonate dianion | [O-]C(=O)CC(=O)[O-] | CO(Om)CH2CO(Om) | $2.3 \cdot 10^7$ | $6.3 \cdot 10^{11}$ | 3007 | de Sémainville et al. (2010) |
| Succinic acid | OC(=O)CCC(=O)O | CO(OH)CH2CH2CO(OH) | $5.0 \cdot 10^3$ | | | de Sémainville et al. (2010) |
| Succinate monoanion | OC(=O)CCC(=O)[O-] | CO(OH)CH2CH2CO(Om) | $1.1 \cdot 10^7$ | | | de Sémainville et al. (2010) |
| Succinate dianion | [O-]C(=O)CCC(=O)[O-] | CO(Om)CH2CH2CO(Om) | $1.8 \cdot 10^7$ | $6.2 \cdot 10^{11}$ | 3127 | de Sémainville et al. (2010) |
| **Ethers and Esters** | | | | | | n = 9 |
| Methyl tert-butyl ether | CC(C)(C)OC | CH3-O-C(CH3)(CH3)CH3 | $3.9 \cdot 10^5$ | | | Rousse and George (2004) |
| Methyl formate | COC=O | CH3-O-CHO | $3.5 \cdot 10^6$ | | | Buxton et al. (2001)[5] |
| Ethyl formate | CCOC=O | CH3CH2-O-CHO | $4.7 \cdot 10^6$ | | | Buxton et al. (2001) |
| Methyl acetate | CC(=O)OC | CH3CO-O-CH3 | $<10^4$ | | | Buxton et al. (2001) |
| Ethyl acetate | CC(=O)OCC | CH3CH2-O-COCH3 | $<10^4$ | | | Buxton et al. (2001) |
| Dimethyl malonate | COC(=O)CC(=O)OC | CH3-O-COCH2CO-O-CH3 | $2.6 \cdot 10^4$ | | | Rousse and George (2004) |
| Dimethyl succinate | COC(=O)CCC(=O)OC | CH3-O-COCH2CH2CO-O-CH3 | $3.4 \cdot 10^4$ | | | Rousse and George (2004) |
| Dimethyl carbonate | COC(=O)OC | CH3-O-CO-O-CH3 | $8.4 \cdot 10^4$ | | | Rousse and George (2004) |
| Diethyl carbonate | CCOC(=O)OCC | CH3CH2-O-CO-O-CH2CH3 | $1.5 \cdot 10^4$ | | | Rousse and George (2004) |
| **Aliphatic polyfunctional compounds** | | | | | | n = 11 |
| Hydroxy acetone | CC(=O)CO | CH3COCH2(OH) | $1.8 \cdot 10^7$ | $4.0 \cdot 10^9$ | 1564 | de Sémainville et al. (2007) |
| Glycolic acid | OCC(=O)O | CH2(OH)CO(OH) | $9.1 \cdot 10^5$ | $4.5 \cdot 10^{11}$ | 3969 | de Sémainville et al. (2007) |
| Glycolate | OCC(=O)[O-] | CH2(OH)CO(Om) | $1.0 \cdot 10^7$ | $1.8 \cdot 10^{11}$ | 3007 | de Sémainville et al. (2007) |
| Lactic acid | CC(O)C(=O)O | CH3CH(OH)CO(OH) | $2.1 \cdot 10^6$ | $1.0 \cdot 10^{11}$ | 3248 | de Sémainville et al. (2007) |
| Lactate | CC(O)C(=O)[O-] | CH3CH(OH)CO(Om) | $1.0 \cdot 10^7$ | $8.3 \cdot 10^{10}$ | 2646 | de Sémainville et al. (2007) |
| Pyruvic acid | CC(=O)C(=O)O | CH3COCO(OH) | $2.4 \cdot 10^6$ | $8.8 \cdot 10^8$ | 1804 | de Sémainville et al. (2007) |
| Pyruvate | CC(=O)C(=O)[O-] | CH3COCO(Om) | $1.9 \cdot 10^7$ | $3.7 \cdot 10^{11}$ | 2887 | de Sémainville et al. (2007) |
| Mesoxalic acid | OC(=O)C(=O)C(=O)O | CO(OH)COCO(OH) | $1.7 \cdot 10^6$ | $5.1 \cdot 10^8$ | 1564 | de Sémainville et al. (2010) |
| Mesoxalate monoanion | OC(=O)C(=O)C(=O)[O-] | | $2.3 \cdot 10^7$ | | | de Sémainville et al. (2010) |
| Mesoxalate dianion | [O-]C(=O)C(=O)C(=O)[O-] | | $4.9 \cdot 10^7$ | $1.4 \cdot 10^{12}$ | 3127 | de Sémainville et al. (2010) |
| Fumaric acid | OC(=O)/C=C/C(=O)O | CO(OH)CdH=CdHCO(OH) | $<1.0 \cdot 10^6$ | | | Neta and Huie (1986)[3] |
| **Polyols and sugars** | | | | | | n = 5 |
| Glycerol | OCC(O)CO | CH2(OH)CH(OH)CH2(OH) | $1.3 \cdot 10^7$ | $1.4 \cdot 10^{12}$ | 3452 | Hoffmann et al. (2009) |
| Erythritol | OCC(O)C(O)CO | CH2(OH)CH(OH)CH(OH)CH2(OH) | $1.4 \cdot 10^7$ | $3.4 \cdot 10^{10}$ | 2321 | Hoffmann et al. (2009) |
| Arabitol | OCC(O)C(O)C(O)CO | CH2(OH)CH(OH)CH(OH)CH(OH)CH2(OH) | $1.5 \cdot 10^7$ | $1.1 \cdot 10^{10}$ | 1997 | Hoffmann et al. (2009) |
| Mannitol | OCC(O)C(O)C(O)C(O)CO | CH2(OH)CH(OH)CH(OH)CH(OH)CH(OH)CH(OH) | $1.4 \cdot 10^7$ | $5.1 \cdot 10^{10}$ | 2466 | Hoffmann et al. (2009) |
| Levoglucosan | OC1C(O)C(O)C2OC1OC2 | C1H2-O-CH(-O-C12H)CH(OH)CH(OH)C2H(OH) | $1.6 \cdot 10^7$ | $2.5 \cdot 10^{10}$ | 2150 | TROPOS measurements |
| **Cyclic aliphatic compounds** | | | | | | n = 6 |
| Cyclopentanol | C1CCCC1O | C1H2CH2CH(OH)CH2C1H2 | $3.2 \cdot 10^6$ | | | Shastri and Huie (1990)[6] |
| Oxetane | C1COC1 | C1H2CH2-O-C1H2 | $1.5 \cdot 10^6$ | | | Shastri and Huie (1990)[6] |
| Tetrahydrofuran | C1CCCC1 | C1H2CH2-O-CH2C1H2 | $1.5 \cdot 10^7$ | | | Herrmann and Zellner (1998)[4] |
| Tetrahydropyran | C1CCCCO1 | C1H2CH2CH2-O-CH2C1H2 | $4.9 \cdot 10^6$ | | | Shastri and Huie (1990) |
| 1,3-dioxane | C1COCOC1 | C1H2CH2-O-CH2-O-C1H2 | $7.7 \cdot 10^5$ | | | Shastri and Huie (1990) |
| 1,4-dioxane | C1OCCOC1 | C1H2-O-CH2CH2-O-C1H2 | $1.3 \cdot 10^6$ | | | Shastri and Huie (1990) |
| **Benzols** | | | | | | n = 6 |
| Benzol | c1ccccc1 | c1HcHcHcHcHc1H | $4.0 \cdot 10^8$ | | | Herrmann et al. (1996) |
| Toluol | c1ccccc1C | c1HcHcHcHcHc1CH3 | $1.2 \cdot 10^9$ | $5.7 \cdot 10^{11}$ | 1800 | Herrmann and Zellner (1998) |
| Ethyl benzol | c1ccccc1CC | c1HcHcHcHcHc1CH2CH3 | $1.3 \cdot 10^9$ | | | Herrmann et al. (1996) |
| Tert-butyl benzol | c1ccccc1C(C)(C)C | c1HcHcHcHcHc1C(CH3)(CH3)CH3 | $1.1 \cdot 10^9$ | | | TROPOS measurements |
| p-xylene | c1cc(C)ccc1C | c1HcHc(CH3)cHcHc1CH3 | $1.6 \cdot 10^9$ | | | Herrmann et al. (1996) |
| Mesitylene | c1c(C)cc(C)cc1C | c1Hc(CH3)cHc(CH3)cHc1CH3 | $1.3 \cdot 10^9$ | | | Herrmann et al. (1996) |
| **Phenols and other aromatic alcohols** | | | | | | n = 10 |
| Phenol | c1ccccc1O | c1HcHcHc(OH)cHc1H | $1.9 \cdot 10^9$ | $1.4 \cdot 10^{12}$ | 2100 | Umschlag et al. (2002) |
| Catechol | c1cccc(O)c1O | c1HcHc(OH)c(OH)cHc1H | $5.6 \cdot 10^8$ | $3.7 \cdot 10^{15}$ | 4691 | Barzaghi and Herrmann (2004) |
| Hydroquinone | c1cc(O)ccc1O | c1Hc(OH)cHcHc(OH)c1H | $1.1 \cdot 10^9$ | | | TROPOS measurements |
| pyrogallol | Oc1c(O)cccc1O | c1HcHc(OH)c(OH)c(OH)c1H | $1.7 \cdot 10^9$ | $6.9 \cdot 10^{10}$ | 1100 | TROPOS measurements |

| Reactant | SMILES | GECKO-A | $k_{298}$ [M$^{-1}$s$^{-1}$] | A [M$^{-1}$s$^{-1}$] | $E_a / R$ [K] | Reference/remarks | n = 130 (81 aliphatic/49 aromatic) |
|---|---|---|---|---|---|---|---|
| o-ethyl phenol | CCc1ccccc1O | c1HcHcHcHc(OH)c1CH2CH3 | $6.7 \cdot 10^8$ | | 2165 | Barzaghi and Herrmann (2004) | |
| p-ethyl phenol | c1cc(CC)ccc1O | c1HcHc(OH)cHcHc1CH2CH3 | $1.6 \cdot 10^9$ | $4.6 \cdot 10^{11}$ | 1672 | TROPOS measurements | |
| p-tert-butyl phenol | c1cc(O)ccc1C(C)(C)C | c1HcHc(OH)cHcHc1C(CH3)(CH3)CH3 | $1.1 \cdot 10^9$ | | | TROPOS measurements | |
| Benzyl alcohol | c1ccccc1CO | c1HcHcHcHcHc1CH2(OH) | $4.5 \cdot 10^8$ | | | Ito et al. (1989b)[3] | |
| o-cresol | c1(O)ccccc1C | c1HcHcHcHc(OH)c1CH3 | $8.5 \cdot 10^8$ | $6.1 \cdot 10^{12}$ | 2646 | Barzaghi and Herrmann (2004) | |
| p-cresol | c1cc(O)ccc1C | c1cc(OH)cHcHc1CH3 | $1.7 \cdot 10^9$ | $9.0 \cdot 10^{11}$ | 1756 | TROPOS measurements | |
| **Aromatic carbonyls and acids** | | | | | | | n = 5 |
| Phenyl acetone | c1ccccc1C(=O)C | c1HcHcHcHcHc1COCH3 | $1.4 \cdot 10^7$ | | | Neta and Huie (1986)[3] | |
| Benzoic acid | c1ccccc1C(=O)O | c1HcHcHcHcHc1CO(OH) | $6.5 \cdot 10^7$ | $4.9 \cdot 10^9$ | 1300 | Umschlag et al. (2002) | |
| 4-methyl benzoic acid | c1cc(C)ccc1C(=O)O | CH3c1cHcHc(CO(OH))cHc1H | $6.0 \cdot 10^8$ | $1.6 \cdot 10^{12}$ | 2400 | Umschlag et al. (2002) | |
| Phenyl acetic acid | c1ccccc1CC(=O)O | c1HcHcHcHcHc1CH2CO(OH) | $1.8 \cdot 10^9$ | | | Neta and Huie (1986) | |
| Phenyl acetate | c1ccccc1CC(=O)[O-] | c1HcHcHcHcHc1CH2CO(Om) | $1.4 \cdot 10^8$ | | | TROPOS measurements | |
| **Anisoles** | | | | | | | n = 4 |
| Anisole | c1ccccc1OC | c1HcHcHcHcHc1-O-CH3 | $1.0 \cdot 10^9$ | | | Herrmann et al. (1995a) | |
| m-methyl anisole | c1c(C)cccc1OC | c1Hc(CH3)cHcHc1-O-CH3 | $2.0 \cdot 10^9$ | | | TROPOS measurements | |
| p-methyl anisole | c1cc(C)ccc1OC | c1HcHc(CH3)cHcHc1-O-CH3 | $3.2 \cdot 10^9$ | | | TROPOS measurements | |
| p-dimethoxy benzol | c1cc(OC)ccc1OC | c1HcHc(-O-CH3)cHcHc1-O-CH3 | $1.2 \cdot 10^9$ | $7.8 \cdot 10^9$ | 500 | TROPOS measurements | |
| **Polyfunctional aromatic compounds** | | | | | | | n = 22 |
| o-nitro phenol | Oc1ccccc1N(=O)=O | c1HcHc(OH)c(NO2)cHc1H | $2.3 \cdot 10^7$ | | | Barzaghi and Herrmann (2004) | |
| p-nitro phenol | c1cc(O)ccc1N(=O)=O | c1Hc(OH)cHcHc(NO2)c1H | $1.4 \cdot 10^9$ | $4.0 \cdot 10^{11}$ | 1684 | TROPOS measurements | |
| 2,4-dinitro phenol | Oc1ccc(N(=O)=O)cc1N(=O)=O | c1Hc(OH)c(NO2)cHc(NO2)c1H | $5.3 \cdot 10^7$ | | | Umschlag et al. (2002)[5] | |
| 2,6-dinitro phenol | Oc1c(N(=O)=O)cccc1N(=O)=O | c1Hc(NO2)c(OH)c(NO2)c1H | $2.8 \cdot 10^8$ | $3.2 \cdot 10^{11}$ | 2165 | TROPOS measurements | |
| 2-nitro-4-methyl phenol | Oc1ccc(C )cc1N(=O)=O | c1Hc(OH)c(NO2)cHc1CH3 | $1.0 \cdot 10^8$ | | | Umschlag et al. (2002) | |
| 2,6-dinitro p-cresol | Oc1c(N(=O)=O)cc(C)cc1N(=O)=O | c1Hc(NO2)c(OH)c(NO2)cHc1CH3 | $1.4 \cdot 10^8$ | | | Umschlag et al. (2002) | |
| 4-hydroxy benzoic acid | c1cc(O)ccc1C(=O)O | c1Hc(OH)cHcHc1CO(OH) | $1.6 \cdot 10^9$ | $3.2 \cdot 10^{11}$ | 1588 | TROPOS measurements | |
| 4-hydroxy benzoate | c1cc(O)ccc1C(=O)[O-] | | $6.0 \cdot 10^9$ | | | Anderson et al. (1987)[7] | |
| 3-nitro benzoic acid | c1c(N(=O)=O)cccc1C(=O)O | c1HcHc(NO2)cHc1CO(OH) | $2.0 \cdot 10^7$ | | | Umschlag et al. (2002) | |
| 4-nitro benzoic acid | c1c(N(=O)=O)ccc1C(=O)O | c1HcHc(NO2)cHcHc1CO(OH) | $2.0 \cdot 10^7$ | | | Umschlag et al. (2002) | |
| 3-nitro-4-methyl benzoate | c1cc(C)c(N(=O)=O)cc1C(=O)[O-] | CH3c1c(NO2)cHc(CO(OH))cHc1H | $3.3 \cdot 10^7$ | | | Umschlag et al. (2002) | |
| o- methoxy phenol | Oc1ccccc1OC | c1HcHcHcHc(OH)c1-O-CH3 | $1.1 \cdot 10^8$ | $2.7 \cdot 10^{12}$ | 3007 | Barzaghi and Herrmann (2004) | |
| p- methoxy phenol | c1cc(O)ccc1OC | c1HcHc(OH)cHcHc1-O-CH3 | $2.8 \cdot 10^9$ | $1.8 \cdot 10^{10}$ | 698 | TROPOS measurements | |
| 2,6-dimethoxy phenol | c1(OC)cccc(OC)c1O | c1HcHcHc(-O-CH3)c(OH)c1-O-CH3 | $1.6 \cdot 10^9$ | $1.0 \cdot 10^{12}$ | 1924 | TROPOS measurements | |
| 3-methoxy-4-hydroxy benzaldehyde | c1c(OC)c(O)ccc1C=O | CH3-O-c1c(OH)cHcHc(c1H)CHO | $1.1 \cdot 10^9$ | $7.8 \cdot 10^{11}$ | 1924 | TROPOS measurements | |
| 2,4-dimethoxy-3-hydroxy benzaldehyde | c1c(OC)c(O)c(OC)cc1C=O | CH3-O-c1c(OH)c(-O-CH3)cHc(c1H)CHO | $1.7 \cdot 10^9$ | $2.8 \cdot 10^{12}$ | 2165 | TROPOS measurements | |
| 4-methoxy benzoic acid | c1cc(OC)ccc1C(=O)O | CH3-O-c1cHcHc(CO(OH))cHc1H | $6.9 \cdot 10^8$ | | | Umschlag et al. (2002) | |
| 4-methoxy benzoate | c1cc(OC)ccc1C(=O)[O-] | | $8.0 \cdot 10^9$ | | | O'Neill et al. (1977)[7] | |
| 3-hydroxy-4-methoxy benzoic acid | c1c(O)c(OC)ccc1C(=O)O | CH3-O-c1c(OH)cHc(CO(OH))cHc1H | $1.0 \cdot 10^9$ | | | TROPOS measurements | |
| 3-methoxy-4-hydroxy benzoic acid | c1c(OC)c(O)ccc1C(=O)O | CH3-O-c1c(OH)cHcHc(c1H)CO(OH) | $1.0 \cdot 10^9$ | $3.8 \cdot 10^{11}$ | 1804 | TROPOS measurements | |
| 3,5-dimethoxy-4-hydroxy benzoic acid | c1c(OC)c(O)c(OC)cc1C(=O)O | CH3-O-c1c(OH)c(-O-CH3)cHc(c1H)CO(OH) | $1.4 \cdot 10^9$ | $2.8 \cdot 10^{12}$ | 2285 | TROPOS measurements | |
| 4-hydroxyphenyl ethyl ether | c1cc(O)ccc1OCC | c1HcHc(OH)cHcHc1-O-CH2CH3 | $8.0 \cdot 10^8$ | | | Barzaghi and Herrmann (2004) | |
| **Polycyclic aromatic compounds** | | | | | | | n = 2 |
| 2-phenylphenol | c1ccccc1c2ccccc2O | c2HcHcHcHc(OH)c2c1cHcHcHcHc1H | $2.4 \cdot 10^8$ | | | Barzaghi and Herrmann (2004) | |
| naphthalene | c1cccc2ccccc12 | c12cHcHcHcHc1cHcHcHc2H | $1.7 \cdot 10^9$ | | | TROPOS measurements | |

Remarks: References taken from: [1]Toyota et al. (2004); [2]CAPRAM database available at http://projects.tropos.de/capram/; [3]NIST database: Ross et al. (1998); [4] de Sémainville et al. (2007); [5]Herrmann (2003); [6] Wayne et al. (1991); [7]Barzaghi and Herrmann (2004)

**S2 Additional information about the evaluation of prediction methods for radical reactions with organic compounds**

This section mainly presents additional tabulated statistical data and graphics to validate the different methods to predict aqueous phase kinetic data of organic compounds with the radicals OH and $NO_3$. Simple correlations are discussed in sections S2.1 to S2.3, while Evans-Polanyi-type correlations and structure activity relationships are discussed in the main article with additional information on Evans-Polanyi-type correlations in section S2.4.

**S2.1 Gas–aqueous phase correlations**

**S2.1.1 OH rate constant prediction**

In a first and very simple attempt, the reactivity of the hydroxyl radical in the gas phase and the aqueous phase have been compared. This method is very easy to use and can be readily implemented in computer assisted rate prediction tools. Although gas phase rate constants are usually quite well known and several recommendations with large sets of experimental values already exist (Atkinson, 2003; Atkinson et al., 2006; Burkholder et al., 2015), the method relies on gas phase kinetic data. If no experimental data is present, the method fails or has to be extended by estimated gas phase kinetic data from a different prediction method, which then increases errors.

For completeness, correlations between the experimental aqueous phase rate constants and the experimental gas phase rate constants are shown in Figure S1 for the different compound classes. In this and all other plots, data points from the various compound classes are colour-coded with the same colours. Regression lines are shown in the same colour as the corresponding data points and the 1:1 line is given as a dashed-dotted grey line. Regression lines of overall data are shown as black solid lines. The corresponding equations for the regression lines in Figure S1 and the statistical data can be found in Table S1. The aqueous phase data used for the correlations is given in Table S1. Gas phase rate constants are taken from Atkinson et al. (2006) and Atkinson (2003) or from the GECKO-A gas phase kinetics database and are listed in Table S2.

Even if this approach is simple, coefficients of determination of up to 0.9 are reached for the simplest compound class of alkanes. For the other compound classes, the coefficients of determination lie between 0.4 and 0.6. The only exception is found for unsaturated hydrocarbons, which do not correlate indicated by an $R^2$ of only 0.04. The poor correlation might result from a different reaction mechanism of this compound class. While all other compound classes investigated react by H-atom abstraction, unsaturated organic compounds prefer radical addition to the double bond – apparently, this step is being influenced by the solvent. Overall, it can be expected to obtain at least the correct order of magnitude of the predicted kinetic data with this prediction method when restricted to H-abstraction reactions.

**Table S1** Parameters for the regression lines $k_{aq}\,/\,M^{-1}\,s^{-1} = A \cdot k_g\,/\,cm^3\,s^{-1} + B$, coefficients of determination $R^2$, standard deviations $\sigma$, and number $N$ of the correlation of the OH rate constants in the gas and the aqueous phase as given in Figure S1.

| Compound class | $A\,/\,M^{-1}\,cm^{-3}$ | $B\,/\,10^9\,M^{-1}\,s^{-1}$ | $R^2$ | $\sigma\,/\,10^9\,M^{-1}\,s^{-1}$ | $N$ |
|---|---|---|---|---|---|
| all data | 0.13±0.02 | 0.19±0.04 | 0.280 | 2.38 | 79 |
| alkanes | 1.08±0.11 | 0.10±0.05 | 0.911 | 0.87 | 11 |
| unsat. HCs | 0.04±0.05 | 0.52±0.18 | 0.085 | 2.91 | 10 |
| alcohols | 0.37±0.09 | 0.04±0.09 | 0.557 | 1.54 | 15 |
| carbonyls | 0.14±0.06 | 0.10±0.09 | 0.251 | 2.30 | 17 |
| acids | 0.29±0.28 | 0.01±0.02 | 0.514 | 0.11 | 3 |
| ethers | 0.10±0.04 | 0.10±0.06 | 0.396 | 0.95 | 11 |
| esters | 0.30±0.12 | 0.03±0.03 | 0.388 | 0.61 | 12 |

[Figure]

[Figure]

**Figure S1** Correlation of aqueous phase rate constants of the hydroxyl radicals with aliphatic organic compounds with the corresponding gas phase rate constants distinguished by compound class. The equations for the regression lines as well as the corresponding $R^2$-, $\sigma$-, and $N$-values are given in Table S1.

**Table S2** Compilation of gas phase kinetic data of hydroxyl radical reactions with organic compounds used for the correlation with the respective aqueous phase reactions.

| Compound | Chemical formula | $k_{2nd}$ / $cm^3$ $molec^{-1}$ $s^{-1}$ | Reference |
|---|---|---|---|
| Methane | $CH_4$ | $6.4 \cdot 10^{-15}$ | Atkinson et al. (2006) |
| Ethane | $CH_3CH_3$ | $2.4 \cdot 10^{-13}$ | Atkinson et al. (2006) |
| Propane | $CH_3CH_2CH_3$ | $1.1 \cdot 10^{-12}$ | Atkinson et al. (2006) |
| Butane | $CH_3[CH_2]_2CH_3$ | $2.3 \cdot 10^{-12}$ | Atkinson et al. (2006) |
| Pentane | $CH_3[CH_2]_3CH_3$ | $3.8 \cdot 10^{-12}$ | Atkinson and Arey (2003) |
| Hexane | $CH_3[CH_2]_4CH_3$ | $5.2 \cdot 10^{-12}$ | Atkinson and Arey (2003) |
| Heptane | $CH_3[CH_2]_5CH_3$ | $6.8 \cdot 10^{-12}$ | Atkinson and Arey (2003) |
| Octane | $CH_3[CH_2]_6CH_3$ | $8.1 \cdot 10^{-12}$ | Atkinson and Arey (2003) |
| Iso-butane | $CH(CH_3)_3$ | $2.1 \cdot 10^{-12}$ | Atkinson and Arey (2003) |
| 2-methyl butane | $CH_3CH_2CH(CH_3)_2$ | $3.6 \cdot 10^{-12}$ | Atkinson and Arey (2003) |
| 2,2,4-trimethyl pentane | $(CH_3)_3CCH_2CH(CH_3)_2$ | $3.3 \cdot 10^{-12}$ | Atkinson and Arey (2003) |
| Ethylene | $CH_2=CH_2$ | $7.9 \cdot 10^{-12}$ | Atkinson et al. (2006) |
| Propylene | $CH_3CH=CH_2$ | $2.9 \cdot 10^{-11}$ | Atkinson et al. (2006) |
| 1-butylene | $CH_3CH_2CH=CH_2$ | $3.1 \cdot 10^{-11}$ | Atkinson and Arey (2003) |
| butadiene | $CH_2=CHCH=CH_2$ | $6.7 \cdot 10^{-11}$ | Atkinson and Arey (2003) |
| Isobutylene | $CH_2=C(CH_3)_2$ | $5.1 \cdot 10^{-11}$ | Atkinson and Arey (2003) |
| Acetylene | $CH\equiv CH$ | $7.8 \cdot 10^{-13}$ | Atkinson et al. (2006) |
| Methanol | $CH_3(OH)$ | $9.0 \cdot 10^{-13}$ | Atkinson et al. (2006) |
| Ethanol | $CH_3CH_2(OH)$ | $3.2 \cdot 10^{-12}$ | Atkinson et al. (2006) |
| Propanol | $CH_3CH_2CH_2(OH)$ | $5.8 \cdot 10^{-12}$ | Atkinson et al. (2006) |
| Butanol | $CH_3[CH_2]_2CH_2(OH)$ | $8.5 \cdot 10^{-12}$ | Atkinson et al. (2006) |
| Pentanol | $CH_3[CH_2]_3CH_2(OH)$ | $1.1 \cdot 10^{-11}$ | Atkinson and Arey (2003) |
| Hexanol | $CH_3[CH_2]_4CH_2(OH)$ | $1.5 \cdot 10^{-11}$ | Atkinson and Arey (2003) |

| | | | |
|---|---|---|---|
| Heptanol | $CH_3[CH_2]_5CH_2(OH)$ | $1.4 \cdot 10^{-11}$ | Atkinson and Arey (2003) |
| Octanol | $CH_3[CH_2]_6CH_2(OH)$ | $1.4 \cdot 10^{-11}$ | Atkinson and Arey (2003) |
| Isobutanol | $(CH_3)_2CH(OH)$ | $5.1 \cdot 10^{-12}$ | Atkinson et al. (2006) |
| 2-butanol | $CH_3CH_2CH(OH)CH_3$ | $8.7 \cdot 10^{-12}$ | Atkinson et al. (2006) |
| Tert-butanol | $(CH_3)_3C(OH)$ | $1.1 \cdot 10^{-12}$ | Atkinson and Arey (2003) |
| 3-pentanol | $CH_3CH_2CH(OH)CH_2CH_3$ | $1.3 \cdot 10^{-11}$ | Atkinson and Arey (2003) |
| Ethylene glycol | $CH_2(OH)CH_2(OH)$ | $7.7 \cdot 10^{-12}$ | Atkinson (1989) |
| 1,2-propanediol | $CH_3CH(OH)CH_2(OH)$ | $1.2 \cdot 10^{-11}$ | Atkinson (1989) |
| Formaldehyde | HCHO | $8.5 \cdot 10^{-12}$ | Atkinson et al. (2006) |
| Acetaldehyde | $CH_3CHO$ | $1.5 \cdot 10^{-11}$ | Atkinson et al. (2006) |
| Propionaldehyde | $CH_3CH_2CHO$ | $2.0 \cdot 10^{-11}$ | Atkinson et al. (2006) |
| Butyraldehyde | $CH_3[CH_2]_2CHO$ | $2.4 \cdot 10^{-11}$ | Atkinson et al. (2006) |
| Valeraldehyde | $CH_3[CH_2]_3CHO$ | $2.8 \cdot 10^{-11}$ | Atkinson and Arey (2003) |
| Isobutyraldehyde | $(CH_3)_2CHCHO$ | $2.6 \cdot 10^{-11}$ | Atkinson and Arey (2003) |
| Acrolein | $CH_2=CHCHO$ | $2.0 \cdot 10^{-11}$ | Magneron et al. (2002) |
| Methacrolein | $CH_2=CH(CH_3)CHO$ | $2.9 \cdot 10^{-11}$ | Atkinson et al. (2006) |
| Crotonaldehyde | $CH_3CH=CHCHO$ | $3.5 \cdot 10^{-11}$ | Magneron et al. (2002) |
| Acetone | $CH_3COCH_3$ | $1.8 \cdot 10^{-13}$ | Atkinson et al. (2006) |
| Hydroxyacetone | $CH_3COCH_2(OH)$ | $3.0 \cdot 10^{-12}$ | Atkinson et al. (2006) |
| Methyl ethyl ketone | $CH_3CH_2COCH_3$ | $1.2 \cdot 10^{-12}$ | Atkinson et al. (2006) |
| Methyl propyl ketone | $CH_3CH_2CH_2COCH_3$ | $4.4 \cdot 10^{-12}$ | Atkinson and Arey (2003) |
| Diethyl ketone | $CH_3CH_2COCH_2CH_3$ | $2.0 \cdot 10^{-12}$ | Atkinson and Arey (2003) |
| Methyl vinyl ketone | $CH_2=CHCOCH_3$ | $2.0 \cdot 10^{-11}$ | Atkinson et al. (2006) |
| Methyl isobutyl ketone | $CH_3COCH_2CH(CH_3)_2$ | $1.3 \cdot 10^{-11}$ | Atkinson and Arey (2003) |
| Glyoxal | CHOCHO | $1.1 \cdot 10^{-11}$ | Atkinson et al. (2006) |
| Methylglyoxal | $CH_3COCOCH_3$ | $1.5 \cdot 10^{-11}$ | Atkinson et al. (2006) |
| Diacetyl | $CH_3COCOCH_3$ | $2.4 \cdot 10^{-13}$ | Atkinson (1989) |
| Acetylacetone | $CH_3COCH_2COCH_3$ | $1.2 \cdot 10^{-11}$ | Atkinson (1989) |
| Acetonyl acetone | $CH_3COCH_2CH_2COCH_3$ | $6.8 \cdot 10^{-13}$ | Atkinson (1989) |
| Formic acid | HC(O)OH | $4.5 \cdot 10^{-13}$ | Atkinson et al. (2006) |
| Acetic acid | $CH_3C(O)OH$ | $7.4 \cdot 10^{-13}$ | Atkinson et al. (2006) |
| Propionic acid | $CH_3CH_2C(O)OH$ | $1.2 \cdot 10^{-12}$ | Atkinson et al. (2006) |
| Dimethyl ether | $CH_3–O–CH_3$ | $2.8 \cdot 10^{-12}$ | Atkinson et al. (2006) |
| Diethyl ether | $CH_3CH_2–O–CH_2CH_3$ | $1.3 \cdot 10^{-11}$ | Atkinson and Arey (2003) |
| Methylal | $CH_3–O–CH_2–O–CH_3$ | $4.9 \cdot 10^{-12}$ | Carter (2010) |
| Diethoxy methane | $CH_3CH_2–O–CH_2–O–CH_2CH_3$ | $1.8 \cdot 10^{-11}$ | Thüner et al. (1999) |
| Ethylene glycol methyl ether | $CH_3–O–CH_2CH_2(OH)$ | $1.3 \cdot 10^{-11}$ | Carter (2010) |
| Ethylene glycol ethyl ether | $CH_3CH_2–O–CH_2CH_2(OH)$ | $1.9 \cdot 10^{-11}$ | Carter (2010) |
| Ethylene glycol butyl ether | $CH_3[CH_2]_3–O–CH_2CH_2(OH)$ | $2.6 \cdot 10^{-11}$ | Carter (2010) |
| Methyl tert-butyl ether | $CH_3–O–C(CH_3)_3$ | $3.2 \cdot 10^{-12}$ | Teton et al. (1996) |
| Ethyl tert-butyl ether | $CH_3CH_2–O–C(CH_3)_3$ | $8.9 \cdot 10^{-12}$ | Teton et al. (1996) |
| Di-isopropyl ether | $(CH_3)_2CH–O–CH(CH_3)_2$ | $1.0 \cdot 10^{-11}$ | Mellouki et al. (1995) |

| Di-tert-butyl ether | $(CH_3)_3C–O–C(CH_3)_3$ | $3.9 \cdot 10^{-12}$ | Average of Nielsen et al. (1995) and Langer et al. (1996) |
|---|---|---|---|
| Ethyl formate | $CH_3CH_2–O–CHO$ | $1.0 \cdot 10^{-12}$ | Carter (2010) |
| Butyl formate | $CH_3[CH_2]_3–O–CHO$ | $3.1 \cdot 10^{-12}$ | Carter (2010) |
| Tert-butyl formate | $(CH_3)_3C–O–CHO$ | $7.5 \cdot 10^{-13}$ | Le Calvé et al. (1997c) |
| Methyl acetate | $CH_3CO–O–CH_3$ | $3.4 \cdot 10^{-13}$ | El Boudali et al. (1996) |
| Methyl propionate | $CH_3CH_2CO–O–CH_3$ | $1.0 \cdot 10^{-12}$ | Carter (2010) |
| Ethyl acetate | $CH_3CH_2–O–COCH_3$ | $1.6 \cdot 10^{-12}$ | Carter (2010) |
| Ethyl propionate | $CH_3CH_2CO–O–CH_2CH_3$ | $2.14 \cdot 10^{-12}$ | Wallington et al. (1988) |
| Propyl acetate | $CH_3[CH_2]_2–O–COCH_3$ | $3.6 \cdot 10^{-12}$ | El Boudali et al. (1996) |
| Methyl butyrate | $CH_3[CH_2]_2CO–O–CH_3$ | $3.4 \cdot 10^{-12}$ | Le Calvé et al. (1997a) |
| Butyl acetate | $CH_3[CH_2]_3–O–COCH_3$ | $5.7 \cdot 10^{-12}$ | El Boudali et al. (1996) |
| Ethyl butyrate | $CH_3[CH_2]_2CO–O–CH_2CH_3$ | $0$ | Ferrari et al. (1996) |
| Isopropyl acetate | $(CH_3)_2CH–O–COCH_3$ | $4.0 \cdot 10^{-12}$ | Le Calvé et al. (1997b) |
| Dimethyl succinate | $CH_3–O–CO[CH_2]_2CO–O–CH_3$ | $1.5 \cdot 10^{-12}$ | Carter (2010) |

**S2.1.2 NO₃ rate constant prediction**

This method is not suitable for $NO_3$ rate constant prediction. There are only few species for which $NO_3$ rate constants exist in both the gas and the aqueous phase. Correlations can only be derived for limited compound classes with a distinct structure and the method is not suitable to predict the whole range of organic compounds relevant for tropospheric chemistry as needed by mechanism auto-generation.

**S2.2 Homologous series of compound classes**

**S2.2.1 OH rate constant prediction**

Linear extrapolations or other suitable regression methods can be used to extrapolate the abundant measurements of small molecules in a homologous series to larger carbon numbers. However, the method is restricted to very defined structures such as alkanes or alkanes with a terminal functionality.

For proof of concept, the method was first tested for alkanes despite their irrelevance for aqueous phase chemistry. Even for pure hydrocarbons, the restrictions of the method can be seen. Figure S2a shows the regression of the homologous series of alkanes. The dataset demonstrates the necessity for a distinction by chemical structure and two regression lines can be found: one for linear (with 8 data points) and one for branched alkanes (with 4 data points). The regression lines, however, show a high accuracy with coefficients of determination $R^2$ of 0.98 and 0.97, respectively.

Figure S2b shows the data for mono-alcohols, which underlines the importance of the structure of the molecules. The rate constants of unsaturated alcohols have a very poor correlation with the carbon number and mono-alcohols with internal hydroxyl groups seem to be totally uncorrelated. Only the regression of linear mono-alcohols with terminal hydroxyl groups shows satisfactory results, which is, therefore, the only regression line for mono-alcoholic compounds shown in Figure S2b. The data of linear alcohols points at another general problem in the data analysis – data selection. In Figure S2b, the last three data points show a different slope in the regression than the first five data points. Data for the C₅ to C₈ alcohols were measured by Scholes and Willson (1967) while the other data are from different sources. It is hard to evaluate whether the different trends derive from different experimental setups or are the consequence of other effects like the approach of the reaction limit. Using separate correlations for the n-alcohol series split between C₅ and C₆ (dashed lines in Figure S2b) instead of the overall trend

(solid line) would result in predicted rate constant of a $C_{12}$ alcohol that is only 74% of that predicted by the overall trend. This demonstrates that careful evaluation is necessary as well as a revision how far extrapolations can be extended to not violate other rules like the limits of reaction.

[Figure]

Figure S2 Homologous series of OH reaction rate constants for various compound classes. Correlation lines are shown in the same colour as the data points. The dashed lines in subfigure b are separate correlations for $C_1$ to $C_5$ and $C_6$ to $C_8$ compounds. Dashed lines in subfigure d refer to correlations without formic acid/formate. In subfigure e, the pink line represents the best fit of the Markov distribution of the fully protonated acids and the grey line is the fit of all protonation states. The respective equations for the regression lines (including the Markov distributions) can be found in in Table S3.

Table S3 Equations of the regression lines, coefficients of determination $R^2$, standard deviations $\sigma$ (or $\chi^2$ for DCAs), and number $N$ for the correlation of the OH reactivity over the carbon number.

| Compound class | Regression | $R^2$ | $\sigma$ / $M^{-1}$ $s^{-1}$ $\chi^{2\,a}$ | $N$ |
|---|---|---|---|---|

Field Code Changed

| | | | | |
|---|---|---|---|---|
| Linear alkanes | $k_{2nd} = (1.38\pm0.08)\cdot10^9 \cdot CN - (1.9\pm0.4)\cdot10^9$ | 0.980 | $5.2\cdot10^8$ | 8 |
| Branched alkanes | $k_{2nd} = (3.7\pm0.4)\cdot10^8 \cdot CN + (3.2\pm0.3)\cdot10^9$ | 0.971 | $1.4\cdot10^8$ | 4 |
| Terminal alcohols | $k_{2nd} = (1.02\pm0.06)\cdot10^9 \cdot CN + (7\pm33)\cdot10^7$ | 0.977 | $4.2\cdot10^8$ | 8 |
| Terminal $C_1$ - $C_5$ alcohols | $k_{2nd} = (9.9\pm0.3)\cdot10^8 \cdot CN + (9\pm12)\cdot10^7$ | 0.996 | $1.1\cdot10^8$ | 5 |
| Terminal $C_6$ - $C_8$ alcohols | $k_{2nd} = (3.5\pm0.3)\cdot10^8 \cdot CN + (4.9\pm0.2)\cdot10^9$ | 0.993 | $4.1\cdot10^7$ | 3 |
| Diols | $k_{2nd} = (7.7\pm0.7)\cdot10^8 \cdot CN + (3\pm3)\cdot10^8$ | 0.975 | $2.2\cdot10^8$ | 5 |
| Monoacids | $k_{2nd} = (5\pm2)\cdot10^8 \cdot CN - (1\pm9)\cdot10^8$ | 0.378 | $1.2\cdot10^9$ | 10 |
| Undissociated monoacids | $k_{2nd} = (7\pm3)\cdot10^8 \cdot CN - (9.6\pm9)\cdot10^8$ | 0.667 | $7.3\cdot10^8$ | 4 |
| Dissociated monoacids | $k_{2nd} = (4\pm4)\cdot10^8 \cdot CN + (8\pm13)\cdot10^8$ | 0.236 | $1.5\cdot10^9$ | 6 |
| Monoacids (woC$_1$) | $k_{2nd} = (1.03\pm0.08)\cdot10^9 \cdot CN - (2.2\pm0.3)\cdot10^9$ | 0.966 | $2.9\cdot10^8$ | 8 |
| Undissociated monoacids (woC$_1$) | $k_{2nd} = (1.09\pm0.5)\cdot10^9 \cdot CN - (2.4\pm1)\cdot10^9$ | 0.852 | $6.4\cdot10^8$ | 3 |
| Dissociated monoacids (woC$_1$) | $k_{2nd} = (1.00\pm0.05)\cdot10^9 \cdot CN - (2.1\pm0.2)\cdot10^9$ | 0.993 | $1.6\cdot10^8$ | 5 |
| DCAs | $k_{2nd} = \dfrac{(4\pm15)\cdot10^7 - (6.8\pm0.3)\cdot10^9}{1+\exp\{CN-(6.0\pm0.1)/(0.8\pm0.1)\}} + (6.8\pm0.3)\cdot10^9$ | 0.958 | $2.2\cdot10^{17}$ | 21 |
| Undissociated DCAs | $k_{2nd} = \dfrac{(2\pm2)\cdot10^8 - (6.5\pm0.5)\cdot10^9}{1+\exp\{CN-(5.9\pm0.2)/(0.8\pm0.1)\}} + (6.5\pm0.5)\cdot10^9$ | 0.995 | $4.9\cdot10^{16}$ | 9 |

[a]for DCAs a Boltzmann distribution is used. For these compound classes $\chi^2$ is given instead of σ.

Another group of alcohols for which the extrapolation of a homologous series works well are terminal diols with an alcohol function at each end of the molecule. Only minimal absolute errors are observed as can be seen from Figure 1 in the main article.

5 Furthermore, the method was tested for linear carboxylic acids. Results are given in Figure S2d and Table S3. These data show no big differences for the rate constants for protonated and deprotonated acids, respectively. Furthermore, for both forms the $C_1$ acid forms an exception with higher rate constants than the ones expected from a linear regression. This effect is much stronger for the deprotonated forms. When omitting formic acid, however, the good performance of the prediction method can

10 be confirmed with a coefficient of determination $R^2$ of 0.97 for the regression line using both, the dissociated and undissociated forms. When investigating both forms separately, the deprotonated form correlates better with a $R^2$ of 0.99 compared to $R^2 = 0.85$ of the protonated form.

Interesting effects are observed for the correlation of linear dicarboxylic acids (DCAs). The dataset for all three forms, the protonated forms, the mono-anions, and the di-anions, is shown in Figure S2e. From this Figure it can be seen that no linear

15 regression is valid for DCAs. There is only a moderate, non-linear increase in the rate constants for DCAs with less than 7 carbon atoms. Rate constants stay below $2\cdot10^9$ M$^{-1}$ s$^{-1}$. Then the rate constants jump to values above $5\cdot10^9$ M$^{-1}$ s$^{-1}$ for DCAs with more than 7 carbon atoms and do, again, only moderately increase. For such a distribution, a sigmoidal function can be used and the Markov distribution has been applied. There is again a good correlation with a $R^2$ of 0.96 for the overall data.

It can only be speculated on the reason why a linear correlation does not apply to dicarboxylic acids. A likely explanation is

20 the special structure of this compound class. The polarity and the large diffusion volume of the carboxyl group may shield the inner methylene groups in the carbon skeleton and hinder any radical attack for small DCAs. If a certain size of the carbon skeleton is reached (i.e. $C_7$), an unproportional increase of the reaction rate constant is seen to values without the shielding effect. These rate constants of large DCAs are close to the diffusion limits of hydroxyl radical reactions (see Schöne et al., 2014), thus, only a moderate increase is observed afterwards. This effect was seen for all dissociation states of carboxylic

25 acids, therefore, different reaction mechanisms such as electron transfer reactions (ETRs) are an unlikely explanation for this effect.

**Moved (insertion) [2]**

**Moved up [1]:** Scholes and Willson (1967)

**Moved up [2]:** Another group of alcohols for which the extrapolation of a homologous series works well are terminal diols with an alcohol function at each end of the molecule. Only minimal absolute errors are observed as can be seen from Figure 1 in the main

**S2.2.2 NO₃ rate constant prediction**

Homologous series are, again, impractical to predict NO₃ rate constants due to the limited experimental dataset. The dataset allowed the determination of regression lines only for several alcohols and for aldehydes. However, for the regressions no more than 8 data points could be used (see Table S4). An exception is the correlation of all alcohols including linear, branched

5   and substituted saturated mono-alcohols and gem-diols. However, the correlation is weak with a coefficient of determination $R^2$ of 0.5. All correlation data can be found in Table S4. Moreover, Figure S3a gives the dataset and regression lines of the NO₃ reaction rate constants of various alcohols plotted against the carbon number. Further compound classes show a bad correlation with the exception of aldehydes with a coefficient of determination $R^2$ of 0.85 as shown in Figure S3b and Table S4.

[Figure]

**Figure S3 Homologous series of NO₃ reaction rate constants of various compound classes as indicated in the figure legend. The respective equations for the regression lines can be found in Table S4.**

**Table S4 Equations of the regression lines, coefficients of determination $R^2$, standard deviations $\sigma$, and number $N$ for the correlation**
20  **of the NO₃ reactivities over the carbon number.**

| Compound class | Regression | $R^2$ | $\sigma$ / M⁻¹ s⁻¹ | N |
|---|---|---|---|---|
| Linear monoalcohols | $k_{2nd} = (5.5\pm1.3)\cdot10^5 \cdot CN + (3.6\pm6.6)\cdot10^5$ | 0.748 | $8.4\cdot10^5$ | 8 |
| - C1 – C3 | $k_{2nd} = (1.33\pm0.02)\cdot10^6 \cdot CN - (7.8\pm0.4)\cdot10^5$ | 1.000 | $2.4\cdot10^4$ | 3 |
| - C4 – C8 | $k_{2nd} = (9.0\pm1.8)\cdot10^5 \cdot CN - (2.0\pm1.1)\cdot10^6$ | 0.894 | $5.7\cdot10^5$ | 5 |
| - with substitutions/ gem-diols | $k_{2nd} = (5.0\pm1.5)\cdot10^5 \cdot CN + (4.7\pm6.6)\cdot10^5$ | 0.501 | $1.1\cdot10^6$ | 13 |
| Polyols | $k_{2nd} = (3.0\pm0.9)\cdot10^6 \cdot CN - (7.1\pm34.2)\cdot10^5$ | 0.746 | $3.7\cdot10^6$ | 6 |
| - sugars | $k_{2nd} = (4.0\pm3.5)\cdot10^5 \cdot CN + (1.2\pm0.2)\cdot10^7$ | 0.400 | $7.7\cdot10^5$ | 4 |
| Aldehydes | $k_{2nd} = (2.2\pm0.5)\cdot10^7 \cdot CN - (2.4\pm1.6)\cdot10^7$ | 0.853 | $1.4\cdot10^7$ | 5 |

The regression lines from Table S6 have been used to predict the rate constants of the nitrate radical with organic compounds. The absolute errors have been determined and plotted as box plots in Figure 1 of the main article. Although the errors are

25   relatively small, it is not recommended to use this method. The dataset to derive the regression lines is small, and possible

(a) Alcohol compounds

changes in only a few experimental values could lead to substantial changes in the predicted rate data. With no use of separate training and validation sets, correlations are too uncertain for any computer-based mechanism auto-generation protocols.

**S2.3 Reactivity comparison between OH and NO₃**

The kinetics database (presented in and ) has been used to compare the reactivities of the hydroxyl and nitrate radicals. Results from the evaluation are shown in Figure S4. In general, the reactivity of NO₃ is smaller than that of OH. On average, the decrease in the reactivity is about 2 orders of magnitude as can be seen from the data points as well as the correlation line of all data (dashed violet line) in Figure S4. The only exception is oxalic acid, where the electron transfer of oxalate with the nitrate radical is so fast that the reactivity of the mono-anion is close to the reactivity of the corresponding OH reaction and that of the di-anion is even higher than the corresponding OH reaction.

The regression lines (see also Table S5) can be used to predict the reaction rates of the nitrate radical when the aqueous OH reactivities are known. However, with coefficients of determination $R^2$ between 0.21 and 0.89 and 0.06 for the overall data, the quality of the prediction is weak. Moreover, the method is restricted to the dataset of the hydroxyl reactions in the aqueous phase and therefore unsuitable for automated rate prediction.

**Table S5.** Equations of the regression lines $\log(k_{NO3} / M^{-1} s^{-1}) = A \cdot \log(k_{OH} / M^{-1} s^{-1}) + B$, **coefficients of determination $R^2$, standard deviations $\sigma$, and number $N$ for the correlation of the OH and NO₃ reactivities.**

| compound class | $A$ / $M^{-1}$ $s^{-1}$ | $B$ / $M^{-1}$ $s^{-1}$ | $R^2$ | $\sigma$ | $N$ |
|---|---|---|---|---|---|
| Alcohols | 1.26±0.24 | 5.74±2.26 | 0.757 | 0.27 | 11 |
| Di- and polyols | 4.58±0.66 | 35.28±6.05 | 0.873 | 0.18 | 9 |
| Carbonyls | 1.43±0.49 | 6.33±4.44 | 0.521 | 0.89 | 10 |
| Mono-carboxylic acids | 1.05±0.22 | 2.77±2.00 | 0.598 | 0.71 | 17 |
| Di-carboxylic acids | 1.39±0.61 | 4.62±4.85 | 0.425 | 1.26 | 9 |
| All data | 0.77±0.15 | 0.33±1.39 | 0.316 | 0.89 | 56 |

[Figure]

[Figure]

**Figure S4** Correlation of the reactivities of the hydroxyl and the nitrate radical with aliphatic organic compounds as listed in the figure legend. The equations for the regression lines as well as the corresponding $R^2$-, $\sigma$-, and $N$-values are given in Table S5.

**S2.4 Evans-Polanyi-type correlations**

**S2.4.1 OH radical reactions**

[Figure]

[Figure]

**Figure S5** Evans-Polanyi-type correlations of hydroxyl radicals with the respective compound classes. In subfigure c, the dotted line represents the linear fit of ketones excluding methylglyoxal, the dashed line represents the correlation of ketones without acetylacetone. For the dashed-dotted line, both compounds have been omitted. The black dashed-dotted line in subfigure c is the overall correlation of all carbonyl compounds except glyoxal and formaldehyde. The respective equations for the regression lines can be found in Table S6.

**Table S6** Parameters for the regression equations $\log(k_H/\text{M}^{-1}\,\text{s}^{-1}) = A\cdot(BDE/\text{kJ mol}^{-1}) + B$ and statistical data derived from the Evans-Polanyi-type correlations of aqueous phase hydroxyl radical reactions with organic compounds for the various compound classes.

| Compound class | $10^2\cdot A^{(a)}$ | $10^{-1}\cdot B^{(b)}$ | $R^2$ | $\sigma$ / $\text{M}^{-1}\,\text{s}^{-1}$ | N |
|---|---|---|---|---|---|
| Alkanes | -(3.86±1.1) | 2.49±0.4 | 0.595 | 0.24 | 11 |
| Linear terminal monoalcohols | -(7.56±1.9) | 3.95±0.7 | 0.732 | 0.20 | 8 |
| All monoalcohols | -(1.88±0.8) | 1.67±0.3 | 0.315 | 0.28 | 15 |
| Diols | -(0.65±0.5) | 1.15±0.2 | 0.126 | 0.22 | 13 |
| Diols and sugars | -(1.41±0.3) | 1.44±0.1 | 0.439 | 0.28 | 25 |
| Monoaldehydes | 2.08±0.4 | 0.20±0.1 | 0.852 | 0.13 | 7 |
| Ketones and diketones | -(2.21±1.2) | 1.75±0.5 | 0.254 | 0.60 | 12 |
| - without methylglyoxal | -(4.49±1.7) | 2.67±0.7 | 0.444 | 0.54 | 11 |
| - without acetylacetone | -(2.67±1.0) | 1.92±0.4 | 0.453 | 0.48 | 11 |
| - without both | -(5.21±1.1) | 2.94±0.4 | 0.745 | 0.35 | 10 |
| Carbonyl compounds (without glyoxal and formaldehyde) | -(2.21±0.6) | 1.75±0.2 | 0.497 | 0.48 | 18 |
| Monocarboxylic acids (without formic acid) | -(4.37±1.3) | 2.60±0.5 | 0.573 | 0.52 | 10 |
| All dicarboxylic acids | -(3.29±1.6) | 2.14±0.6 | 0.257 | 0.88 | 14 |
| Unsubstituted carboxylic acids | -(8.93±2.8) | 4.44±1.1 | 0.601 | 0.78 | 5 |

[a] in l kJ$^{-1}$ s$^{-1}$, [b] in M$^{-1}$ s$^{-1}$

**5  2.4.2 NO₃ radical reactions**

**Table S7** Parameters for the regression equations $\log(k_H/\text{M}^{-1}\,\text{s}^{-1}) = A\cdot(BDE/\text{kJ mol}^{-1}) + B$ and statistical data derived from the Evans-Polanyi-type correlations of aqueous phase nitrate radical reactions with organic compounds for the various compound classes.

| Compound class | $10^2\cdot A^{(a)}$ | $10^{-1}\cdot B^{(b)}$ | $R^2$ | $\sigma$ / $\text{M}^{-1}\,\text{s}^{-1}$ | N |
|---|---|---|---|---|---|
| Monoalcohols | -(6.91±1.8) | 3.35±0.7 | 0.628 | 0.49 | 11 |
| Diols | -(2.76±0.7) | 1.70±0.3 | 0.680 | 0.35 | 9 |
| All alcohols | -(3.45±0.6) | 1.96±0.3 | 0.616 | 0.47 | 20 |
| Carbonyl compounds | -(2.89±1.7) | 1.73±0.6 | 0.256 | 1.41 | 10 |
| Carboxylic acids (without formic and mesoxalic acid) | -(5.65±1.7) | 2.80±0.7 | 0.698 | 0.61 | 7 |

[a] in l kJ$^{-1}$ s$^{-1}$, [b] in M$^{-1}$ s$^{-1}$

**S2.4.3 Improved Evans-Polanyi-type correlations**

[Figure]

[Figure]

**Figure S6 Improved Evans-Polanyi-type correlations of hydroxyl radical reactions with organic compounds for the respective compound classes. The grey line in subfigure b represents the quadratic regression of all alcohol compounds except linear mono-alcohols and the dashed orange line in subfigure c is the regression line of all aldehydes (including glyoxal as only dialdehyde). The respective equations for the regression lines can be found in Table S8.**

Table S8 Parameters for the regression equations $\log(k/\mathrm{M}^{-1}\,\mathrm{s}^{-1}) = A\cdot(\sum BDE/\mathrm{kJ\ mol}^{-1})^2 + B\cdot(\sum BDE/\mathrm{kJ\ mol}^{-1}) + C$ and statistical data derived from the improved Evans-Polanyi-type correlations of aqueous phase hydroxyl radical reactions with organic compounds for the various compound classes.

| Compound class | $10^8\cdot A^{(a)}$ | $10^4\cdot B^{(b)}$ | $C^{(c)}$ | $R^2$ | $\sigma$ / M$^{-1}$ s$^{-1}$ | N |
|---|---|---|---|---|---|---|
| Linear alkanes | -(2.21±0.7) | 3.90±0.7 | 8.29±0.17 | 0.986 | 0.04 | 7 |
| Branched alkanes | -(0.94±0.2) | 1.46±0.5 | 9.22±0.07 | 0.998 | 0.04 | 4 |
| All alkanes | -(3.46±1.0) | 4.92±1.1 | 8.11±0.26 | 0.923 | 0.08 | 11 |
| Linear terminal mono-alcohols | -(2.77±0.5) | 3.97±0.4 | 8.46±0.09 | 0.987 | 0.04 | 8 |
| All other alcohols* | -(2.31±2.5) | 2.84±2.0 | 8.59±0.36 | 0.346 | 0.15 | 32 |
| All alcohols | 0.55±1.2 | 0.90±1.0 | 8.91±0.22 | 0.523 | 0.17 | 40 |
| Monoaldehydes | -(6.44±3.8) | 4.39±1.8 | 8.80±1.09 | 0.810 | 0.11 | 7 |
| All aldehydes | -(17.5±14.5) | 11.2±6.6 | 7.81±0.61 | 0.630 | 0.43 | 8 |
| Ketones | -(1.67±14.3) | 3.49±9.5 | 8.14±1.50 | 0.238 | 0.46 | 10 |
| All carbonyl compounds | -(6.19±5.2) | 5.48±3.1 | 8.14±0.43 | 0.288 | 0.41 | 23 |
| Monocarboxylic acids | -(6.62±11.1) | 7.49±6.6 | 7.19±0.86 | 0.548 | 0.46 | 12 |
| Di- and polycarboxylic acids | -(4.70±4.7) | 7.81±3.9 | 6.65±0.66 | 0.665 | 0.66 | 14 |
| All carboxylic acids | -(4.33±3.3) | 7.04±2.6 | 6.97±0.44 | 0.602 | 0.57 | 26 |

$^{(a)}$ in mol l kJ$^{-2}$ s$^{-1}$, $^{(b)}$ in l kJ$^{-1}$ s$^{-1}$, $^{(c)}$ in M$^{-1}$ s$^{-1}$

*except outliers neopentanol and pinacol

[Figure]

[Figure]

Figure S7 Plot of predicted versus experimental data for hydroxyl radical reactions separated by compound class together with the linear regression lines. For the calculation of the predicted values, the regression lines of Table S8 together with the kinetic data from have been used. For ketones, the dashed line includes the outlier acetylacetone while the solid line does not. The black dashed-dotted line is the line of same reactivity.

**Table S9 Parameters for the linear regression equations** $\log(k_H/\text{M}^{-1}\,\text{s}^{-1}) = A\cdot(BDE/\text{kJ mol}^{-1}) + B$ **and statistical data derived from the improved Evans-Polanyi-type correlations of aqueous phase nitrate radical reactions with organic compounds for the various compound classes.**

| Compound class | $10^4\cdot A^{(a)}$ | $B^{(b)}$ | $R^2$ | $\sigma$ / $\text{M}^{-1}\,\text{s}^{-1}$ | N |
|---|---|---|---|---|---|
| Monoalcohols and gem-diols | $1.18\pm0.27$ | $5.87\pm0.11$ | 0.601 | 0.18 | 15 |
| Di- and polyols | $0.91\pm0.36$ | $6.71\pm0.15$ | 0.611 | 0.10 | 6 |
| Carbonyl compounds | $5.44\pm0.91$ | $5.82\pm0.23$ | 0.799 | 0.34 | 10 |
| Carboxylic acids | $-(3.92\pm4.3)$ | $5.23\pm0.78$ | 0.173 | 0.67 | 6 |

[Figure]

$^{(a)}$ in l kJ$^{-1}$ s$^{-1}$, $^{(b)}$ in M$^{-1}$ s$^{-1}$

**Table S10 Parameters for the quadratic regression equations** $\log(k_{exp}/\text{M}^{-1}\,\text{s}^{-1}) = A\cdot(\sum BDE/\text{kJ mol}^{-1})^2 + B\cdot(\sum BDE/\text{kJ mol}^{-1}) + C$ **and statistical data derived from the improved Evans-Polanyi-type correlations of aqueous phase nitrate radical reactions with organic compounds for the various compound classes.**

| Compound class | $10^8\cdot A^{(a)}$ | $10^4\cdot B^{(b)}$ | $C^{(c)}$ | $R^2$ | $\sigma$ / $\text{M}^{-1}\,\text{s}^{-1}$ | N |
|---|---|---|---|---|---|---|
| Monoalcohols and gem-diols | $-(1.85\pm1.6)$ | $2.77\pm1.4$ | $5.59\pm0.27$ | 0.639 | 0.18 | 15 |
| Di- and polyols | $-(6.71\pm2.2)$ | $6.37\pm1.8$ | $5.68\pm0.35$ | 0.905 | 0.05 | 6 |
| Carbonyl compounds | $-(3.84\pm10.0)$ | $7.16\pm4.6$ | $5.67\pm0.44$ | 0.803 | 0.36 | 10 |
| Carboxylic acids | $8.15\pm106.3$ | $-(6.71\pm36.7)$ | $5.44\pm2.83$ | 0.175 | 0.77 | 6 |

[Figure]

$^{(a)}$ in mol l kJ$^{-2}$ s$^{-1}$, $^{(b)}$ in l kJ$^{-1}$ s$^{-1}$, $^{(c)}$ in M$^{-1}$ s$^{-1}$

[Figure]

**Figure S8 Box plots for the absolute errors of the improved Evans-Polanyi-type correlations of nitrate radical reactions with the respective compound classes.**

**S2.5 Evaluation of structure-activity relationships**

**Table S11.** Parameters for the regression equations ($k_{calc}$ / M$^{-1}$ s$^{-1}$) = A· ($k_{exp}$ / M$^{-1}$ s$^{-1}$) + B and statistical data derived from the evaluation process of the structure-activity relationship by Doussin and Monod (2013).

| Compound class | $A$ | $10^{-8}·B$ | $R^2$ | $\sigma$ / $10^8$ M$^{-1}$ s$^{-1}$ | N |
|---|---|---|---|---|---|
| Alkanes | 0.80±0.08 | 4.28±4.8 | 0.908 | 6.81 | 11 |
| Monoalcohols | 0.97±0.09 | 1.52±4.0 | 0.886 | 7.73 | 16 |
| Di- and polyols | 1.40±0.21 | 0.10±5.3 | 0.701 | 10.75 | 21 |
| Carbonyl compounds[a] | 1.14±0.15 | -0.65±3.6 | 0.828 | 7.78 | 14 |
| Monocarboxylic acids | 0.96±0.10 | 1.48±1.8 | 0.869 | 4.38 | 16 |
| Dicarboxylic acids | 0.90±0.07 | 3.20±2.1 | 0.902 | 6.49 | 19 |
| Polyfunctional compounds | 0.44±0.29 | -0.004±1.7 | 0.333 | 5.22 | 24 |

[a] including dicarbonyl compounds except acetylacetone

**Table S12.** Parameters for the regression equations ($k_{calc}$ / M$^{-1}$ s$^{-1}$) = A· ($k_{exp}$ / M$^{-1}$ s$^{-1}$) + B and statistical data derived from the evaluation process of the structure-activity relationship by Minakata et al (2009).

| Compound class | $A$ | $10^{-8}·B$ | $R^2$ | $\sigma$ / $10^8$ M$^{-1}$ s$^{-1}$ | N |
|---|---|---|---|---|---|
| Alkanes | 0.98±0.11 | 4.04±6.4 | 0.891 | 9.11 | 11 |
| Monoalcohols | 0.99±0.17 | 4.04±7.5 | 0.697 | 1.49 | 16 |
| Di- and polyols | 0.78±0.36 | 21.4±9.0 | 0.201 | 18.39 | 21 |
| Carbonyl compounds[a] | 0.89±0.17 | -0.43±4.1 | 0.695 | 8.87 | 14 |
| Monocarboxylic acids | 1.09±0.18 | -1.89±3.2 | 0.726 | 7.86 | 16 |
| Dicarboxylic acids | 1.22±0.07 | 2.65±2.1 | 0.934 | 6.75 | 20 |
| Polyfunctional compounds | 0.58±0.29 | -0.11±3.7 | 0.160 | 11.48 | 23 |

[a] including dicarbonyl compounds except acetylacetone

**S3. Sensitivity runs of crucial parameters**

**S3.1 Influence of the chosen SAR**

[Figure]

5  **Figure S9 Concentration-time profiles for selected organic compounds (a – f) as well as pH value (g) and dry and organic mass (h) in the sensitivity runs investigating the influence of the chosen SAR. Model runs with the standard set of SARs (Doussin and Monod, 2013 complimented by Minakata et al., 2009, see Table 2 in the main article) are represented by red solid lines. Blue dashed lines represent model runs, where the SAR by Minakata et al. (2009) was given sole preference.**

**S3.2 Processing of the organic mass fraction**

[Figure]

[Figure]

**Figure S10** Concentration-time profile of the total particulate dry mass and the organic particulate dry mass in the base scenario 'orig'.

[Figure]

**Figure S11** Aqueous phase concentration-time profiles of OH and NO$_3$ radicals in runs introducing parameterisations for radical reactions of WSOC/HULIS in comparison to the original mechanism.

**S3.3 Influence of nitrate radical chemistry under remote conditions**

[Figure]

**(a)** Butenedial

**(b)** Fumaric acid

**(c)** Mesoxalic acid

**(d)** Oxalic acid

**(e)** 1,4-dioxo-2-hydroxy-butyl-3-nitrate

**(f)** Formyl nitrate

C4.0α ——— C4.0β -------- C4.0γ ············

**Figure S12 Concentration-time profiles of selected organic compounds calculated with different subsets of a CAPRAM test scenario under urban conditions, where the alpha version is with the full chemical scheme, the beta version without nitrate radical chemistry of unsaturated organic compounds, and the gamma version with only the nitrate radical chemistry already present in CAPRAM 3.0n (see also explanations in subsection 3.4 of the main article).**

**S3.4 Investigation of the sensitivity of cut-off parameters for minor branches**

[Figure]

**Figure S13 Overview of the reduction potential for mechanisms with different cut-off parameters for minor branches.**

[Figure]

**Figure S14 Concentration-time profiles for selected organic compounds in sensitivity runs investigating the influence of the cut-off parameter for minor product channels. Model runs have been performed under urban summer conditions with 8 non-permanent cloud periods (blue shades). Red solid lines represent concentrations of the runs with a 0.5% threshold, blue dashed lines with a 3% threshold, green dotted lines with a 5% threshold and orange solid lines with a 10% threshold.**

**S3.5 Investigations on peroxy radical chemistry**

To demonstrate the validity to neglect the recombination channels of α-hydroxy peroxy radicals, a simple simulation has been performed, where these types of peroxy radicals are oxidised with the recombination and HO₂ elimination rate constants as given in the main article. The simulation was initialised with typical radical concentrations and no sources of peroxy radicals were defined. Only two sink reactions were implemented in the run, the HO₂ elimination channel leading to a carboxyl acid with a rate constant of 1000 s$^{-1}$ and the peroxy radical recombination, where all products were combined ($k_{2nd}$ = 7.3·10$^8$ M$^{-1}$ s$^{-1}$). With typical radical concentrations of 10$^{-13}$ M and the ubiquitous availability of water ($c$ = 55.5 M), the difference in the rate constants of more than 5 orders of magnitude is still not enough to produce visible amounts of recombination products as can be seen from Figure S15. Therefore, it is safe to consider HO₂ as the only reaction pathway and reduce the size of the chemical mechanism.

[Figure]

**Figure S15 Concentration-time profiles for the aqueous phase decay of α-hydroxy peroxy radicals and the build-up of their oxidation products.**

**S4 Additional information about the LEAK chamber runs**

Table S13 Additional uptake processes and initial aqueous phase reactions used in the scenarios UPT and RXN. Scenario RXN$_{0.4}$ is the same as RXN, but with reduced WSCO yields by a factor of 0.4.

| Process[a] | UPT[b] | RXN/RXN$_{0.4}$ [c] |
|---|---|---|
| (g) ⇌ | 9.6·10$^{-2}$ | 9.6·10$^{-2}$ |
| + OH ⟶ 0.4 WSOC | | 6.4·10$^{9}$ [d] |
| (g) ⇌ | 2.34·10$^{5}$ | 2.34·10$^{-5}$ [e] |
| + OH ⟶ 0.4 WSOC | | 6.89·10$^{9}$ [f] |

| Process[a] | UPT[b] | RXN/RXN$_{0.4}$ [c] |
|---|---|---|
| (structure) ⇌ (structure) [g] | $5.92 \cdot 10^6$ | $5.92 \cdot 10^{6}$ [e] |
| (structure) + OH ⟶ 0.4 WSOC | | $1.19 \cdot 10^{10}$ [f] |
| (structure) ⇌ (structure) [g] | $3.81 \cdot 10^1$ | $3.81 \cdot 10^{1}$ [e] |
| (structure) + OH ⟶ 0.4 WSOC | | $2.00 \cdot 10^9$ |
| (structure) ⇌ (structure) [g] | $1.02 \cdot 10^1$ | $1.02 \cdot 10^{1}$ [e] |
| (structure) [g] + OH ⟶ 0.4 WSOC | | $1.30 \cdot 10^9$ |
| (structure) ⇌ (structure) [g] | $6.92 \cdot 10^4$ | $6.92 \cdot 10^4$ |
| (structure) + OH ⟶ 0.4 WSOC | | $2.29 \cdot 10^9$ |
| (structure) [g] ⇌ (structure) | $2.23 \cdot 10^4$ | $3.10 \cdot 10^3$ |
| (structure) + H$_2$O ⇌ (structure) | | $6.2$ |
| (structure) + OH ⟶ 0.4 WSOC | | $6.33 \cdot 10^{10}$ |
| (structure) + OH ⟶ 0.4 WSOC | | $1.06 \cdot 10^{11}$ |
| (structure) ⇌ (structure) [g] | $9.89 \cdot 10^{5}$ [g] | $9.89 \cdot 10^5$ |
| (structure) ⇌ (structure) + H$^+$ | | $2.32 \cdot 10^{-4}$ |
| (structure) + OH ⟶ 0.4 WSOC | | $2.47 \cdot 10^{10}$ |
| (structure) + OH ⟶ 0.4 WSOC | | $1.06 \cdot 10^{11}$ |
| (structure) [g] ⇌ (structure) | $6.76 \cdot 10^5$ | $1.70 \cdot 10^{-4}$ |
| (structure) + H$_2$O ⇌ (structure) | | $3.39 \cdot 10^1$ |
| (structure) + 2 H$_2$O ⇌ (structure) | | $3.15$ |
| (structure) + OH ⟶ 0.4 WSOC | | $1.5 \cdot 10^9$ |

| Process[a] | UPT[b] | RXN/RXN$_{0.4}$ [c] |
|---|---|---|
| [structure] + OH ⟶ 0.4 WSOC | | $1.26 \cdot 10^9$ |
| [structure] + OH ⟶ 0.4 WSOC | | $1.64 \cdot 10^9$ |
| [structure] (g) ⇌ [structure] | $7.07 \cdot 10^7$ | $1.21 \cdot 10^4$ |
| [structure] + 2 H$_2$O ⇌ [structure] | | $7.69$ |
| [structure] + 2 H$_2$O ⇌ [structure] | | $5.42 \cdot 10^1$ |
| [structure] + 3 H$_2$O ⇌ [structure] | | $3.07 \cdot 10^1$ |
| [structure] ⟶ 0.5 (0.2) [structure] + 0.5 (0.2) WSOC | | $1.00 \cdot 10^{-1}$ |
| [structure] + OH ⟶ 0.4 WSOC | | $1.12 \cdot 10^9$ |
| [structure] ⟶ 0.5 (0.2) [structure] + 0.5 (0.2) WSOC | | $1.00 \cdot 10^{-1}$ |
| [structure] + OH ⟶ 0.4 WSOC | | $1.29 \cdot 10^9$ |
| [structure] ⟶ 0.5 (0.2) [structure] + 0.5 (0.2) WSOC | | $1.00 \cdot 10^{-1}$ |
| [structure] + OH ⟶ 0.4 WSOC | | $1.31 \cdot 10^9$ |
| [structure] ⟶ 0.5 (0.2) [structure] + 0.5 (0.2) WSOC | | $1.00 \cdot 10^{-1}$ |
| [structure] + OH ⟶ 0.4 WSOC | | $1.72 \cdot 10^9$ |
| [structure] (g) ⇌ [structure] | $4.99 \cdot 10^6$ | $6.31 \cdot 10^2$ |
| [structure] + H$_2$O ⇌ [structure] | | $1.80 \cdot 10^3$ |
| [structure] + 2 H$_2$O ⇌ [structure] | | $3.02 \cdot 10^3$ |
| [structure] + OH ⟶ 0.4 WSOC | | $1.31 \cdot 10^9$ |
| [structure] + OH ⟶ 0.4 WSOC | | $1.41 \cdot 10^9$ |

Field Code Changed

Field Code Changed

Field Code Changed

Field Code Changed

Field Code Changed

Field Code Changed

Field Code Changed

| Process[a] | UPT[b] | RXN/RXN$_{0.4}$ [c] |
|---|---|---|
| [structure] + OH ⟶ 0.4 WSOC | | $1.97 \cdot 10^9$ |
| [structure] (g) ⇌ [structure] | $9.01 \cdot 10^{5}$ [g] | $2.05 \cdot 10^5$ |
| [structure] + H$_2$O ⇌ [structure] | | $1.90$ |
| [structure] + H$^+$ ⇌ [structure] | | $1.5 \cdot 10^{-3}$ |
| [structure] + OH ⟶ 0.4 WSOC | | $4.68 \cdot 10^8$ |
| [structure] + OH ⟶ 0.4 WSOC | | $5.50 \cdot 10^8$ |
| [structure] + OH ⟶ 0.4 WSOC | | $2.13 \cdot 10^8$ |
| [structure] (g) ⇌ [structure] | $2.14 \cdot 10^4$ | $1.84 \cdot 10^3$ |
| [structure] + H$_2$O ⇌ [structure] | | $3.00$ |
| [structure] ⟶ 0.5 (0.2) [structure] + 0.5 (0.2) WSOC | | $1.00 \cdot 10^{-1}$ |
| [structure] + OH ⟶ 0.4 WSOC | | $1.09 \cdot 10^8$ |
| [structure] ⟶ 0.5 (0.2) [structure] + 0.5 (0.2) WSOC | | $1.00 \cdot 10^{-1}$ |
| [structure] + OH ⟶ 0.4 WSOC | | $4.24 \cdot 10^9$ |
| [structure] (g) ⇌ [structure] | $7.75 \cdot 10^3$ | $2.85 \cdot 10^3$ |
| [structure] + H$_2$O ⇌ [structure] | | $1.72$ |
| [structure] + OH ⟶ 0.4 WSOC | | $5.68 \cdot 10^8$ |
| [structure] + OH ⟶ 0.4 WSOC | | $1.35 \cdot 10^9$ |
| [structure] (g) ⇌ [structure] | $1.70 \cdot 10^6$ | $8.69 \cdot 10^2$ |
| [structure] + H$_2$O ⇌ [structure] | | $1.95 \cdot 10^3$ |

Field Code Changed
Field Code Changed
Field Code Changed
Field Code Changed
Field Code Changed
Field Code Changed
Field Code Changed
Field Code Changed
Field Code Changed

| Process[a] | UPT[b] | RXN/RXN$_{0.4}$ [c] |
|---|---|---|
| (structure) + OH ⟶ 0.4 WSOC | | $1.08 \cdot 10^{10}$ |
| (structure) + OH ⟶ 0.4 WSOC | | $1.09 \cdot 10^{10}$ |
| (structure) (g) ⇌ (structure) | $2.54 \cdot 10^{9}$ | $6.96 \cdot 10^{5}$ |
| (structure) + H$_2$O ⇌ 0.9 (structure) + 0.1 (structure) | | $3.65 \cdot 10^{3}$ |
| (structure) + OH ⟶ 0.4 WSOC | | $1.43 \cdot 10^{9}$ |
| (structure) + OH ⟶ 0.4 WSOC | | $1.91 \cdot 10^{9}$ |
| (structure) + OH ⟶ 0.4 WSOC | | $2.29 \cdot 10^{9}$ |
| (structure) (g) ⇌ (structure) | $7.39 \cdot 10^{4}$ | $7.09 \cdot 10^{-1}$ |
| (structure) + H$_2$O ⇌ 0.06 (structure) + 0.94 (structure) | | $1.04 \cdot 10^{5}$ |
| (structure) + OH ⟶ 0.4 WSOC | | $1.07 \cdot 10^{9}$ |
| (structure) + OH ⟶ 0.4 WSOC | | $1.37 \cdot 10^{9}$ |
| (structure) + OH ⟶ 0.4 WSOC | | $1.83 \cdot 10^{9}$ |
| (structure) (g) ⇌ (structure) | $2.84 \cdot 10^{3}$ | $2.00 \cdot 10^{3}$ |
| (structure) + H$_2$O ⇌ (structure) | | $3.92 \cdot 10^{-1}$ |
| (structure) + OH ⟶ 0.4 WSOC | | $2.89 \cdot 10^{8}$ |
| (structure) + OH ⟶ 0.4 WSOC | | $6.80 \cdot 10^{8}$ |
| (structure) (g) ⇌ (structure) | $2.66 \cdot 10^{7}$ | $3.61 \cdot 10^{4}$ |
| (structure) + H$_2$O ⇌ (structure) | | $4.28 \cdot 10^{1}$ |

Field Code Changed

| Process[a] | UPT[b] | RXN/RXN$_{0.4}$[c] |
|---|---|---|
| (structure) + 2 H$_2$O ⇌ (structure) | | $9.85 \cdot 10^{1}$ |
| (structure) + 2 H$_2$O ⇌ (structure) | | $5.91 \cdot 10^{1}$ |
| (structure) + 3 H$_2$O ⇌ (structure) | | $1.00 \cdot 10^{-1}$ |
| (structure) → 0.5 (0.2) (structure) + 0.5 (0.2) (structure) | | $1.39 \cdot 10^{8}$ |
| (structure) + OH → 0.4 WSOC | | $1.00 \cdot 10^{-1}$ |
| (structure) → 0.5 (0.2) (structure) + 0.5 (0.2) (structure) | | $4.19 \cdot 10^{8}$ |
| (structure) + OH → 0.4 WSOC | | $1.00 \cdot 10^{-1}$ |
| (structure) → 0.25 (0.1) (structure) + 0.25 (0.1) (structure) + 0.25 (0.1) (structure) + 0.25 (0.1) (structure) | | $8.22 \cdot 10^{8}$ |
| (structure) + OH → 0.4 WSOC | | $1.00 \cdot 10^{-1}$ |
| (structure) → 0.5 (0.2) (structure) + 0.5 (0.2) (structure) | | $8.11 \cdot 10^{8}$ |
| (structure) + OH → 0.4 WSOC | | $1.00 \cdot 10^{-1}$ |
| (structure) → 0.5 (0.2) (structure) + 0.5 (0.2) (structure) | | $1.23 \cdot 10^{9}$ |
| (structure) + OH → 0.4 WSOC | | $1.00 \cdot 10^{-1}$ |
| (structure) (g) ⇌ (structure) | $8.16 \cdot 10^{5}$ | $1.40 \cdot 10^{2}$ |
| (structure) + 2 H$_2$O ⇌ (structure) | | $6.28 \cdot 10^{1}$ |
| (structure) + OH → 0.4 WSOC | | $1.91 \cdot 10^{9}$ |
| (structure) + OH → 0.4 WSOC | | $1.68 \cdot 10^{9}$ |

[a] Branching ratios from the scenario RXN$_{0.4}$ are given in blue, from all other scenarios in black.

[b] Column shows $K_{H,eff}$ in M$^{n}$ atm$^{-1}$ (n = order of reaction) for the scenario UPT. Data according to the new CAPRAM/GECKO-A protocol unless reference is indicated by further footnotes.

[c] Column shows $K_H$ in M atm$^{-1}$, $K_{hyd}$ in M$^{m-n}$, $K_{diss}$ in M, $k_n$ in M$^{-(n-1)}$ s$^{-1}$ for scenarios RXN and RXN$_{0.4}$. Data according to the new CAPRAM/GECKO-A protocol unless reference is indicated by further footnotes.

[d] Sehested et al. (1975)

[e] Estimated with EPI Suite 4.1 (http://www.epa.gov/opptintr/exposure/pubs/episuite.htm)

[f] Rate constant calculated with the SAR by Minakata et al. (2009) ignoring the oxygen atoms in the –O–O– group as this group is bound solely to quaternary carbon atoms and thus in β-position of any reactive group. As the group itself contains no H-atoms and is, therefore, unreactive and no β-parameters are used in the SAR βυ Minakata et al. (2009), it was possible to derive a rate constant with the SAR without further assumptions.

[g] Contributions of dissociation to the effective Henry's Law constant calculated manually as GROMHE does not consider dissociations of carboxylic acids.

[Figure]

**Figure S16 Modelled particle pH in the different sensitivity runs for the mesitylene oxidation experiment.**

[Figure]

**Figure S17 Malic acid concentrations as monitored by PTR-MS in the LEAK chamber during the mesitylene oxidation experiment.**

[Figure]

**Figure S18 Concentrations-time profiles of the major gas phase oxidation products methylglyoxal (a) and acetic acid (b) in the sensitivity runs TMB, UPT, RXN$_{0.4}$, and RXN.**